# Summarizing the state of the terrestrial biosphere in few dimensions

Guido Kraemer[1,2,3], Gustau Camps-Valls[2], Markus Reichstein[1,3], and Miguel D. Mahecha[1,3]

[1]Max Planck Institute for Biogeochemistry, 07745 Jena, Germany
[2]Image Processing Laboratory, Universitat de València, 46980 Paterna (València), Spain
[3]German Centre for Integrative Biodiversity Research (iDiv) Halle-Jena-Leipzig, 04103 Leipzig, Germany

**Correspondence:** Guido Kraemer (gkraemer@bgc-jena.mpg.de)

**Abstract.** In times of global change, we must closely monitor the state of the planet in order to understand the full complexity of these changes. In fact, each of the Earth's subsystems—i.e. the biosphere, atmosphere, hydrosphere, and cryosphere—can be analyzed from a multitude of data streams. However, since it is very hard to jointly interpret multiple monitoring data streams in parallel, one often aims for some summarizing indicator. Climate indices, for example, summarize the state of atmospheric circulation in a region. Although such approaches are also used in other fields of science, they are rarely used to describe land surface dynamics. Here, we propose a robust method to create global indicators for the terrestrial biosphere using principal component analysis based on a high-dimensional set of relevant global data streams. The concept was tested using 12 explanatory variables representing the biophysical state of ecosystems and land-atmosphere water, energy, and carbon fluxes. We find that three indicators account for 82% of the variance of the selected biosphere variables in space and time across the globe. While the first indicator summarizes productivity patterns, the second indicator summarizes variables representing water and energy availability. The third indicator represents mostly changes in surface albedo. Anomalies in the indicators clearly identify extreme events, such as the Amazon droughts (2005 and 2010) and the Russian heatwave (2010). The anomalies also allow us to interpret the impacts of these events. The indicators can also be used to detect and quantify changes in seasonal dynamics. Here we report for instance increasing seasonal amplitudes of productivity in agricultural areas and arctic regions. We assume that this generic approach has great potential for the analysis of land-surface dynamics from observational or model data.

## 1 Introduction

Today, humanity faces negative global impacts of land use and land cover change (Song et al., 2018), global warming (IPCC, 2014), and associated losses of biodiversity (IPBES, 2019; Díaz et al., 2019), to only mention the most prominent transformations. Over the past decades, new satellite missions (e.g. Berger et al., 2012; Schimel and Schneider, 2019), along with the continuous collection of ground based measurements (e.g. Wingate et al., 2015; Nasahara and Nagai, 2015; Baldocchi, 2020), and the integration of both (Papale et al., 2015; Babst et al., 2017; Jung et al., 2019) have increased our capacity to monitor the Earth's surface enormously. However, there are still large knowledge gaps limiting our capacity to monitor and understand the current transformations of the Earth system (Steffen et al., 2015; Rosenfeld et al., 2019; Yan et al., 2019; Piao et al., 2020).

Many of recent changes due to increasing anthropogenic activity are manifested in long-term transformations. One prominent example is "global greening" that has been attributed to fertilization effects, temperature increases, and land-use intensification (de Jong et al., 2011; Zhu et al., 2016; Piao et al., 2019). It is also known that phenological patterns change in the wake of climate change (Schwartz, 1998; Parmesan, 2006). However, these phenological patterns vary regionally. In "cold" ecosystems one may find decreased seasonal amplitudes on primary production due to warmer winters (Stine et al., 2009). Elsewhere, seasonal amplitude may increase e.g. in agricultural areas due to the so called "green revolution" (Zeng et al., 2014; Chen et al., 2019). Another change in terrestrial land-surface dynamics is induced by increasing frequencies and magnitudes of extreme events (Barriopedro et al., 2011; Reichstein et al., 2013). The consequences for land-ecosystems have yet to be fully understood (Flach et al., 2018; Sippel et al., 2018), and require novel detection and attribution methods tailored to the problem (Flach et al., 2017; Mahecha et al., 2007a). While extreme events are typically only temporary deviations from a normal trajectory, ecosystems may change their qualitative state permanently, for example shift from grassland to shrubland. Such shifts or tipping points can be induced by changing environmental conditions or direct human influence, and pose yet another problem that needs to be considered (Lenton et al., 2008). The question we address here is, how to uncover and summarize changes in land-surface dynamics in a consistent framework. The idea is to simultaneously take advantage of a large array of global data streams, without addressing each observed phenomenon in a specific domain only. We seek to develop an integrated approach to uncover changes in the land-surface dynamics based on a very generic approach.

The problem of identifying patterns of change in high-dimensional data streams is not new. Extracting the dominant features from high-dimensional observations is a well-known problem in many disciplines. One approach is to manually define indicators that are known to represent important properties such as the "Bowen Ratio" (Bowen, 1926, find a more complete description of the concept in Section 3.3), another one consists in using machine learning to extract unique, and ideally independent features from the data. In the climate sciences, for instance, it is common to summarize atmospheric states using Empirical Orthogonal Functions (EOF), also known as Principal Component Analysis (PCA; Pearson, 1901). The rationale is that dimensionality reduction only retains the main data features, which makes them easier accessible for analysis. One of the most prominent examples is the description of the El Niño Southern Oscillation (ENSO) dynamics in the multivariate ENSO index (MEI; Wolter and Timlin, 2011), an indicator describing the state of the regional circulation patterns at a certain point in time. The MEI is a very successful index that can be easily interpreted and used in a variety of ways, most basically it provides a measure for the intensity and duration of the different quasi-cyclic ENSO events but it can also be associated with its characteristic impacts: E.g. seasonal warming, changes in seasonal temperatures and overall dryness in the Pacific Northwest of the United States (Abatzoglou et al., 2014), drought related fires in the Brazilian Amazon (Aragão et al., 2018), and crop yield anomalies (Najafi et al., 2019).

In plant ecology, indicators based on dimensionality reduction methods are used to describe changes to species assemblages along unknown gradients (Legendre and Legendre, 1998; Mahecha et al., 2007a). The emerging gradients can be interpreted using additional environmental constraints, or based on internal plant community dynamics (van der Maaten et al., 2012). It is also common to compress satellite based Earth Observations via dimensionality reduction to get a notion of the underlying dynamics of terrestrial ecosystems. For instance, Ivits et al. (2014) showed that one can understand the impacts of droughts and

heatwaves based on a compressed view of the relevant vegetation indices. In general, dimensionality reduction is the method of choice to compress high-dimensional observations in a few (ideally) independent components with little loss of information (Van Der Maaten et al., 2009; Kraemer et al., 2018).

Understanding changes in land-atmosphere interactions is a complex problem, as all aforementioned patterns of change may occur and interact: land cover change may alter biophysical properties of the land surface such as (surface) albedo with consequences for the energy balance (Song et al., 2018). Long-term trends in temperature, water availability, or fertilization may impact productivity patterns and biogeochemical processes (Zhu et al., 2016; Sitch et al., 2015). In fact, these land surface dynamics have implications on multiple dimensions and require monitoring of biophysical state variables such as leaf area index, albedo, etc., as well as associated land-atmosphere fluxes of carbon, water, and energy.

Here, we aim to summarize these high-dimensional surface dynamics and make them accessible to subsequent interpretations and analyses such as mean seasonal cycles (MSC), anomalies, trend analyses, breakpoint analyses, and the characterization of ecosystems. Specifically, we seek a set of uncorrelated, yet comprehensive, state indicators. We want to have a set of very few indicators that represent the most dominant features of the above described temporal ecosystem dynamics. These indicators should also be uncorrelated, so that one can study the system state by looking and interpreting each indicator independently. The approach should also give an idea of the general complexity contained in the available data streams. If more than a single indicator is required to describe land surface dynamics accurately, then these indicators shall describe very different aspects. While one indicator may describe global patterns of change, others could be only relevant in certain regions, for certain types of ecosystems, or for specific types of impacts. The indicators shall have a number of desirable properties: (1) Representing the overall state of observations comprising the system in space and time. (2) Carrying sufficient information to allow for reconstructing the original observations faithfully from these indicators. (3) Being of much lower dimensionality than the number of observed variables. (4) Allowing intuitive interpretations.

In this work, we first introduce a method to create such indicators, then we apply the method to a global set of variables describing the biosphere. Finally, to prove the effectiveness of the method, we interpret the resulting set of indicators and explore the information contained in the indicators by analyzing them in different ways and relating them to well known phenomena.

## 2 Methods

### 2.1 Data

Table 1 gives an overview of the data streams used in this analysis (for a more detailed description see Appendix A). For an effective joint analysis of more than a single variable, the variables have to be harmonized and brought to a single grid in space and time. The Earth System Data Lab (ESDL; www.earthsystemdatalab.net; Mahecha et al., 2019) curates a comprehensive set of data streams to describe multiple facets of the terrestrial biosphere and associated climate system. The data streams are harmonized as analysis ready data on a common spatiotemporal grid (equirectangular $0.25°$ grid in space and 8 days in time, 2001–2011), forming a 4d hypercube, which we call a "data cube". The ESDL not only curates Earth system data, but also

**Table 1.** Variables used describing the biosphere, for a description of the variables, see Appendix A.

| Variable | Details | Source |
|---|---|---|
| Black Sky Albedo | Directional reflectance | Muller et al. (2011) |
| Evaporation | $[\text{mm day}^{-1}]$ | Martens et al. (2017) |
| Evaporative Stress | Modeled water stress | Martens et al. (2017) |
| fAPAR | fraction of absorbed photosynthetically active radiation | Disney et al. (2016) |
| Gross Primary Productivity (GPP) | $[\text{gCm}^{-2}\text{day}^{-1}]$ | Tramontana et al. (2016); Jung et al. (2019) |
| Latent energy (LE) | $[\text{Wm}^{-2}]$ | Tramontana et al. (2016); Jung et al. (2019) |
| Net Ecosystem Exchange (NEE) | $[\text{gCm}^{-2}\text{day}^{-1}]$ | Tramontana et al. (2016); Jung et al. (2019) |
| Root-Zone Soil Moisture | $[\text{m}^3\text{m}^{-3}]$ | Martens et al. (2017) |
| Sensible Heat (H) | $[\text{Wm}^{-2}]$ | Tramontana et al. (2016); Jung et al. (2019) |
| Surface Soil Moisture | $[\text{mm}^3\text{mm}^{-3}]$ | Martens et al. (2017) |
| Terrestrial Ecosystem Respiration (TER) | $[\text{gCm}^{-2}\text{day}^{-1}]$ | Tramontana et al. (2016); Jung et al. (2019) |
| White Sky Albedo | Diffuse reflectance | Muller et al. (2011) |

comes with a toolbox to analyze this data efficiently. For this study, we chose all available variables in the ESDL v1.0 (the most recent version available at the time of analysis), divided the available variable into meteorological and biospheric variables and
discarded the atmospheric variables. We also discarded variables with distributions that are badly suited for a linear PCA (e.g. burnt area contains mostly zeros) and variables with too many missing values. The only dataset that was added post hoc was fAPAR which represents an important aspect of vegetation which was not available in the data cube at the time of analysis (it is part of the most recent version of the data cube).

The datasets taken from Tramontana et al. (2016); Jung et al. (2019) are derived from flux tower measurements (Baldocchi,
2020). The flux towers are not equally distributed in climate space, i.e. there are many flux towers in temperate areas, but much less in tropic and arctic regions, which may lead to less accurate data in these regions. These data sets also exclude large arid areas such as the Sahara and Gobi deserts and parts of the Arabian Peninsula which may affect the resulting loadings of the PCA slightly.

In this study, each variable was normalized globally to zero mean and unit variance to account for the different units of the
variables, i.e. transform the variables to have standard deviations from the mean as the common unit. Because the area of the pixel changes with latitude in the equirectangular coordinate system used by the ESDL, the pixels were weighted according to the represented surface area. Only spatiotemporal pixels without any missing values were considered in the calculation of the covariance matrix.

## 2.2 Dimensionality Reduction with PCA

As a method for dimensionality reduction, we used a modified principal component analysis to summarize the information contained in the observed variables. PCA transforms the set of $d$ centered and, in this case, standardized variables into a subset

of $p$, $1 \leq p \leq d$, principal components (PCs). Each component is uncorrelated with the other components, while the first PCs explain the largest fraction of variance in the data.

The data streams consist of $d = 12$ observed variables at the same time and location. Each observation is defined in a $d$-dimensional space, $\mathbf{x}_i \in \mathbb{R}^d$, and we define the dataset by collecting all samples in the matrix $\mathbf{X} = [\mathbf{x}_1 | \cdots | \mathbf{x}_n] \in \mathbb{R}^{d \times n}$. The observations are repeated in space and time and lie on a grid of lat $\times$ lon $\times$ time. In our case, we have $n = |\text{lat}| \times |\text{lon}| \times |\text{time}| = 720 \times 1440 \times 506 = 524,620,800$ observations, where $|\cdot|$ denotes the cardinality of the dimension. Note that the actual number of observations was lower, $n = 106,360,156$, because we considered land points only and removed missing values.

The fundamental idea of PCA is to project the data to a space of lower dimensionality that preseerves the covariance structure of the data. Hence, the fundament of a PCA is the computation of a covariance matrix, $\mathbf{Q}$. When all variables are centered to global zero mean and normalized to unit variance, the covariance matrix can be in principle estimated as

$$\mathbf{Q} = \frac{1}{n-1}\mathbf{X}\mathbf{X}^T = \frac{1}{n-1}\sum_{i=1}^{n}\mathbf{x}_i\mathbf{x}_i^T. \tag{1}$$

However, in our case the data cube lies on a regular $0.25°$ grid and estimating $\mathbf{Q}$ as above would lead to overestimating the influence of dynamics in relatively small pixels of high latitudes compared to lower latitudes where each data point represent larger areas. Hence, one needs a weighted approach to calculate the covariance matrix,

$$\mathbf{Q} = \frac{1}{w}\sum_{i=1}^{n}w_i\mathbf{x}_i\mathbf{x}_i^T, \tag{2}$$

where $w_i = \cos(\text{lat}_i)$ and $\text{lat}_i$ is the latitude of observation $i$, $w = \sum_{i=1}^{n} w_i$ is the total weight, and $n$ the total number of observations. Equation (2) has the additional property that it can be computed sequentially on very big datasets, such as our Earth System Data Cube, by a consecutively adding observations to an initial estimate.

Note that the actual calculation of the covariance matrix is even more complicated, because summing up many floating-point numbers one-by-one can lead to large inaccuracies due to precision issues of floating-point numbers and instabilities of the naive algorithm (Higham, 1993; the same holds for the implementations of the `sum` function in most software used for numerical computing). Here, we used the Julia package `WeightedOnlineStats.jl`[1] (implemented by the first author of this paper), which uses numerically stable algorithms for summation, higher precision numbers, and a map-reduce scheme that further minimizes floating point errors.

Based on this weighted and numerically stable covariance matrix, the PCA can be computed using an eigendecomposition of the covariance matrix,

$$\mathbf{Q} = \mathbf{V}\mathbf{\Lambda}\mathbf{V}^T \in \mathbb{R}^{d \times d}. \tag{3}$$

In this case, the covariance matrix $\mathbf{Q}$ is equal to the correlation matrix because we standardized the variables to unit variance. $\mathbf{\Lambda}$ is a diagonal matrix with the the eigenvalues, $\lambda_1, \ldots, \lambda_d$, in the diagonal in decreasing order and $\mathbf{V} \in \mathbb{R}^{d \times d}$, the matrix with

---

[1]DOI: 10.5281/zenodo.3360311, repository: https://github.com/gdkrmr/WeightedOnlineStats.jl/

the corresponding eigenvectors in columns. $\mathbf{V}$ can project the new incoming input data $\mathbf{x}_i$ (centered and standardized) onto the retained PCs,

$$\mathbf{y}_i = \mathbf{V}^T \mathbf{x}_i \in \mathbb{R}^d, \tag{4}$$

where $\mathbf{y}_i$ is the projection of the observation $\mathbf{x}_i$ onto the $d$ PCs.

The canonical measure of the quality of a PCA is the fraction of explained variance by each component, $\sigma_i^2$, calculated as

$$\sigma_i^2 = \frac{\lambda_i}{\sum_{i=1}^d \lambda_i}. \tag{5}$$

To get a more complete measure of the accuracy of the PCA, we used the "reconstruction error" in addition to the fraction of explained variance. PCA allows a simple projection of an observation onto the first $p$ PCs and a consecutive reconstruction of the observations from this $p$-dimensional projection. This is achieved by

$$\mathbf{Y}_p = \mathbf{V}_p^T \mathbf{X} \in \mathbb{R}^{p \times n} \text{ and } \mathbf{X}_p = \mathbf{V}_p \mathbf{Y}_p \in \mathbb{R}^{d \times n}, \tag{6}$$

where $\mathbf{Y}_p$ is the projection onto the first $p$ PCs, $\mathbf{V}_p$ the matrix with columns consisting of the eigenvectors belonging to the $p$ largest eigenvalues, and $\mathbf{X}_p$ the observations reconstructed from the first $p$ PCs.

The reconstruction error, $\mathbf{e}_i$, was calculated for every point, $\mathbf{x}_i$ in the space–time domain based on the reconstructions from the first $p$ principal components:

$$\mathbf{e}_i = \mathbf{V}_p \mathbf{V}_p^T \mathbf{x}_i - \mathbf{x}_i \in \mathbb{R}^d. \tag{7}$$

As this error is explicit in space, time and variable, it allows for disentangling the contribution of each of these domains to the total error. This can be achieved by estimating e.g. the (weighed) mean square error,

$$\mathrm{MSE} = \frac{1}{w} \sum_i w_i \mathbf{e}_i^2. \tag{8}$$

This approach can give a better insight into the compositions of the error than a single global error estimate based on the
eigenvalues.

## 2.3    Pixel-wise analyses of time series

The principal components estimated as described above are ideally low-dimensional representations of the land-surface dynamics that require further interpretation. These components have a temporal dynamics that needs to be understood in detail. One crucial question is how the dynamics of a system of interest deviates from it's expected behaviour at some point in time.
A classical approach is inspecting the "anomalies" of a time series, i.e. the deviation from the mean seasonal cycle at a certain day of year.

Another key description of such system dynamics are trends. We estimated trends of the indicators as well as of their seasonal amplitude using the Theil–Sen estimator. The advantage of the Theil–Sen estimator is its robustness to up to 29.3% of outliers

(Theil, 1950; Sen, 1968), while ordinary least squares regression is highly sensitive to such values. The calculation of the estimator consists simply on computing the median of the slopes spanned by all possible pairs of points

$$\text{slope}_{ij} = \frac{z_i - z_j}{t_i - t_j}, \tag{9}$$

where $z_i$ is the value of the response variable at time step $i$ and $t_i$ the time at time step $i$. In our experiments, we computed the slopes separately per pixel and principal component with time as the predictor and the value of the principal component as the response variable.

To test the slopes for significance, we used the Mann-Kendall statistics (Mann, 1945; Kendall, 1970) and adjusted the resulting $p$-values with the Benjamini-Hochberg method to control for the false discovery rate (Benjamini and Hochberg, 1995). Slopes with an adjusted $p < 0.05$ were deemed significant.

To identify disruptions in trajectories, breakpoint detection provides a good framework for analysis. For the estimation of breakpoints, the generalized fluctuation test framework (Kuan and Hornik, 1995) was used to test for the presence of breakpoints. The framework uses recursive residuals (Brown et al., 1975) such that a breakpoint is identified when the mean of the recursive residuals deviates from zero. We used the implementation in Zeileis et al. (2002). For practical reasons, here we only focus on the largest breakpoint.

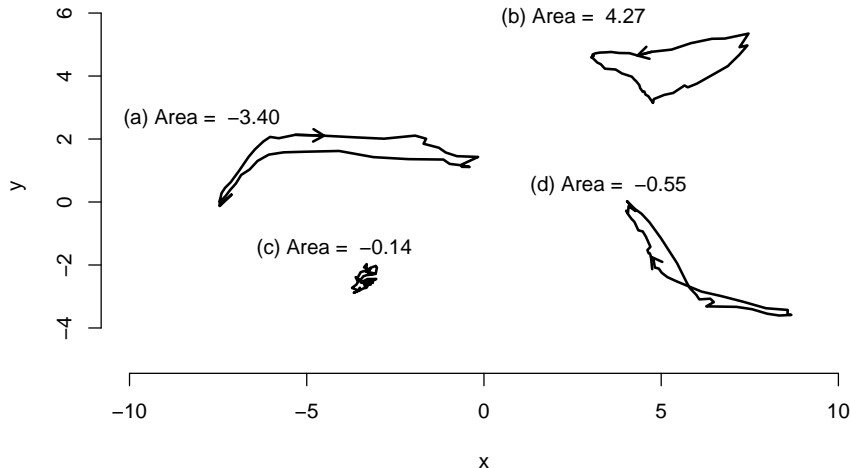

**Figure 1.** Example polygons and their areas, Eq. 10, the arrows indicate the directionality. (a) Clockwise polygon, has a negative area. (b) Counterclockwise polygon, has a positive area. (c) Chaotic polygon, has a very low area. (d) Polygon with a single intersection, has both a clockwise and counterclockwise portion. The clockwise portion is slightly larger than the counterclockwise portion, therefore the area is slightly negative.

The analysis of a different type of dynamic considers bivariate relations. In the context of oscillating signals it is particularly instructive to quantify their degree of phase shift and direction—even if both signals are not linearly related. A "hysteresis" would be such a pattern describing that the pathways $A \rightarrow B$ and $B \rightarrow A$ between states $A$ and $B$ differ (Beisner et al., 2003). We estimated hysteresis by calculating the area inside the polygon formed by the mean seasonal cycle of the combinations of

two components.

$$\text{Area} = \frac{1}{2} \sum_{i=1}^{n} x_i(y_{i+1} - y_{i-1}), \tag{10}$$

where $n = 46$, the number of time steps in a year, $x_i$ and $y_i$ the mean seasonal cycle of two PCs at time step $i$, respectively.
The polygon is circular, i.e. the indices wrap around the edges of the polygon so that $x_0 = x_n$ and $x_{n+1} = x_1$. This formula gives the actual area inside the polygon only if it is non-self-intersecting and the vertices run counterclockwise. If the vertices run clockwise, the area is negative. If the polygon is shaped like an 8, the clockwise and counterclockwise parts will cancel each other (partially) out. Trajectories that have a larger amplitudes will also tend to have larger areas as illustrated in fig. 1.

## 3 Results and Discussion

In the following, we first briefly present and discuss the quality of the global dimensionality reduction (Sect. 3.1) and interpret the individual components from an ecological point of view (Sect. 3.2). We summarize the global dynamics that we uncovered in the low-dimensional space (Sect. 3.3). We characterize the contained seasonal dynamics (Sect. 3.4), including spatial patterns of hysteresis (Sect. 3.5). We then describe global anomalies of the identified trajectories (Sect. 3.6), and discuss the identified anomalies in depth based on local phenomena (Sect. 3.7). Finally, we present global trends and their breakpoints (Sect. 3.7).

### 3.1 Quality of the PCA

Figure 2a shows the explained fraction of variance (Eq. 5) for the global PCA based on the entire data cube. The two leading components explain 73% of the variance from the 12 variables; additional components contribute relatively little additional variance ($PC_3$ contributes 9%, all subsequent PCs less than 7%) each. This results in a "knee" at component 3, which suggests that two indicators are sufficient to capture the major global dynamics of the terrestrial land surface, but we will also consider the third components in the following analyses (Cattell, 1966).

We estimated the reconstruction error sequentially up to the first three principal components (fig. 3). Regions that do not fit the model well show a higher reconstruction error. Considering one component only, highest reconstruction errors appear in high latitudes but decrease strongly with each additional component and nearly vanish if the third component is included.

### 3.2 Interpretation of the PCA

The first PC summarizes variables that are closely related to primary productivity (GPP, LE, NEE, fAPAR), and therefore are highly interrelated (see fig. 2b). The energy for photosynthesis comes from solar radiation, fAPAR is an indicator for the fraction of light used for photosynthesis. The available photosynthetic radiation is used by photosynthesis to fix $CO_2$ and to produce sugars that maintain the metabolism of the plant. The total uptake of $CO_2$ is reflected in GPP, which is also closely related to water consumption. The flow of water within the plant is not only essential to enable photosynthesis but also drives the transport of nutrients from the roots. The uplift of water in the plant is ultimately driven by transpiration—together with evaporation from soil surfaces one can observe the integrated latent energy needed for the phase transition (LE). However,

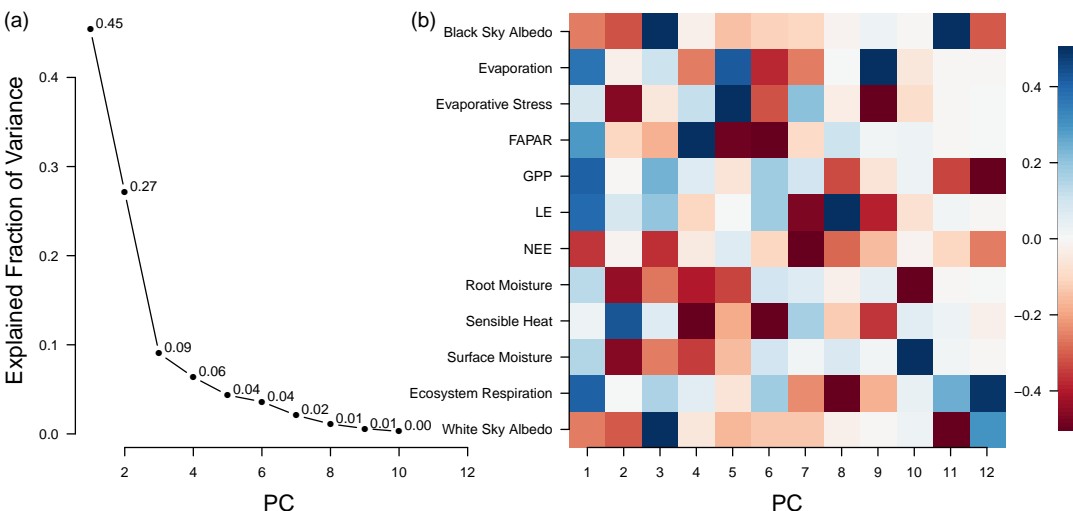

**Figure 2.** (a) Fraction of explained variance of the PCA by component. Knee at component three suggest that components four and higher do not contribute much to total variance. (b) Rotation matrix of the global PCA model (also called *loadings*, eq. 4). The columns of the rotation matrix describe the linear combinations of the (centered and standardized) original variables that make up the principal components. $PC_1$ is dominated by primary productivity related variables, $PC_2$ two by variables describing water availability, $PC_3$ by variables describing albedo. Values of the rotation matrix are clamped to the range $[-0.5, 0.5]$, the actual range of the values is $[-0.73, 0.74]$, and $[-0.46, 0.54]$ for the first three components.

ecosystems also respire; $CO_2$ is produced by plants in energy consuming processes as well as by the decomposition of dead organic materials via soil microbes and other heterotrophic organisms. This total respiration can be observed as terrestrial ecosystem respiration (TER). The difference between GPP and TER is the net ecosystem exchange (NEE) rate of $CO_2$ between ecosystems and the atmosphere (Chapin et al., 2006). GPP and TER are also well represented in the first dimension.

The second component represents variables related to the surface hydrology of ecosystems (see fig. 2b). Surface moisture, evaporative stress, root-zone soil moisture, and sensible heat, are all essential indicators for the state of plant available water. While surface moisture is a rather direct measure, evaporative stress is a modeled quantity summarizing the level of plant stress: A value of zero means that there is no water available for transpiration, while a value of one means that transpiration equals the potential transpiration (Martens et al., 2017). Root-zone soil moisture is the moisture content of the root zone in the soil, the moisture directly available for root uptake. If this quantity is below the wilting point, there is no water available for uptake by the plants. Sensible heat is the exchange of energy by a change of temperature, if there is enough water available, then most of the surface heat will be lost due to evaporation (latent heat), with decreasing water availability more of the surface heat will be lost due to sensible heat, making this also an indicator of dryness.

We observe that the third component is most strongly related to albedo (fig. 2b). Albedo describes the overall reflectiveness of a surface, here we refer to broadband (400-3000nm) surface albedo, for an exact definitions see Appendix A. Light surfaces, such as snow and sand, reflect most of the incoming radiation, while surfaces that have a high liquid water content or active

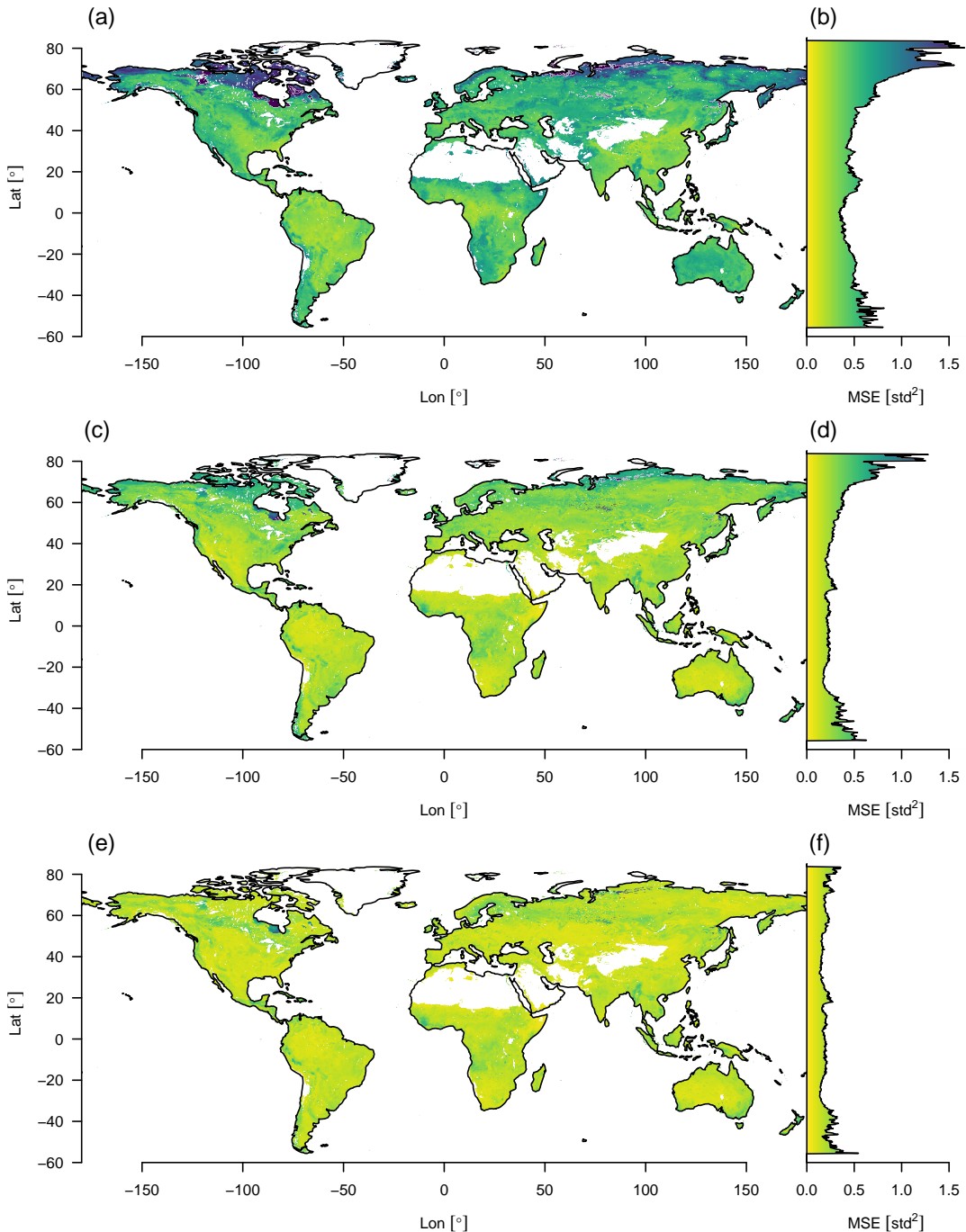

**Figure 3.** Reconstruction error of the data cube using varying numbers of principal components aggregated by the mean squared error. Reconstruction errors aggregated over all time steps and variables are shown in the left column: (a) Using only the first component; (c) Using the first two; (e) Using the first three. Corresponding right plots (b, d, f) show the mean reconstruction error aggregated by latitudes.

vegetation absorb most of the incoming radiation. Local changes to albedo can be caused by many reasons, e.g. snow fall, vegetation greening/browning, or land use change.

The relation of $PC_3$ to productivity and hydrology is opposite to what we would expect from an albedo axis. Because vegetation uses radiation as an energy source, albedo is negatively correlated with the productivity of vegetation, hence the negative correlation of albedo with $PC_1$. Given that water also absorbs radiation we can observe a negative correlation of albedo with $PC_2$ (see fig. 2b). We observe that $PC_1$ and $PC_2$ are positively correlated with $PC_3$ on the positive portion of their axes (see fig. 4d and f), which means counterintuitively that the index representing albedo is positively correlated with primary productivity and moisture content. Finally we can observe that $PC_1$ and $PC_2$ have a much higher reconstruction error in snow covered regions, which is strongly improved by adding $PC_3$ (see fig. 3f). Therefore the third component should be regarded mostly as binary variable that introduces snow cover, as the other information that is usually associated with albedo is already contained in the first two components.

### 3.3 Distribution of points in PCA space

The bivariate distribution of the first two principal components forms a "triangle" (gray background in fig. 4a). At the high end of $PC_1$ we find one point of the triangle in which ecosystems have a high primary productivity (high values of GPP, fAPAR, LE, TER, and evaporation), mostly limited by radiation. On the lower end of the principal component one we find the other two points of the triangle describing two alternative states of low productivity: These can happen either when the second principal component coincides with temperature limitation (the negative extreme of the second principal component) as seen in the lower left corner of the distribution in fig. 4a and b or due to water limitation (positive extreme of the second principal component, the upper left corner in fig. 4a). This pattern reflects the two essential global limitations of GPP in terrestrial ecosystems (Anav et al., 2015).

Both components form a subspace in which most of the variability of ecosystems takes place. Component one describes productivity and component two the limiting factors to productivity. Therefore, we can see that most ecosystems with high values on component one (a high productivity) are at the approximate center of component two. When ecosystems are found outside the center of component two, they have lower values on component one (lower productivity) because they are limited by water or temperature (see fig. 4b).

To further interpret the "triangle" we analyze how the Bowen ratio embeds into the space of the first two dimensions. Energy fluxes from the surface into the atmosphere can either represent a radiative transfer (sensible heat) or evaporation (latent heat). Their ratio is the "Bowen ratio", $B = \frac{H}{LE}$, (Bowen, 1926; see also fig. 5). When water is available most of the available energy will be dissipated by evaporation, $B < 1$, resulting in a high latent heat flux. Otherwise, the transfer by latent heat will be low and most of the incoming energy has to be dissipated via sensible heat, $B > 1$. In higher latitudes, there is relatively limited incoming radiation and temperatures are low, therefore there is not much energy to be dissipated and both heat fluxes are low. A high sensible heat flux is an indicator for water limitation.

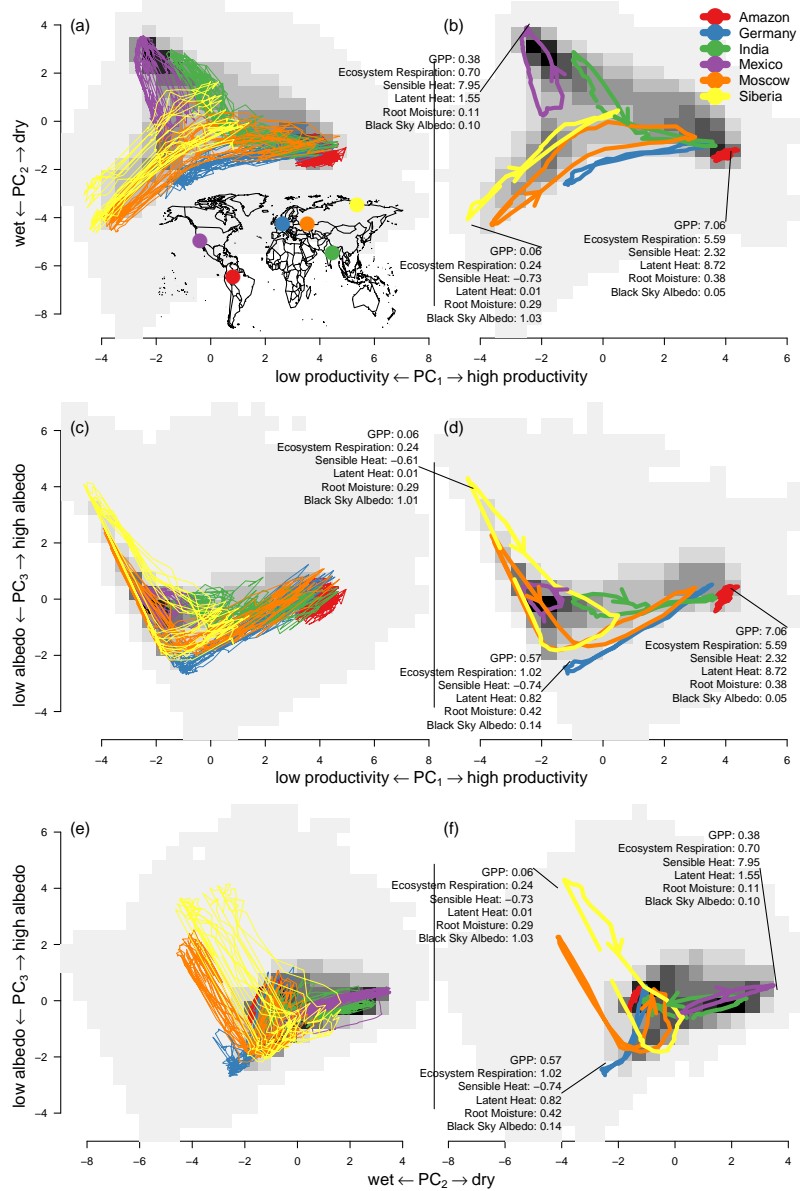

**Figure 4.** Trajectories of some points (colored lines) and the area weighted density over principal components one and two (the gray background shading shows the density) for (left column) the raw trajectories and (right column) the mean seasonal cycle. The trajectories are shown in the space of PC$_1$–PC$_2$ (first row), PC$_1$–PC$_3$ (second row), and PC$_2$–PC$_3$ (third row). The trajectories were chosen to cover a large area in the space of the first two principal components. Some of the trajectories have an arrow indicating the direction. The numbers illustrate the value of some variables, for units see tab. 1. Description of the points: Red: Tropical Rainforest, 67.625°W, 2.625°S; Blue: Maritime climate, 7.375°E, 52.375°N; Green: Monsoon climate, 82.375°E, 22.375°N; Purple: Subtropical, 117.625°W, 34.875°N; Orange: Continental climate, 44.875°E, 52.375°N; Yellow: Arctic climate, 119.875°E, 72.375°N.

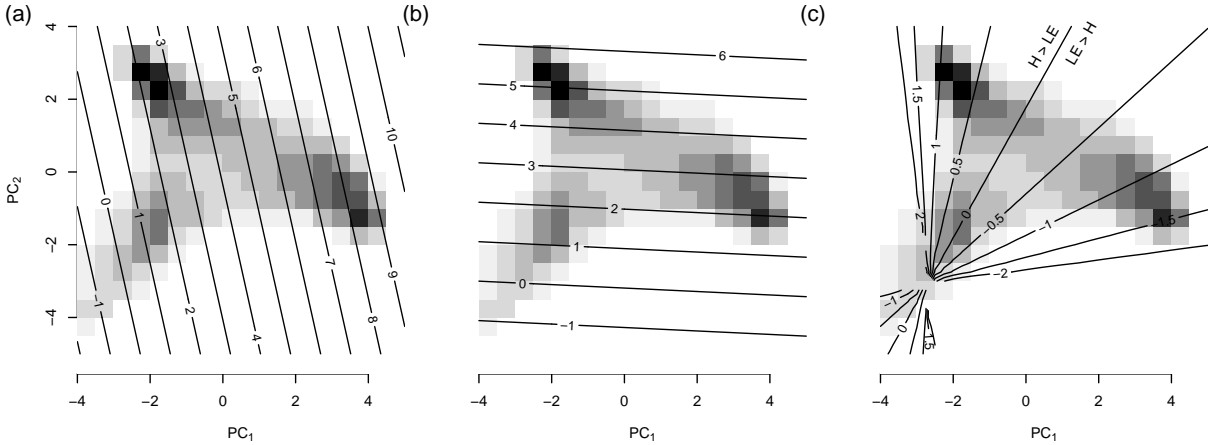

**Figure 5.** The background shading show the distribution of the mean seasonal cycle of the spatial points (see fig. 4). The contour lines represent the reconstruction of the variables from the first two principal components. The reconstructed variables are (a) Latent Heat (LE), (b) Sensible heat (H), and (c) $\log_{10}\left(\frac{\text{Sensible Heat}}{\text{Latent Heat}}\right)$, the $\log_{10}$ of the Bowen ratio. Note that the LE and H have been considered in the construction of the PCs, and hence are a linear function of the PCs. The Bowen ratio, instead, was not considered here and clearly responds in a nonlinear form.

## 3.4 Seasonal Dynamics

The leading principal components represent most of the variability of the space spanned by the observed variables, summarizing the state of a spatiotemporal pixel efficiently. This means that the PCs track the state of a local ecosystem over time (fig. 4 left column) or, in case of the mean seasonal cycle, time of the year (fig. 4 right columns). For a representation of the state of the first three components in time and space, see appendix fig. B1.

A first inspection reveals a substantial overlap of seasonal cycles of very different regions of the world. We also see that very different ecosystems may reach very similar states in the course of the season, even though their seasonal dynamics are very different. For instance, a mid-latitude pixel (blue trajectory in fig. 4) shows very similar characteristics to tropical forests during peak growing season. This indicates that an ecosystem of the mid-latitudes can reach similar levels of productivity and water availability as a tropical rain forest (see also SI fig. C1). Likewise, on the first two components, many high latitude areas show similar characteristics to midlatitude areas during winter on the (low latent and sensible energy release as well as low GPP) and many dry areas such as deserts show similar characteristics to areas with a pronounced dry season, e.g. the Mediterranean.

Depending on their position on Earth, ecosystem states can shift from limitation to growth during the year (fig. 4b, e.g. Forkel et al., 2015). For example, the orange trajectory in fig. 4, an area close to Moscow, shifts from a temperature limited state in winter to a state of very high productivity during summer. Other ecosystems remain in a single limitation state with only slight shifts, such as the red trajectory in fig. 4. In the corner of maximum productivity of the distribution, we find tropical forests characterized by a very low seasonality. We also observe that very different ecosystems can have very similar characteristics

during their peak growing season, e.g. green (located in north east India), blue (north west Germany), and orange (located close to Moscow) trajectories have very similar characteristics during peak growing season compared to the red trajectory.

The third component shows a different picture. Due to a consistent winter snow cover in higher latitudes the albedo is much higher and the amplitude of the mean seasonal cycle is much larger than in other ecosystems. Other areas show comparatively little variance on the third component and their relation to productivity and moisture content is even positively correlated to the third component, which is the opposite of what is expected from an albedo axis.

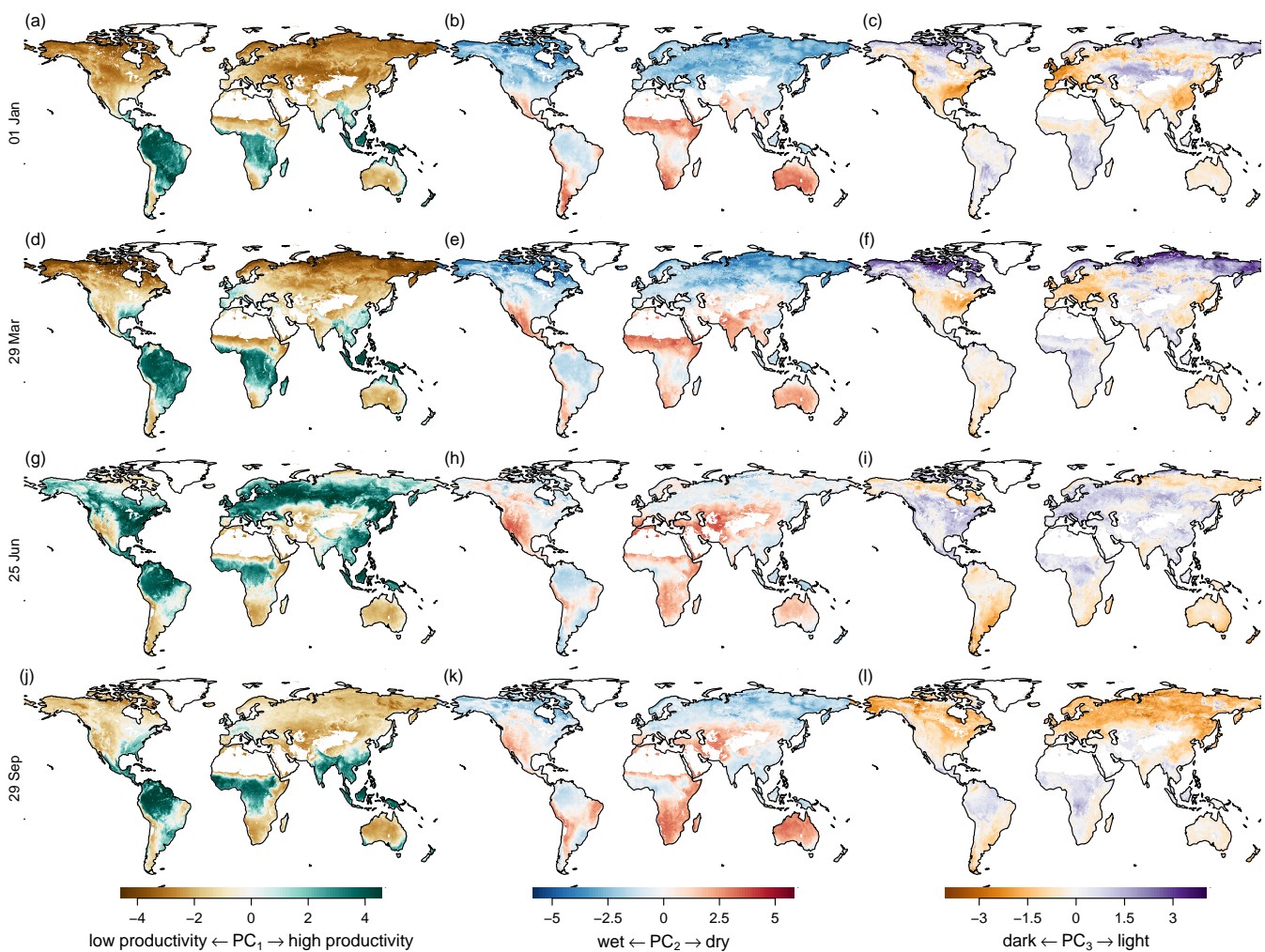

**Figure 6.** Mean seasonal cycle of the first three principal components (in columns) during the seasons (in rows). Left column: first principal component. Middle column: second principal component. Right column: third principal component. Rows from top to bottom: equally spaced intervals during the year. Values have been clamped to 0.7 times their range to increase contrast.

The global pattern of the first principal component follows the productivity cycles during summer and winter (fig. 6, left column) of the northern hemisphere, with positive values (high productivity, green) during summer and negative values (low

productivity, brown) during winter. The tropics show high productivity all year. The global pattern shows the well known green
wave (Schwartz, 1994, 1998) because the first dimension integrates over all variables that correlate with plant productivity.

The second principal component (fig. 6, middle column) tracks water deficiency: red and light red values indicate water
deficiency, light blue values excess water, and dark blue water growth limitation due to cold. Areas which are temperature
limited during winter but have a growing season during summer, such as boreal forests, change from dark blue in winter to
light blue during the growing season. Areas which have low productivity during a dry season change their coloring from red to
light red during the growing season, e.g the north west of Mexico/south west of the United States.

The third principal component (fig. 6, right column) tracks surface reflectance. Therefore we can see the highest values in
the arctic region during winter, other areas vary much less in their reflectance throughout the year. Again, the third component
shows a counterintuitive behavior in midlatitudes, as it is positively correlated with productivity and therefore shows the
opposite behaviour of what would be expected from an indicator tracking albedo.

Although the principal components are globally uncorrelated, they covary locally (see fig. D1). Ecosystems with a dry season
have a negative covariance between $PC_1$ and $PC_2$ while ecosystems that cease productivity in winter have a positive covariance.
Cold arid steppes and boreal climates show a negative covariance between the $PC_1$ and $PC_3$, while other ecosystems that have
a strong seasonal cycle show a positive correlation, many tropical ecosystems don't show a large covariance. A very similar
picture paints the covariance between $PC_2$ and $PC_3$, boreal and and steppe ecosystems show a negative covariance, while most
other ecosystems show a more or less pronounced positive covariance, again depending on the strength of the seasonality.

Observing the mean seasonal cycle of the principal components gives us a tool to characterize ecosystems and may also
serve as a basis for further analysis, such as a global comparison of ecosystems (Metzger et al., 2013; Mahecha et al., 2017).

### 3.5 Hysteresis

The alternative return path between ecosystem states forming the hysteresis loops arise from the ecosystem tracking seasonal
changes in the environmental condition, e.g. summer–winter or dry–rainy seasons (fig. 4b). Hysteresis is a common occurrence
in ecological systems (Folke et al., 2004; Blonder et al., 2017; Renner et al., 2019). For instance, a hysteresis loop can be
found when plotting soil respiration against soil temperature (Tang et al., 2005). The sensitivity of soil respiration to soil
temperature changes seasonally due to changing soil moisture and photosynthesis (by supplying carbon to the rhyzosphere)
producing a seasonally changing hysteresis effect (Gaumont-Guay et al., 2006; Richardson et al., 2006; Zhang et al., 2018).
Biological variables also show a hysteresis effect in their relations with atmospheric variables, e.g. Mahecha et al. (2007b)
found a hysteresis effect between seasonal NEE, temperature, and a number of other ecosystem and climate related variables.
Here we look at the mean seasonal cycles of pairs of indicators and the area they enclose.

The orange trajectory (area close to Moscow) in fig. 4b shows that the paths between maximum and minimum productivity
can be very different, in contrast to the blue trajectory located in the north west of Germany which also has a very pronounced
yearly cycle but shows no such effect. Fig. 4 also indicates that the area inside the means seasonal cycles of $PC_1$–$PC_2$ and
$PC_1$–$PC_3$ show important characteristics while hysteresis in $PC_2$–$PC_3$ is a much less pronounced feature, i.e. we can only see
a pronounced area inside the yellow curve in fig. 4f.

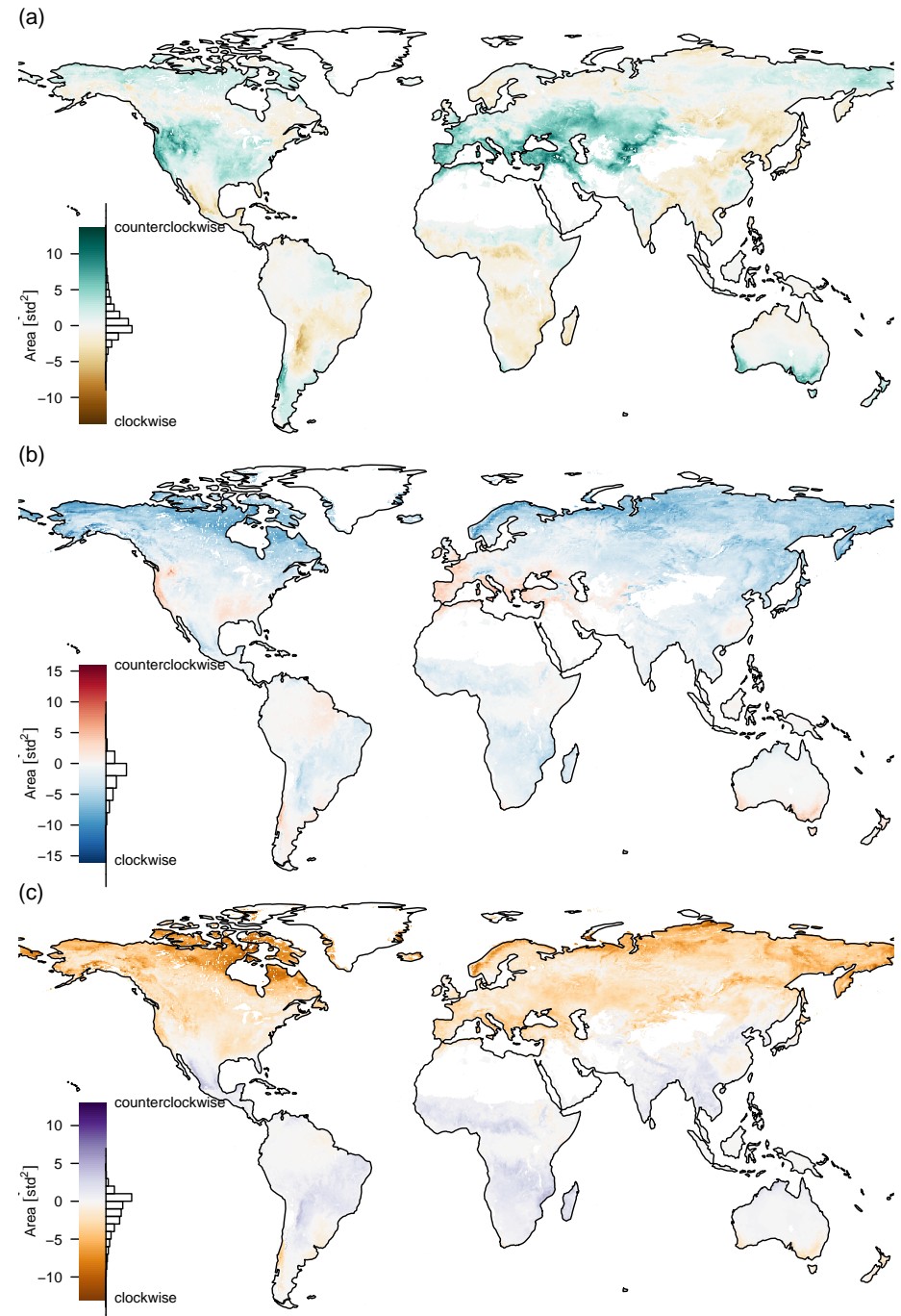

**Figure 7.** The area inside the mean seasonal cycles of (a) $PC_1$–$PC_2$, (b) $PC_1$–$PC_3$, and (c) $PC_2$–$PC_3$. The area is positive if direction is counterclockwise and negative if the direction is clockwise. Most of the trajectories need a strong seasonal cycle to show a pronounced hysteresis effect. If the mean seasonal cycle intersects, the areas cancel each other out, e.g. the green trajectory of 4b.

The trajectories that show a more pronounced anticlockwise hysteresis effect in $PC_1$–$PC_2$ (fig. 7a) are areas with a warm and temperate climate and partially those that have a snow climate with warm summers, i.e. areas that have pronounced growing, dry, and wet seasons and therefore shift their limitations more strongly during the year, i.e. the moisture reserves deplete during growing season and therefore the return path has higher values on the second principal component (the climatic zones are taken from the Köppen–Geiger classification; Kottek et al., 2006). We can also see that areas with dry winters tend to have a clockwise hysteresis effect, e.g. many areas in East Asia, due to the humid summers there is no increasing water limitation during the summer months which causes a decrease on $PC_2$ instead of an increase. Other areas with clockwise hysteresis can be found in winter dry areas in the Andes and the winter dry areas north and south of the African rainforests. Tropical rainforests do not show any hysteresis effect due to their low seasonality. In general we can say that the area inside the mean seasonal cycle trajectory of $PC_1$–$PC_2$ depends mostly on water availability in the growing and non-growing season, i.e. the contrast of wet summer and dry winter vs. dry summer and wet winter.

The hysteresis effect on $PC_1$–$PC_3$ (fig. 7b) shows a pronounced counterclockwise MSC trajectory mostly in warm temperate climates with dry summers, while it shows a clockwise MSC trajectory in most other areas, again tropical rainforests are an exception due to their low seasonality. The most pronounced clockwise MSC trajectories can be found in tundra climates in arctic latitudes, where we have a consistent winter snow cover and a very short growing period. A counterclockwise rotation can be found in summer dry areas, such as the Mediterranean and California, but also some more more humid areas, such as the south east United States and the south east coast of Australia. In these areas we can find a decrease on $PC_3$ in during the non-growing phase which probably corresponds to a drying out of the vegetation and soils.

The hysteresis effect on $PC_2$–$PC_3$ (fig. 7c) mostly depends on latitude, there is a large counterclockwise effect in the very northern parts, due to the large amplitude of $PC_3$, the amplitude gets smaller further south until the rotation reverses in winter dry areas at the the northern and southern extremes of the tropics and disappears on the equatorial humid rain forests.

We can see that the hysteresis of pairs of indicators represents large scale properties of climatic zones. Not only the area enclosed gives interesting information, but also the direction of the rotation. Hysteresis can give information on the seasonal availability of water, seasonal dry periods or snowfall. With the method presented here, we can not observe intersecting trajectories, which would probably give even more interesting insights (e.g. the green trajectory in fig. 4b).

### 3.6   Anomalies of the Trajectories

The deviation of the trajectories from their mean seasonal cycle should reveal anomalies and extreme events. These anomalies have a directional component which makes them interpretable the same way as the original PCs, therefore one can infer the state of the ecosystem during an anomaly. For instance the well-known Russian heatwave in summer 2010 (Flach et al., 2018) appears in fig. 8 as a dark brown spot in the southern part of the affected area, indicating lower productivity, and as a thin green line in the northern parts, indicating an increased productivity. This confirms earlier reports in which only the southern agricultural ecosystems were negatively affected by the heatwave, while the northern predominantly forest ecosystems rather benefited from the heatwave in terms of primary productivity (Flach et al., 2018).

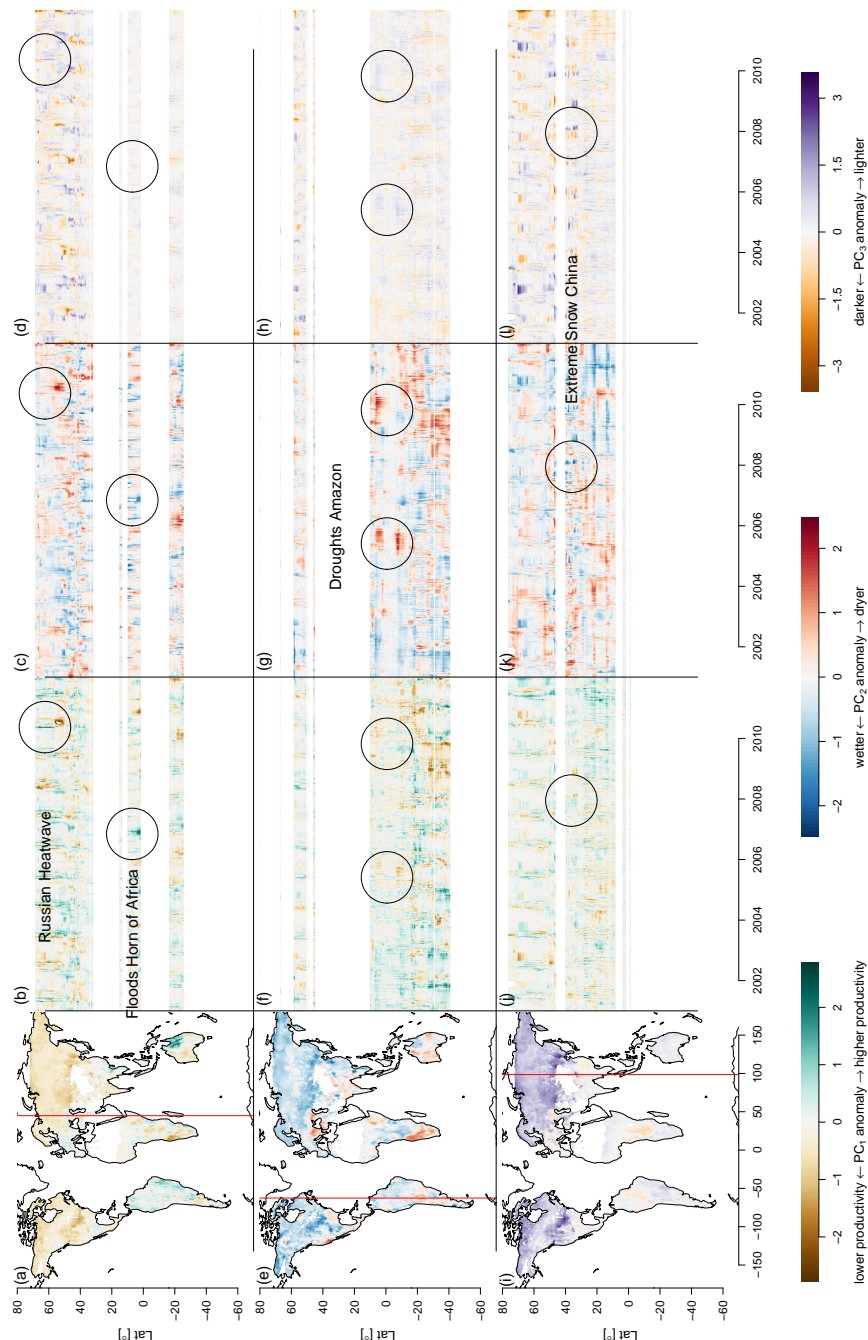

**Figure 8.** Anomalies of the first three principal components; Brown–green contrast shows the anomalies on PC$_1$, a relative low productivity or greening respectively. Blue–red contrast shows the anomalies on PC$_2$, a relative wetness or dryness respectively. Brown–purple contrast shows the anomaly on PC$_3$, a relative deviation in albedo. (a), (e), and (i) are map showing the anomalies of PC$_1$–PC$_3$ on the 1/1/2001 respectively. (b), (c), and (d) show longitudinal cuts of PC$_1$–PC$_3$ at the red vertical line in sub-figure (a) respectively. The effects of of the floods on the Horn of Africa (2006) and the Russian heatwave (2010) are highlighted by circles. (f), (g), and (h) show longitudinal cuts of PC$_1$–PC$_3$ at the red vertical line in sub-figure (e) respectively. Strong droughts in the Amazon during 2005 and 2010 can be observed as large red spots on the fringes of the Amazon basin (highlighted by circles). (j), (k), and (l) show longitudinal cuts of PC$_1$–PC$_3$ at the red vertical line in sub-figure (i) respectively. A strong snowfall event affecting Central and Southern China is marked in circles.

Another example of an extreme event that we find in the PCs is the very wet November rainy season of 2006 in the Horn of Africa after a very dry rainy season in the previous year. This event was reported to bring heavy rainfall and flooding events which caused an emergency for the local population but also an increased ecosystem productivity (Nicholson, 2014). The rainfall event appears as green and blue spots in fig. 8b and c, preceded by the drought events which appear as red and brown spots.

Figures 8f and g also show the strong drought events in the Amazon, particularly the droughts of 2005 and 2010 (Doughty et al., 2015; Feldpausch et al., 2016) appear strongly north and south of the Amazon basin. The central Amazon basin does not show these strong events, because the observable response of the ecosystem was buffered due to the large water storage capacity in the central Amazon basin.

Another extreme event that can be seen is the extreme snow and cold event affecting Central and South China in January 2008, causing the temporary displacement of 1.7 million people and economic losses of approximately US $ 21 billion (Hao et al., 2011). This event shows up clearly on $PC_2$ and $PC_3$ as cold and light anomalies respectively (see fig. 8k and f).

## 3.7  Single Trajectories

Observing single trajectories can give insight into past events that happened at a certain place, such as extreme events or permanent changes in ecosystems. The creation of trajectories is an old method used by ecologists, mostly on species assembly data of local communities, to observe how the composition changes over time (e.g. Legendre et al., 1984; Ardisson et al., 1990). In this context, we observe how the states of the ecosystems inside the grid-cell shift over time, which comprises a much larger area than a local community but is probably also less sensitive to very localized impacts than a community level analysis. One of the main differences of the method applied here to the classical ecological indicators is that the trajectories observed here are embedded into the space spanned by a single global PCA and therefore we can compare a much broader range of ecosystems directly.

The seasonal amplitude of the trajectory in the Brazilian Amazon increases due to deforestation and crop growth cycles. Figure 9a shows an area in the Brazilian Amazon in Rondônia (9.5°S, 63.5°W) which was affected by large scale land use change and deforestation. It can be seen that the seasonal amplitude increases strongly after the beginning of 2003. Reasons for this increased amplitude could lie in any of the following reasons or a combination of them: Deforestation decreases water storage capability and dries out soils causing larger variability in ecosystem productivity. Therefore, during periods of no rain, large scale deforestation can cause a shift in local scale circulation patterns causing lower local precipitation (Khanna et al., 2017). Crop growth and harvest causes an increased amplitude in the cycle of productivity. An analysis of the trajectory can point to the nature of the change, however finding the exact causes for the change requires a deeper analysis.

The 2010 Russian heatwave has a very clear signal in the trajectories, fig. 9b shows the deviation of the trajectory during the Russian heatwave (red line) in an area east of Moscow (56°N 45.5°E). In the southern grass- and croplands, the heatwave caused the productivity to drop significantly during summer due to a depletion of soil moisture. In the northern forested parts affected, the heatwave caused an increase in ecosystem productivity during spring due to higher temperatures combined with sufficient water availability. This shows the compound nature of this extreme event (see fig. 8a and Flach et al. 2018). The

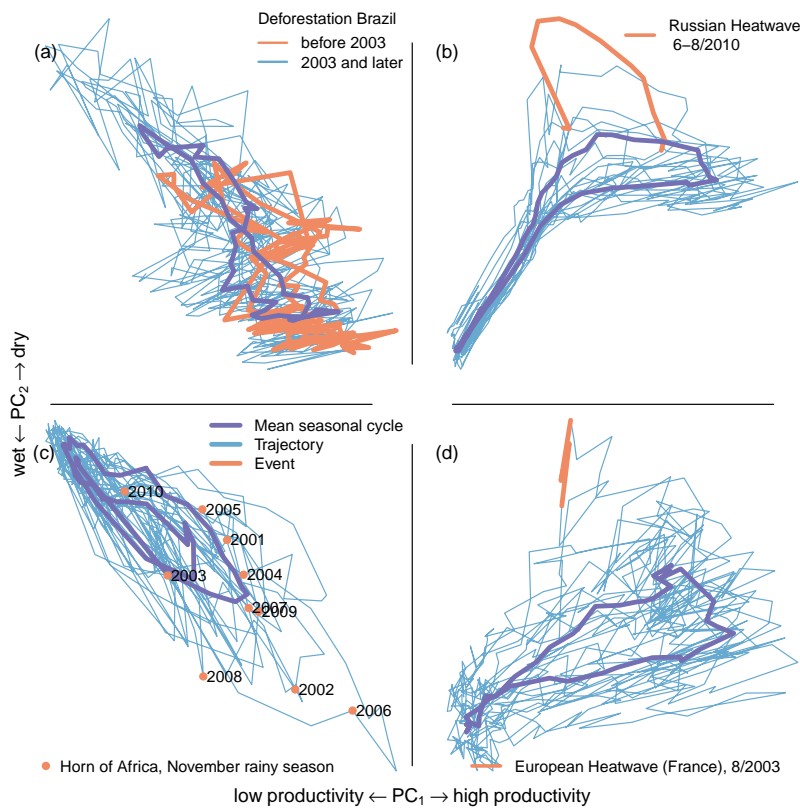

**Figure 9.** Trajectories of the first two Principal Components for single pixels. (a) Deforestation increases the seasonal amplitude of the first two PCs (Brazilian rainforest, 9.5°S 63.5°W). The red line shows the trajectory before 2003, the blue line the trajectory 2003 and later, a strong increase in seasonal amplitude can be observed after 2003. (b) The heatwave is clearly visible in the trajectory (red, Russian heatwave, summer 2010, 56°N 45.5°E). (c) Rainfall in the short raining season (November/December) influences agricultural yield and can cause flooding (extreme flooding after drought, 11/2006, 3°N 45.5°E). (d) European heatwave in Summer 2003 was one of the strongest on record (France, 47.2°N 3.8°E). The mean seasonal cycle of the trajecotries is shown in purple.

analysis of the trajectory points directly towards the different types of extremes and responses that happened in the biosphere during the heatwave.

Variability of rainfall during the November rainy season in the Horn of Africa (3°N 45.5°E, fig. 9c) shows the trajectory and points in November of the observed time. The November rain has implications for food security because the second crop season depends on it. In 2006, the rainfall events were unusually strong and caused widespread flooding and disaster but also higher ecosystem productivity (cf. also fig. 8). This was especially devastating because it followed a long drought that caused crop failures. Note also the two rainy seasons in the mean seasonal cycle (purple line if fig. 9c).

The 2003 European heatwave is reflected in the trajectories just a the 2010 Russian heatwave. Figure 9d shows the trajectory during the August 2003 heat wave in Europe (France, 47.2°N 3.8°E). The heatwave was unprecedented and caused large scale

400  environmental, health, and economics losses (Ciais et al., 2005; García-Herrera et al., 2010; Miralles et al., 2014). The 2010 heatwave was stronger than the 2003 heatwave but the strongest parts of the 2010 heatwave were in eastern Europe (cf., fig. 8), while the center of the 2003 heatwave was located in France.

As we have seen here, observing single trajectories in reduced space can give us important insights into ecosystem states and changes that occur. While the trajectories can point us towards abnormal events, they can only be the starting points for
405  deeper analysis to understand the details of such state changes.

## 3.8  Trends in Trajectories

The accumulation of $CO_2$ in the atmosphere should cause an increase in global productivity of plants due to $CO_2$ fertilization, while larger and more frequent droughts and other extremes may counteract this trend. Satellite observations and models have shown that during the last decades the world's ecosystems have greened up during growing seasons. This is explained by $CO_2$
410  fertilization, nitrogen deposition, climate change and land cover change (Zhu et al., 2016; Huang et al., 2018; Anav et al., 2015). Tropical forests showed especially strong greening trends during growing season.

General patterns of trends that can be observed are a positive trend (higher productivity) on the first principal component in many arctic regions, many of these regions also show a wetness trend, with the notable exception of the western parts of Alaska which have become dryer. This is important, because wildfires play a major role in these ecosystems (Jolly et al., 2015;
415  Foster et al., 2019). these changes are also accompanied by a decrease on $PC_3$ due to a loss in snow cover. A large scale dryness trend can also be observed across large parts of western Russia. Increasing productivity can also be observed on large parts of the the Indian subcontinent and eastern Australia. Negative trends in the first component can also be observed: they are generally smaller and appear in regions around the Amazon and the Congo basin, but also in parts of western Australia. The main difference from previous analyses on the observations presented here is that e.g. Zhu et al. (2016) looked only at trends
420  during the growing season while this analysis uses the entire time series to calculate the slope.

In the Amazon basin, we find a dryness trend accompanied by a decrease in productivity and a slight increase in $PC_3$; In the Congo basin, we find a wetness trend and an increasing productivity in the northern parts, while the southern part and woodland south of the Congo basin show a strong dryness trend with decreased productivity. This is different to the findings of Zhou et al. (2014), who found a widespread browning of vegetation in the entire Congo basin for the April-May-June seasons
425  during the period 2000–2012. The finding of Zhou et al. (2014) is not reflected in our data, especially compared to the areas surrounding the Congo basin, we can find only minor browning effects inside the basin and our findings are more in line with the global greening (Zhu et al., 2016), which show a browning mostly outside the Congo basin.

In eastern Australia we find a strong wetness and greenness trend which is due to Australia having a "milennium drought" since the mid nineties with a peak in 2002 (Nicholls, 2004; Horridge et al., 2005) and extreme floods in 2010–2011 (Hendon
430  et al., 2014).

Large parts of the Indian subcontinent shows a trend towards higher productivity and an overall wetter climate. The greening trend in India happens mostly over irrigated cropland. However browning trends over natural vegetation have been observed but do not emerge in our analysis (Sarmah et al., 2018). A very notable greening and wetness trend can be observed in Myanmar

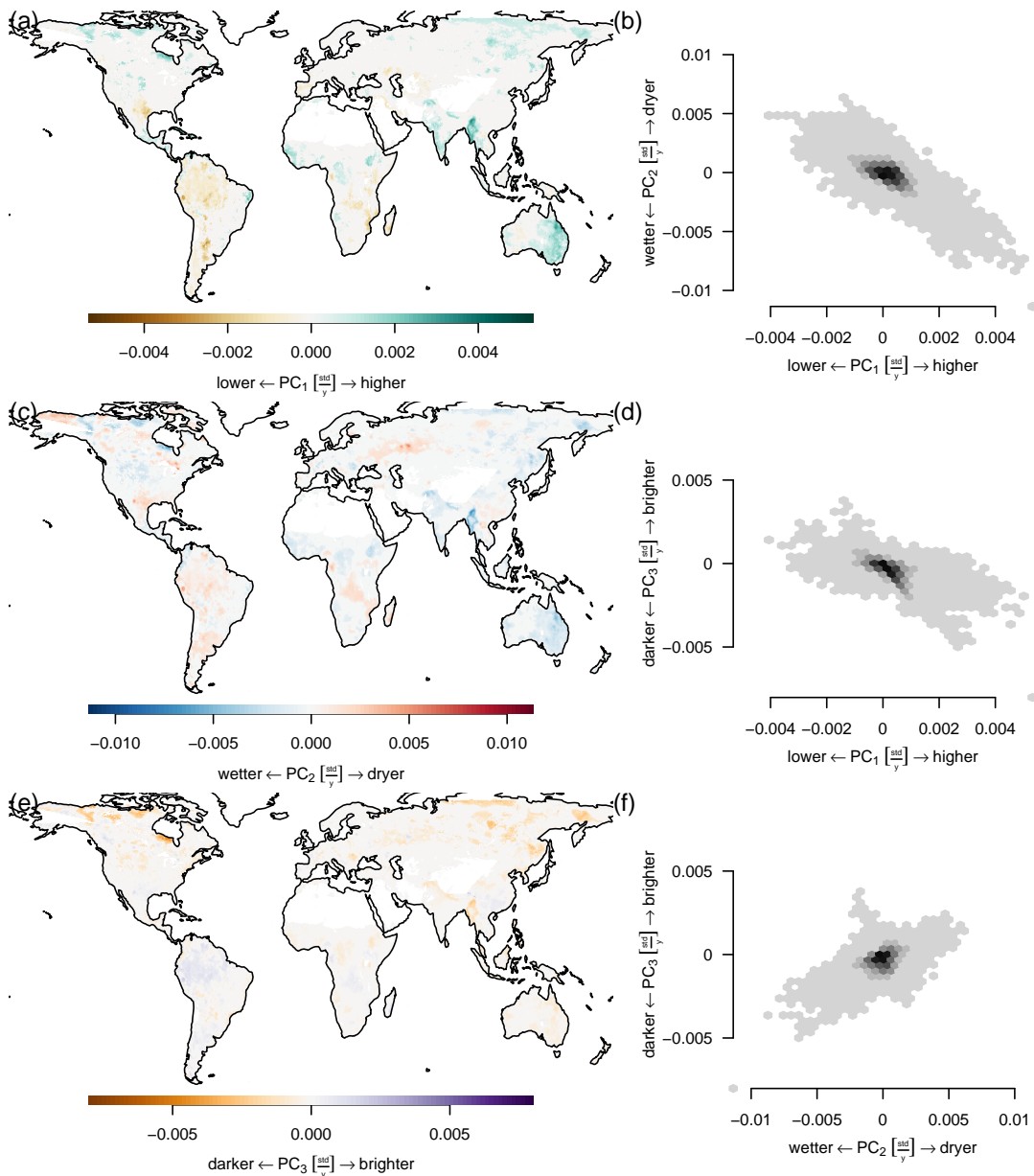

**Figure 10.** (a), (c), (e) Trends in $PC_1$–$PC_3$ respectively (2001–2011). (b), (d), (f) Bivariate distribution of trends. Trends were calculated using the Theil–Sen estimator, (a), (c), and (e) show significant trends only ($p < 0.05$, Benjamini–Hochberg adjusted).

due to an increase in intense rainfall events and storms, although the central part experienced some strong droughts at the same time (Rao et al., 2013). In Myanmar we also find one of the strongest trends in $PC_3$ outside of the Artic.

In large parts of the Arctic, a trend towards higher productivity can be observed, vegetation models attribute this general increase in productivity to $CO_2$ fertilization and climate change. The changes also cause changes to the characteristics of the seasonal cycles (Forkel et al., 2016). Stine et al. (2009) found a decreased seasonal amplitude of surface temperature over norther latitudes due to winter warming.

The seasonal amplitude of atmospheric $CO_2$ concentrations has been increasing due to climate change causing longer growing seasons and changing vegetation cover in northern ecosystems (Forkel et al., 2016; Graven et al., 2013; Keeling et al., 1996). Therefore we checked for trends in the seasonal amplitude, but because each time series only consists of 11 values (one amplitude per year), after adjusting the $p$-values for false discovery rate, we could not find a significant slope. However, there were many significant slopes with the unadjusted $p$-values, see the appendix, fig. E1.

Another way to detect changes to the biosphere consists in the detection of breakpoints, which has been applied successfully to detect changes in global NDVI time series (de Jong et al., 2011; Forkel et al., 2013), or generally to detect changes in time series (Verbesselt et al., 2010). A proof of concept analysis can be found in fig. F1, we hope that applying this method to indicators instead of variables can detect a wider range of breakpoints analyzing a single time series.

### 3.9 Relations to other PCA-type analyses

One of the most popular applications of PCA in meteorology are EOFs, which applies PCA typically on a single variables, i.e. on a data set with the dimensions lat × lon × time, although EOFs can be calculated from multiple variables. EOFs can be calculated in $S$-mode and $R$-mode. If we matricize our data cube so that we have time in rows and lat × lon × variables in columns, then $S$-mode PCA works on the correlation matrix of the combined variable and space dimension. In $T$-mode, the PCA works on the correlation matrix formed by the time dimension (Wilks, 2011). The PCA presented here works slightly different: (1) We did a different matricization (lat × lon × time in rows and variables in columns) and then (2) the PCA works on the correlation matrix formed by the variables, therefore in this framework we could call this a $V$-mode PCA.

Ecological analyses use PCA usually with matrices of the shape object × descriptors, when calculating the PCA on the correlation matrix formed by the objects, then it is called a $Q$-mode analysis, when the PCA is applied on the correlation matrix formed by the variables, then it is called an $R$-mode analysis (Legendre and Legendre, 1998). The PCA done in this study is closest to an $R$-mode analysis, in the present case the descriptors are the various data streams and the objects are the spatiotemporal pixels.

Using PCA as a method for dimensionality reduction means that we are assuming linear relations among features. A non-linear method could possibly be more efficient in reducing the number of variables, but would also have significant disadvantages. In particular: nonlinear methods typically require tuning specific parameters, objective criteria are often lacking, a proper weighting of observations is difficult, the methods are often not reversible, and it is harder to interpret the resulting indicators due to their nonlinear nature (Kraemer et al., 2018). The salient feature of PCA is that an inverse projection is well defined and allows for a deeper inspection of the errors, which is not the case for nonlinear methods which learn a highly flexible

transformation that is hard to invert. Therefore interpretability of the transform in meaningful physical units in the input space is often not possible. In the machine learning community, this problem is known as the "pre-imaging problem" (Mika et al., 1999; Arenas-Garcia et al., 2013) and is a matter of current research.

## 4 Conclusions

To monitor the complexity of the changes occurring in times of an increasing human impact on the environment, we used PCA to construct indicators from a large number of data streams that track ecosystem state in space and time on a global scale. We showed that a large part of the variability of the terrestrial biosphere can be summarized using three indicators. The first emerging indicator represents carbon exchange, the second indicator shows the availability of water in the ecosystem, while the third indicator represents mostly a binary variable that indicates the presence of snow cover. The distribution in the space of the first two principal components reflects the general limitations of ecosystem productivity. Ecosystem production can either be limited by water or energy.

The first three indicators can detect many well-known phenomena without analyzing variables separately due to their compound nature. We showed that the indicators are capable of detecting seasonal hysteresis effects in ecosystems, as well as breakpoints, e.g. large scale deforestation. The indicators can also track other changes to the seasonal cycle such as patterns of changes to the seasonal amplitudes and trends in ecosystems. Deviations from the mean seasonal cycle of the trajectories indicate extreme events such as the large scale droughts in the Amazon during 2005 and 2010 and the Russian heat wave of 2010. The events are detected in a similar fashion as with classical multivariate anomaly detection methods while directly providing information on the underlying variables.

Using multivariate indicators we gain a high level overview of phenomena in ecosystems and the method therefore provides an interesting tool for analyses where it is required to capture a wide range of phenomena which are not necessarily known a priori. Future research should consider nonlinearities, adding data streams describing other important biosphere variables (e.g. related to biodiversity and habitat quality), and including different subsystems, such as the atmosphere or the anthroposphere.

*Code and data availability.* The data are available and can be processed at https://www.earthsystemdatalab.net/index.php/interact/data-lab/, last accessed 30 March 2020. The exact dataset and a docker container to reproduce the analysis can be found under https://doi.org/10.5281/zenodo.3733766. The code to reproduce this analysis is available under http://doi.org/10.5281/zenodo.3733783 and https://github.com/gdkrmr/summarizing_the_state_of_the_biosphere, last accessed 30 March 2020.

## Appendix A: Description of variables

Variables used describing the biosphere can be found in tab. 1, here we provide a more complete description of all variables:

**Black Sky Albedo** is the reflected fraction of total incoming radiation under direct hermispherical reflectance, i.e. direct illumination (Muller et al., 2011). This dataset is the broadband surface albedo including the visible, the near, and the shortwave-infrared spectrum (400–3000nm). It is derived from the SPOT4-VEGETATION, SPOT5-VEGETATION2, and the MERIS satellite sensors.

**White Sky Albedo** is the reflected fraction of total incoming radiation under bihemispherical reflectance, i.e. diffuse illumination (Muller et al., 2011). Together with black sky albedo it can be used to estimate the albedo under different illumination conditions. This dataset is the broadband surface albedo including the visible, the near, and the shortwave-infrared spectrum (400–3000nm). This dataset is derived from the SPOT4-VEGETATION, SPOT5-VEGETATION2, and the MERIS satellite sensors.

**Evaporation** $[\mathrm{mm/day}]$ is the amount of water evaporated per day, depends on the amount of available water and energy. This dataset is based on the GLEAMv3 model (Martens et al., 2017), using satellite data from ESA CCI and SMOS to derive a number of variables.

**Evaporative Stress** modeled water stress for plants, zero means that the vegetation has no water available for transpiration and one means that transpiration equals potential transpiration. This dataset is based on the GLEAMv3 model (Martens et al., 2017), using satellite data from ESA CCI and SMOS to derive a number of variables.

**fAPAR** the fraction of absorbed photosynthetically active radiation, a proxy for plant productivity (Disney et al., 2016). This dataset is based on the GlobAlbedo dataset (http://globalbedo.org) and the MODIS fAPAR and LAI products.

**Gross Primary Productivity (GPP)** $[\mathrm{gCm}^{-2}\mathrm{day}^{-1}]$ the total amount of carbon fixed by photosynthesis (Tramontana et al., 2016). This dataset is derived from upscaling eddy covariance tower observations to a global scale using machine learning methods.

**Terrestrial Ecosystem Respiration (TER)** $[\mathrm{gCm}^{-2}\mathrm{day}^{-1}]$ the total amount of carbon respired by the ecosystem, includes autotrophic and heterotropic respiration (Tramontana et al., 2016). This dataset is derived from upscaling eddy covariance tower observations to a global scale using machine learning methods.

**Net Ecosystem Exchange (NEE)** $[\mathrm{gCm}^{-2}\mathrm{day}^{-1}]$ The total exchange of carbon of the ecosystem with the atmosphere $\mathrm{NEE} = \mathrm{GPP} - \mathrm{TER}$ (Tramontana et al., 2016). This dataset is derived from upscaling eddy covariance tower observations to a global scale using machine learning methods.

**Latent energy (LE)** $[\mathrm{Wm}^{-2}]$ the amount of energy lost by the surface due to evaporation (Tramontana et al., 2016). This dataset is derived from upscaling eddy covariance tower observations to a global scale using machine learning methods.

**Sensible Heat (H)** $[\mathrm{Wm}^{-2}]$ the amount of energy lost by the surface due to radiation (Tramontana et al., 2016). This dataset is derived from upscaling eddy covariance tower observations to a global scale using machine learning methods.

**Root-Zone Soil Moisture** $[\mathrm{m}^3\mathrm{m}^{-3}]$ the moisture content of the root zone. This dataset is based on the GLEAMv3 model (Martens et al., 2017), using satellite data from ESA CCI and SMOS to derive a number of variables.

**Surface Soil Moisture** $[\text{mm}^3\text{mm}^{-3}]$ the soil moisture content at the soil surface. This dataset is based on the GLEAMv3 model (Martens et al., 2017), using satellite data from ESA CCI and SMOS to derive a number of variables.

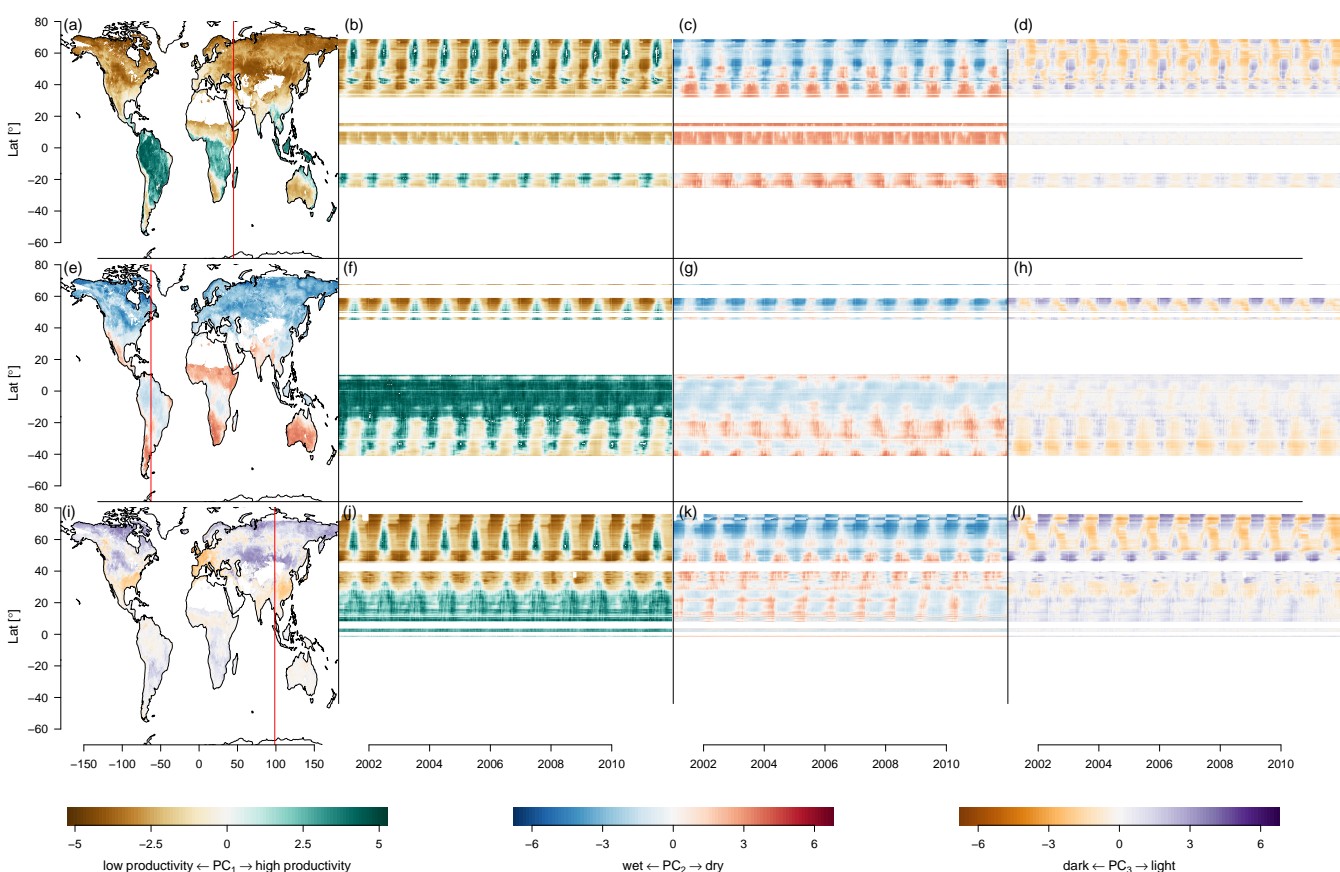

**Figure B1.** Time and space patterns of PC$_1$–PC$_3$, the cutpoints are the same as in fig. 8. Brown–green contrast shows the state of PC$_1$, from low to high productivity. Blue–red contrast shows the state of PC$_2$, from cold to dry. Brown–purple contrast shows the state of PC$_3$, from dark to light. (a), (e), and (i) are map showing the state of PC$_1$–PC$_3$ on the 1/1/2001 respectively. (b), (c), and (d) show longitudinal cuts of PC$_1$–PC$_3$ at the red vertical line in sub-figure (a) respectively. (f), (g), and (h) show longitudinal cuts of PC$_1$–PC$_3$ at the red vertical line in sub-figure (e) respectively. (j), (k), and (l) show longitudinal cuts of PC$_1$–PC$_3$ at the red vertical line in sub-figure (i) respectively.

## Appendix C: Mean Seasonal Cycle Extrema

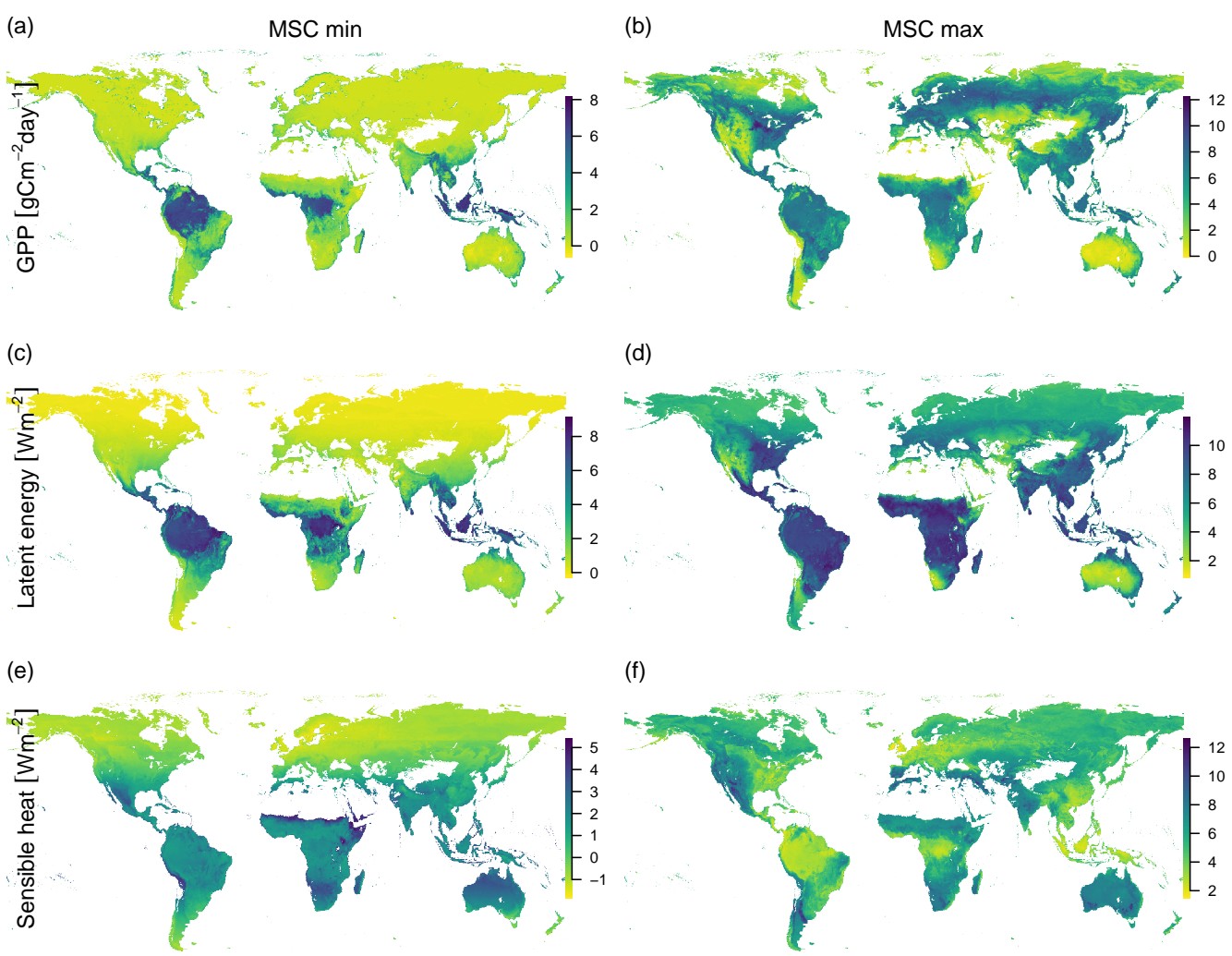

**Figure C1.** Shows the minimum (left column) and maximum (right column) mean seasonal cycles of GPP (upper row), Latent Heat (middle row), and Sensible heat (lower row). This illustrates the similarity of possibly very different ecosystems in terms of productivity and limitations. During peak growing season, many mid latitude areas have a similar productivity and latent energy release as tropical rainforests (subfigure b and d). The highest maximum seasonal sensible heat loss can be found in dry areas around the world and is lowest in areas with a wet climate such as tropical rainforests and maritime climates (subfigure f).

## Appendix D: Spatial covariances of the components

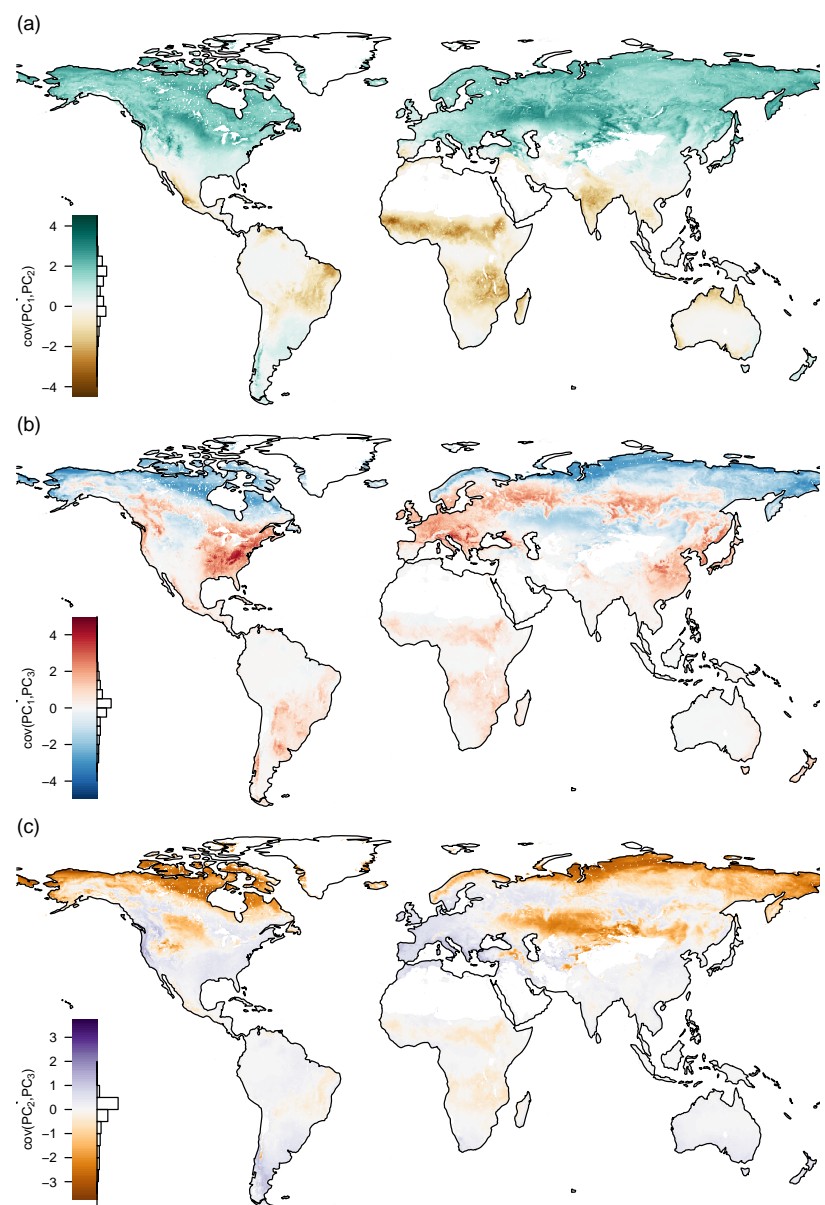

**Figure D1.** Pairwise covariances of the first three principal components mean seasonal cycles by space. (a) $\mathrm{cov}(PC_1, PC_2)$, (b) $\mathrm{cov}(PC_1, PC_3)$, and (c) $\mathrm{cov}(PC_2, PC_3)$. The bar charts show the distribution of the covariances. It can be seen that although two principal components are globally uncorrelated by their way of construction, they covary locally.

## Appendix E: Changes in the Seasonal Amplitude

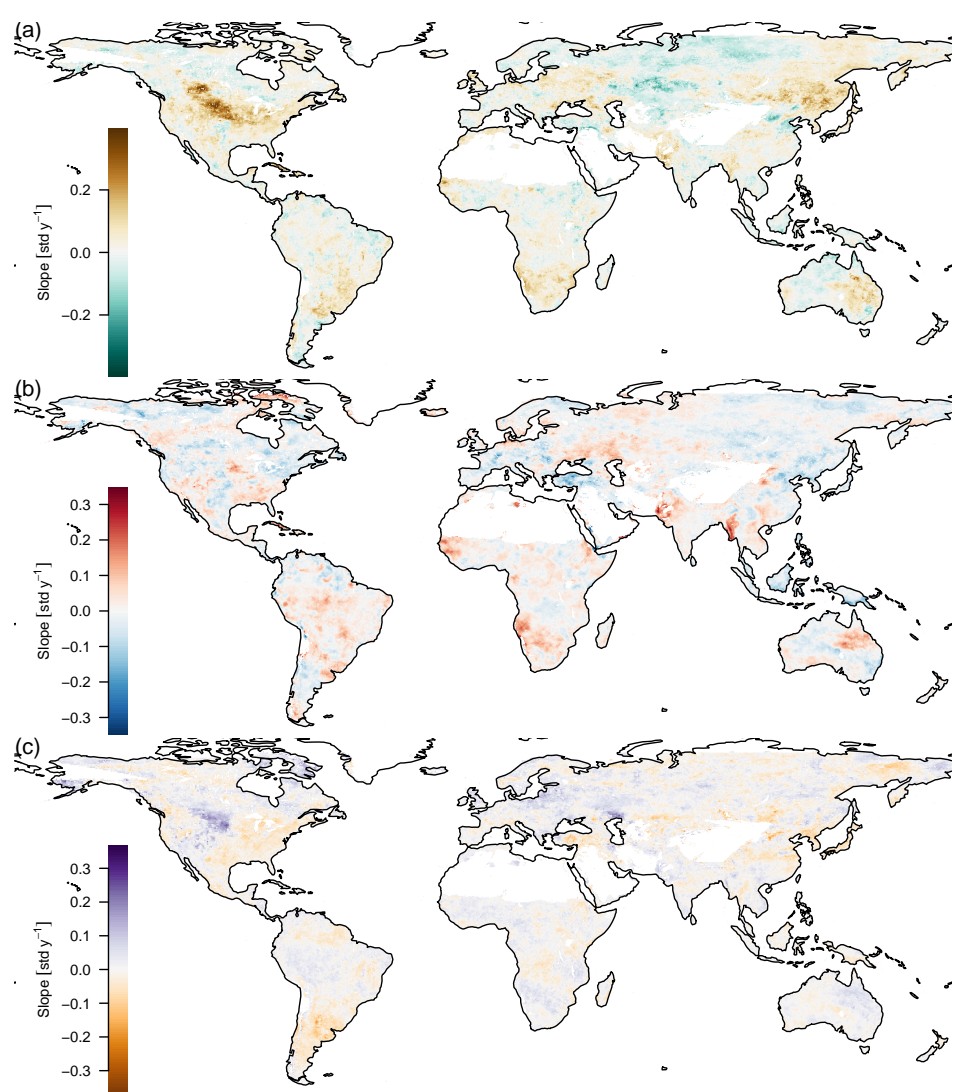

**Figure E1.** Trends in the amplitude of the yearly cycle, 2001–2011, Theil–Sen estimators only significant slopes ($p < 0.05$, *unadjusted*) are shown. Because there is only a single amplitude per year and therefore only 11 data points per time series, the Benjamini–Hochberg adjusted $p$-values are not significant.

## Appendix F: Breakpoints in Trajectories

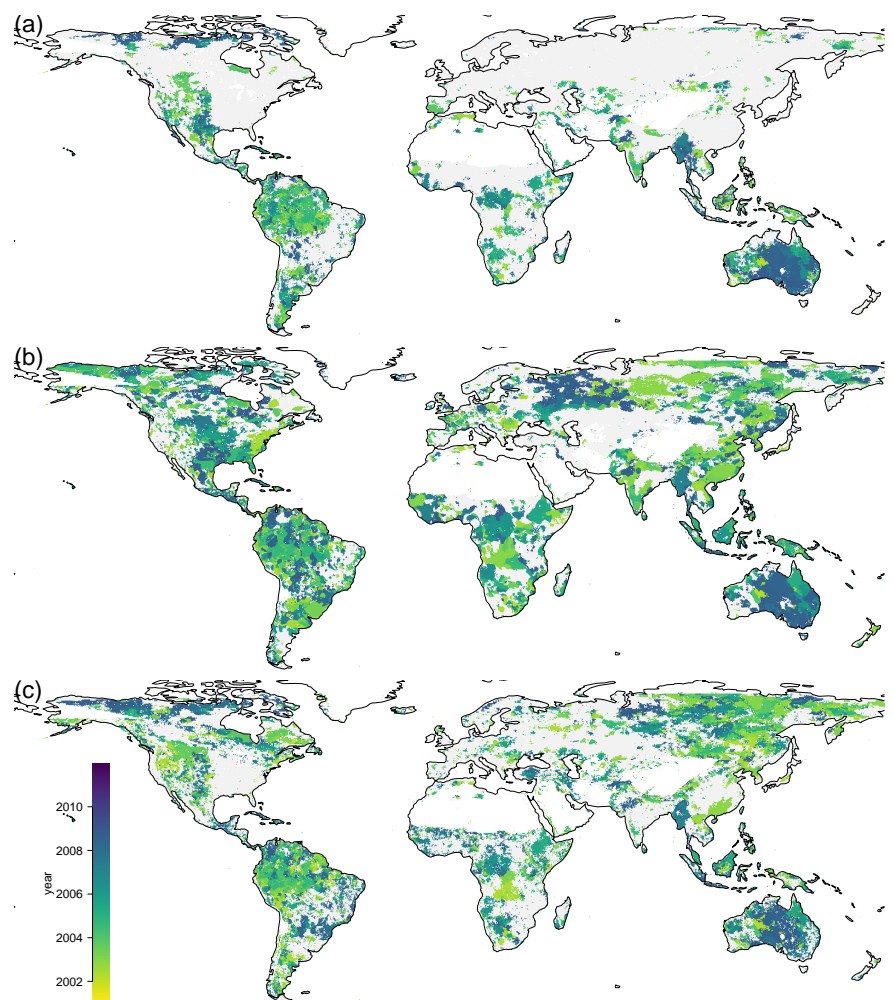

**Figure F1.** Breakpoint detection, (a) on $PC_1$, (b) on $PC_2$, and (c) on $PC_3$, the color indicates the year of the biggest breakpoint if a significant breakpoint was found, grey if there was no significant breakpoint found.

535 As the environmental conditions change, due to climate change and human intervention, the local ecosystems may change gradually or abruptly. Detecting these changes is very important for monitoring the impact of climate change and land use change onto the ecosystems. We applied breakpoint detection on the trajectories (fig. F1).

 Breakpoints on the first component were found in the entire Amazon and the largest breakpoint is dated in the year 2005 during the large drought event. The entire eastern part of Australia shows its largest breakpoint towards the end of the time

540 series because of a La Niña event, which caused lower temperatures and higher rainfall than usual during the years 2010 and 2011.

*Author contributions.* GK and MDM designed the study in collaboration with MR and GCV. GK conducted the analysis and wrote the manuscript with contributions from all co-authors.

*Competing interests.* The authors declare that they have no conflict of interest.

*Acknowledgements.* This study is funded by the Earth system data lab—a project by the European Space Agency. MDM and MR have been supported by the H2020 EU project BACI under grant agreement No 640176. GCV work has been supported by the EU under the ERC consolidator grant SEDAL-647423. We thank Fabian Gans and German Poveda for useful discussions. We thank Jake Nelson for proofreading the manuscript. We thank four anonymous reviewers for very helpful suggestions and the editorial advice by Kirsten Thonicke that improved the manuscript greatly

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
