# Peer review of "Summarizing the state of the terrestrial biosphere in few dimensions"

_Biogeosciences, 2019_

## Referee Comment (RC1) · Anonymous Referee #1 · 16 Sep 2019

I appreciated reading the discussion paper Summarizing the state of the terrestrial biosphere in few dimensions by Guido Kraemer and colleagues. The paper presents an approach for summarizing key variables on the terrestrial biosphere into fewer independent components using established multi-variate methods. They exemplify their approach by showing several trajectories across space and time and by highlighting some major anomalies visible in their data.

While the work is well presented and scientifically sound, I have some major concerns regarding the publication of the manuscript in its current form:

1) The authors state that the first two components explain large parts of the variance and that the 'knee' is reached with the second component. However, inspecting Figure 1a, it seems that the 'knee' is reached with the third component, which still explains

9% of the variance. I was a little confused that the third component was disregarded throughout the whole manuscript, without giving a strong justification. Figure 2b indicates that the third component might be strongly connected to albedo. I encourage the authors to either expand their analysis to also include the third component, or to give a very strong argument for its exclusion. As it stands now, the decision to only inspect the first two components is very subjective.

2) I am missing a strong discussion/conclusion on how the manuscript advances scientific progress. Putting it into simple terms, the authors apply PCA – a widely used and established method – to a set of existing data sets. As such, it is not really a novel methodological development, but rather a demonstration of what could be done with global datasets as provided though the Earth System Data Lab. While this is not a deal-breaker per sé, the authors could greatly advance their manuscript by explaining how this approach can be used by other scientists, that is how it will advance the science of the terrestrial biosphere.

3) Many of the results are buried in the Appendix but never picked-up in the main text. In fact, Figure A1, B1, D1 and C1 were never referenced in the main text. The authors thus present many results in the Appendix that are not discussed in the main manuscript and thus the reader is left alone with her own interpretation. As some of the results are quite crucial for evaluating the method (e.g., the errors presented in B1), I strongly encourage the authors to thoroughly discuss them in their manuscript.

4) The writing needs improvement for turning this already good manuscript into an excellent manuscript. For example, the authors often describe their figures, instead of the results (Figure X shows...). It would be much more interesting to read about the main result instead (A influences B (Figure X)). I am sure the senior authors of this manuscript can do a great job in revising the manuscript to make it more accessible and exciting for the reader.

5) There are some wording and spelling/grammar issues, some of which listened below:

L. 16: Suggest removing 'the' before 'global'.

L. 27: Spring is not a phenological event. Could use onset of bud-flush or similar.

L. 74: Not clear how standardization accounts for differences in scales. What scales? Spatial? Temporal?

L. 138: The breakpoint detection comes out of the blue. Why is this done? What was the rational behind? This needs a decent introduction.

L 142: Same as above. The term hysteresis is never introduced before, but then explained in the results section (L. 239). As a reader, I would love to hear the details upfront, instead of reading about them in the results/discussion.

L. 148: Maybe include an example figure here, instead of referencing to the results already.

L. 151: 'We see that...' is not a good opener. Directly describe the result, be precise and upfront (e.g., The first two components explained 73% of the variance (Figure 1a))

L. 160: What is the pre-imaging problem? Please do not assume that the reader reads up the details in the reference provided. Either avoid naming it or give a brief description.

L. 162: Again, not the best opener. The first sentence of a paragraph should summarize the main point of the paragraph (topic sentence), allowing the reader to skim through the manuscript. This sentence just describes where the reader can find a result, but nothing about the result itself.

L. 164: Odd formulation (two times related).

L. 174ff: his paragraph actually described the indicators used and does not discuss the results. This could go into the methods description or should be more clearly related

to the actual results. Figure 2: What are 'some points'? How were they chosen?

L. 139: As said before, this is rather introduction than results/discussion. I would have very much appreciated reading this in the introduction.

L. 258: rephrase: . . . and can therefore be interpreted. . .

L. 282: Again, put the result in the spotlight, not the figure showing the result.

L. 305: Occur instead of occurring.

L. 312: Move 'especially' after 'showed'.

L. 313: Repeats methods.

L. 320: Why did you calculate the trends from the full data? Would it have been better to use the growing season as well to facilitate comparison? Please give a reasoning why you do it differently.

L. 324: Something odd with the sentence starting with 'Inside. . .'.

L. 327: Remove 'a' before 'browning'.

L. 349: The breakpoints are actually never shown, nor discussed. The conclusion is thus not really based on data here.

L. 352: in, not 'ina'.

---

## Referee Comment (RC2) · Anonymous Referee #2 · 19 Sep 2019

This is a very interesting paper addressing some important issues of big data analysis for ecology studies. It is rich in analyses and provides some new views on an old method (PCA). I particularly liked the analysis of trajectories that I found quite powerful, notably for case studies. Yet I found it difficult to understand what key research questions are addressed in this paper. This is important to clarify at the end of the introduction as the authors is providing us with a suit of analyses that may resemble (for non PCA-expert) an attempt of addressing many (all?) questions without real rationale. The readers need to have a clear (concise) view of the objectives of this paper, and they need to be guided through the analyses by referring back to the main research questions.

In addition, I also have a major concern related to the set of inputs data used to feed

the PCA. I agree that PCA is a powerful tool to deal with correlated variables, yet I have difficulties understanding why the authors have decided to include variables that are obviously highly correlated. To my opinion, vegetation productivity proxies are overrepresented as well as those related to water availability and stress. It puts some doubts in my head as to whether the finding of PC1 (primary productivity) and PC2 (surface hydrology) driving the state of the biosphere in space and time is truly original (or just purely mathematical). It is therefore important for the authors to justify the set of original variables. A suggestion could also be to decrease the number of input variables (removing obvious redundant proxies) as the amount of data to be condensed is mainly coming from the 8days interval used for the analysis.

Finally I also have other comments and concerns - notably related to the structure of the manuscript - that would need to be addressed by the authors prior publication of their research (see attached report for details).

— Detailed comments

(1) Abstract

The authors start off the abstract by mentioning the importance of detecting abrupt and gradual changes in terrestrial ecosystem but do not develop further in the introduction. In the method section, the detection of breakpoints reappears but no results are presented or discussed (except for the appendix A). The authors should decide whether to consider the detection of abrupt changes as a real research question for this study.

(2) Introduction

As stated in my main comment, I find that there is somewhat a mismatch between the introduction and the method section. In the introduction, the authors touch upon many issues related to assessing and attributing changes of biosphere properties. However apart from creating a new set of independent, 'essential' variables, they do not clearly mention what other research questions this study is going to address ; whereas in

the methods they mention PCA, trend and breakpoints analyses. Clearly stating the research questions for this study would help the readers to understand the rationale behind each analysis.

(3) Data and methods

The description of the data slightly too minimalistic, including in the appendix F. Mentioning the input data (satellite, climate or others) feeding into each dataset would be helpful. The observation period used for this study is also not mentioned.

L75. This statement is not always valid (e.g. in the case of equal-area projection). The sentence would be clearer if the authors would mention the projection system used here.

L77. The authors mentioned that they used a modified PCA, reading from the description given in the following lines, the PCA applied here seems to be standard. Could the authors provide some explanations to why / how the PCA has been modified? It should also clarify whether they applied the PCA in s or t-mode.

Per-pixel analysis. It would be nice here to make a link to the (extended – see comment above) research questions in order to understand directly the rationale for such analyses.

(4) Results

General comment: I highly suggest to split the results and discussion into two separate sections. It will facilitate the reading and will allow the authors to emphasise better the originality of their work. Example: L155-161, L164-173, L175-182, L235-246, etc. should not be in a results section s.s., but would rather belong to a discussion (or even introduction or method). Please consider at least moving all methods description and introduction to new concepts to the respective adequate sections.

L153 and Figure 1. The authors mentioned that there is a knee at component 2. I believe it is rather at component 3. This component still contribute to the total variability

to a share of almost 10%, therefore the authors should either include it in the rest of the analysis or provide an adequate justification not to.

Also I generally miss a figure presenting together the temporal and the spatial patterns for the main PCs. This could be put as supplementary material.

In the caption of Fig.1 I would recommend to change the term axis 1 and 2 by PC1 and 2. The comment also applies to the text itself (Ex. L190).

L183. Please describe in the first sentence what the triangle is made of.

L203. 'movement of a spatiotemporal pixel in variable space', please rephrase. A pixel cannot be moving spatiotemporally, like in a sliding puzzle.

L221-224. This should be described in the methods section and should be linked to a key research questions.

(5) Conclusion

L341. The results of the breakpoints analyses were not reported or discussed in the main text, therefore the statement 'To monitor gradual and abrupt changes in times of global change' do not hold.

Appendixes. Some results presented in the appendixes do not appear in the main text, e.g. Figures A1 and B1. The authors should maybe decide on the key results to be presented here and maybe save some others for a follow-up paper?

(6) Two final comments for reflexion:

- The authors have applied PCA on time series of 8day variables without considering any lag or accumulation effect in the response of a given variable. Would it be fair to say that legacy effects might not be captured adequately by such analysis?

- The authors refer to the MEI in the introduction as an example of a successful PCA-based indicator. Could the authors elaborate on the requirement for operationalising

their methods (e.g. if one would like to use the new indicators operationally, how frequently should the PCA be updated?).

---

## Referee Comment (RC3) · Anonymous Referee #3 · 20 Sep 2019

Review for Manuscript bg-2019-307

This manuscript entitled "Summarizing the state of the terrestrial biosphere in few dimensions" is well-thought and well-written, and fits the scope of Biogeosciences, so overall, I am favourable to get it published there. I do have some concerns which I would like to see addressed by the authors, and I also have several recommendations to improve the manuscript before getting it published. Please find these points below.

My first point regards the interpretation of the first to PCA components. Having the first related to productivity and the second to water availability is indeed interesting and useful to summarize that state of vegetation. However, I believe some more effort is needed to more clearly separate these 2 in their interpretation. Productivity is inevitably dependent on water availability, so in principle, one wonders why these would be the

[Figure]

first 2 components, which by definition should be orthogonal and 'unrelated'. I suppose this is perhaps because these refer to signals at different scales, PC1 describing an overall general state of potential productivity of the system at that location, while PC2 describes more events of water shortages and or excesses that are not directly related to the stationary potential productivity. Am I correct? Could you please clarify/elaborate on this to help readers better understand how these two axes should be 'read'.

Much related to the previous point, isn't it surprising that the 2 first principal components have such similar spatio-temporal patterns in Figure 3? These seem very highly correlated, which is something I would not have expected from the first two components which explain the maximum of variance in two orthogonal direction. Can you help me grasp this apparent paradox? In a way having such similar patterns make me wonder how useful having 2 PC is instead of only 1? Of course you do show the value of the 2D space in figure 2, but even there, much of the variation goes along the PC1 axis. Your selected cases in the anomalies in Figure 5 also generally go in the same direction of lower productivity coinciding with dryer conditions (Russian heatwave, droughts in Amazon), or vice versa (Floods in horn of Africa). Perhaps a stronger focus in general throughout the paper should be made on highlighting the much more specific cases where the two PCs give different but complementary information rather that going in the same direction.

I think you should also explore the third component. It does represent 9% of the variance, which is not so little, but above all it seems to be quite different from the first 2 in that it reacts much more to the albedo, which you hardly mention in the entirety of the manuscript. Could this be related to biophysical effects that vegetation could have on the climate? E.g. to understand where radiative vs non-radiative mechanisms dominate their effect on local temperature, for instance.

The behavior of the biosphere is much related to the elevation. While I know the effect of elevation should be reflected in the other variables, this is still dependent of modelling assumptions that may end up diluting the effect of elevation. Yet elevation is a

variable that is very well measured, and which could contribute to summarizing the terrestrial biosphere. So why not including such a variable in the PCA? I know changes in elevation are minimal (and probably very difficult to detect) and having a static variable with respect to all the other dynamic ones you propose is a bit odd, but still, what are your arguments for not doing so? I think some discussion on this is warranted.

The paper generally could be improved by curating more the structure. Several points on this: - Section 3.2 could benefit from some introduction naming what you intend to calculate first (get trends, test significativity, get breakpoints, hysteresis) before going in the details. This part could also be more pedagogic, providing more rational on why you do these things. - Parts of the 'discussion' should be much further after the 'results', such as lines 155-162 which should come in some kind of 'caveats and perspective about the method' section - Section 3.2 is very unbalanced with respect to 3.1. Probably best to reorganize to avoid 'sub-sub-sections' and have subsections from 3.1 to 3.5 - Parts describing concepts, such as Hysteresis (lines 235-246) should not appear in the results but before, either in methods or introduction.

Lines 74, 75: how do you manage intermittent gaps in the data? Does this affect your averages and your normalization? Also, please clarify if the normalization is based on the entire data cube for each variable, or is the normalization done per time frame?

Line 182: don't you mean sensible heat instead of latent heat?

Figure 1: caption could be more instructive, perhaps somehow say there what the reader should understand/read from the "rotation matrix".

Figure 7: surprised to see the strong pattern in Eastern Australia. Is this corroborated in other studies?

Regarding all trend analyses, make sure you more clearly mention in the captions the extend of the period you are considering, as these are not long-term trends and could thus be mis-interpreted.

For clarity and readability, figures with maps could benefit from either a dark background on the oceans or a line vector showing the coasts, as many of the colour scales use very light colours which are confounded with the white background.

I wonder if the breakpoint detection is really useful if it is not more mentioned and elaborated in the main text and just left in appendix. I would recommend to bring it in as a main figure if something strong can be extracted from there, and otherwise remove it entirely from the methods. Eventually you could include it in supplementary, but then include the description of the breakpoint methodology only there.

On the other hand, I would strongly recommend to integrate the Figure C1 in the main text as you do talk in detail about the Bowen ratio and how the 2 PCs do characterize it well.

Figure 1D I have a bit of a hard time to make good use of it as it is. Are the values in normalized units or absolute values? Would it not be prefereable to have the same scale for MSC min and MSC max? Do you refer to this figure in the main text.

There are some typos in several places. Make sure to address them.

---

## Referee Comment (RC4) · Anonymous Referee #4 · 20 Sep 2019

The authors present a well-written manuscript on the analysis of two principal components derived from a set of biosphere variables, one related to vegetation productivity and the other one related to water stress. The trajectories of those components over time reveal interesting seasonal patterns, inter-annual changes and anomalies, and can be used to track extreme events and state shifts of ecosystems/biomes. Therefore, I believe that this is a novel and relevant contribution to Biogeosciences.

My major concern lies in the fact that the authors select mainly variables related to productivity and water availability, and thus not surprisingly the PCA shows those two major axes. I wonder whether just selecting for example GPP and evaporative stress for the analysis of time trajectories would give the same results, but it might be easier to interpret than principal components representing a mix of variables. Can the

authors elaborate in more depth what is the advantage of using PCs in this context? For describing the state of the terrestrial biosphere, I think the authors are missing a very important component related to biodiversity, habitat quality, intactness, forest degradation and fragmentation. These aspects are crucial to describe the state of the terrestrial biosphere. There is still research needed to develop these as operational data streams, but a few examples are available at least at one point in time, e.g. Global Habitat Heterogeneity from EarthEnv, datasets from Global Forest Watch, Dynamic Habitat Indices DHI from Silvislab. This might not be sufficient (in terms of temporal resolution) to include it for this analysis, but the results from this study could be compared to those datasets (especially the DHI) and the need and relevance of global biodiversity and habitat intactness/quality information should be discussed.

Minor comments: L18: new satellite missions, add: Schimel, D., Schneider, F., Bloom, A., Bowman, K., Cawse-Nicholson, K., Elder, C., . . . Zheng, T. (2019). Flux towers in the sky: global ecology from space. New Phytologist, nph.15934. https://doi.org/10.1111/nph.15934

L25: green revolution, add: Chen, C., Park, T., Wang, X., Piao, S., Xu, B., Chaturvedi, R. K., . . . Myneni, R. B. (2019). China and India lead in greening of the world through land-use management. Nature Sustainability, 2(2), 122–129. https://doi.org/10.1038/s41893-019-0220-7

L27: changes are not only occurring in the onset of spring, but also browning trends, see:

- Garonna, I., de Jong, R., de Wit, A. J. W., Mücher, C. A., Schmid, B., & Schaepman, M. E. (2014). Strong contribution of autumn phenology to changes in satellite-derived growing season length estimates across Europe (1982 - 2011). Global Change Biology, 20(11), 3457–3470. https://doi.org/10.1111/gcb.12625

- Garonna, I., de Jong, R., & Schaepman, M. E. (2016). Variability and evolution of global land surface phenology over the past three decades (1982-2012). Global

Change Biology, 22(4), 1456–1468. https://doi.org/10.1111/gcb.13168

L35: if a principal component is a mix of productivity measures, I don't necessarily think it's more intuitive to interpret than a simple GPP map.

L63: What do you mean by "of parts"? Parts of what?

L75: Isn't this dependent on the coordinate system and/or projection? What is the coordinate system used? And why not try to use an equal-area projection (e.g. equal earth projection)?

L152: So what is contributing to the third component. It's still 9% of explained variance!

L162: Figure 1b is not very intuitive to me. What exactly does it show and how do you read from this that the first component represents productivity and the second hydrology? The figure doesn't seem to show any clear patterns to me. Could you also show the biplots of PC1 and 2, and PC2 and 3?

L177/178: check spelling

Figure 2: Very interesting figure! A degraded or stressed system might show different trajectories, could you somehow visualize the difference between intact and degraded ecosystems?

L258: check spelling

Figure 5: third line, the effects of the drought

Figure 6: This figure is a bit confusing to me. Could you improve the legends? I don't see an increase in seasonal amplitude in 6a, but maybe I just don't read this figure correctly. (b-c-d) seem to show the mean seasonal cycle and an event, but what do we see in 6a?

L305: changes that occurring?

L340

Additional research is needed to better represent biodiversity, habitat quality and intactness, forest degradation and fragmentation, etc... See:

- Jetz, W., Cavender-Bares, J., Pavlick, R., Schimel, D., Davis, F. W., Asner, G. P., . . . Ustin, S. L. (2016). Monitoring plant functional diversity from space. Nature Plants, 2(3), 16024. https://doi.org/10.1038/nplants.2016.24

- Chiarucci, A., & Piovesan, G. (2019). Need for a global map of forest naturalness for a sustainable future. Conservation Biology, 00(0), cobi.13408. https://doi.org/10.1111/cobi.13408

- Nicholas C. Coops, Michael A. Wulder, (2019). Breaking the Habit(at), Trends in Ecology & Evolution, Volume 34, Issue 7, https://doi.org/10.1016/j.tree.2019.04.013.

L352: detected ina a similar fashion

---

## Author Comment (AC1) · 11 Nov 2019

[11pt]article [utf8]inputenc [T1]fontenc graphicx grffile longtable wrapfig rotating [normalem]ulem amsmath textcomp amssymb capt-of hyperref

[Figure]

**Replies to the Anonymous Referees**

Guido Kraemer, Gustau Camps-Valls, Markus Reichstein, and Miguel D. Mahecha

November 11, 2019

**Contents**

**1 Anonymous Referee #1**

The authors thank the reviewer for the his time and thorough comments, we think that the comments greatly improved the manuscript. We have addressed them below.

**1.1 General Remarks**

I appreciated reading the discussion paper Summarizing the state of the terrestrial biosphere in few dimensions by Guido Kraemer and colleagues. The paper presents an approach for summarizing key variables on the terrestrial biosphere into fewer independent components using established multi-variate methods. They exemplify their

approach by showing several trajectories across space and time and by highlighting some major anomalies visible in their data.

While the work is well presented and scientifically sound, I have some major concerns regarding the publication of the manuscript in its current form:

**1.1.1 Authors' reply**

We thank the reviewer for his positive and very thorough review and the very helpful comments that we have now addressed the open issues as we will show below. We especially thank the reviewer for the detailed review of the overall structure of the manuscript and the many small details that have been improved due to is comments.

**1.2 Concerns**

**1.2.1 1) The number of dimensions**

The authors state that the first two components explain large parts of the variance and that the 'knee' is reached with the second component. However, inspecting Figure 1a, it seems that the 'knee' is reached with the third component, which still explains 9% of the variance. I was a little confused that the third component was disregarded throughout the whole manuscript, without giving a strong justification. Figure 2b indicates that the third component might be strongly connected to albedo. I encourage the authors to either expand their analysis to also include the third component, or to give a very strong argument for its exclusion. As it stands now, the decision to only inspect the first two components is very subjective.

1. Authors' reply The reviewer is right that the 3rd dimension still contains important information, therefore we included component 3 into the manuscript. We think

that the addition of the third component improved the manuscript substantially and want to thank the reviewer for this.

2. Changes:

- Added axis 3 to the manuscript (for details, see the list of changes at the end of this letter).
- Flipped axis 3 so that higher values for PC3 mean higher albedo

**1.2.2   2) Scientific novelty and usefulness**

I am missing a strong discussion/conclusion on how the manuscript advances scientific progress. Putting it into simple terms, the authors apply PCA – a widely used and established method – to a set of existing data sets. As such, it is not really a novel methodological development, but rather a demonstration of what could be done with global datasets as provided though the Earth System Data Lab. While this is not a deal-breaker per sé, the authors could greatly advance their manuscript by explaining how this approach can be used by other scientists, that is how it will advance the science of the terrestrial biosphere.

1. Authors' reply Thank you for this critique and comment which has many dimensions. At first glance the reviewer is right: we simply applied a PCA to a highly curated global data set - a data cube contained in the Earth system data lab. But, altough the method is similar to EOFs in climatology, where the matricization (the flattening of the 4th order tensor, variables $\times$ time $\times$ longitude $\times$ latitude, to a matrix) happens maintaining time, there are some differences in our approach: We are maintaining both, space and time and reduce only over the variables, as far as the authors are aware, this has not been done on global data. This is in our view an innovation, as we account, for the first time, for the many redundancies in

high-dimensional Earth observations. We have carefully reviewed the literature, but do not find a study that has investigated the global covariations of multiple Earth observation data streams. This is the main novelty of our work. Also, the use of a simple PCA algorithm is not incidental here: we seek for a method that learns a data transformation that is invertible, and allows us to measure/compute the reconstruction error in meaningful physical units. This cannot be done with more complicated/sophisticated nonlinear machine learning methods, where the (probably more accurate) transform is hard to analyze. We have included a comment in that direction, and pointed out the advantages and shortcomings of PCA versus other nonlinear dimensionality reduction methods in section 3.1.

2. L201, Added:

**1.3 Differences from other PCA-type analyses**

One of the most popular applications of PCA in meteorology is the EOF analysis, which are typically done with single variables, i.e. on a data set with the dimensions $lat \times lon \times time$, although EOFs can be calculated from multiple fields. The resulting vector of indicators is calculated by multiplying the original data with the eigenvectors (typically only the first one), which represents the state of the entire spatial extent at a certain point in time and reducing over the spatial dimension. This is a PCA analysis that is very similar from a mathematical standpoint but very different from how we interpret the result. In an EOF, eigenvectors form maps that represent standing oscillations which are uncorrelated, while in the present study, the eigenvectors represent the influence of variables on the final indicators. Another important difference consists in EOFs maintaining the temporal dimension, while the present study maintains the spatial dimensions as well as the time dimension.

Ecological analyses often compare different sites, i.e. a spatial dimension, if

the site measurements have repetitions in time. The observed variables often are species assemblages or environmental properties, which is the feature that is being reduced by the method of dimensionality reduction. The maintained dimensions therefore are space and, if present, time. Therefore the ecological application of dimensionality reduction is more similar to the present analysis than EOFs, the main difference lies in the type of features used, here we use variables that describe the exchange of ecosystems with their environment, while ecological analyses usually use species assemblage data. Another important difference lies in scale and type of data, ecological analyses usually use plot level data, observed at certain points, while our analysis uses global data that is arranged in a grid.

The present analysis uses multivariate data streams that are not bound to a certain point in time or space and removes the redundancies in these data streams and leaves the user with fewer indicators to worry about. In the future the number of data streams and the amount of data will only increase and therefore the utility of such a method will increase, too.

**1.3.1  3) Too many results in the appendix**

Many of the results are buried in the Appendix but never picked-up in the main text. In fact, Figure A1, B1, D1 and C1 were never referenced in the main text. The authors thus present many results in the Appendix that are not discussed in the main manuscript and thus the reader is left alone with her own interpretation. As some of the results are quite crucial for evaluating the method (e.g., the errors presented in B1), I strongly encourage the authors to thoroughly discuss them in their manuscript.

1. Authors' reply: Thank you for the observation! We agree that we have a lot of results in the appendix. To improve this situation, we have moved parts of the

appendix into the main text. We have also added references to the figures into the text.

2. Changes:

- Moved the section "Reconstruction Error" from the appendix to L201.
- Moved figure C1 ("bowen ratio") into the text.
- We added the corresponding references. L340: A1, L183: B1, L195: C1 (was already there), L210: D1 (was already there, L339: E1 (was already there).

**1.3.2   4) Writing**

The writing needs improvement for turning this already good manuscript into an excellent manuscript. For example, the authors often describe their figures, instead of the results (Figure X shows. . . ). It would be much more interesting to read about the main result instead (A influences B (Figure X)). I am sure the senior authors of this manuscript can do a great job in revising the manuscript to make it more accessible and exciting for the reader.

1. Authors' reply: We thank the reviewer for the pointing this out, and have revised many aspects of the paper, we hope that we have corrected the manuscript accordingly.

**1.3.3   5) Spelling/grammar**

There are some wording and spelling/grammar issues, some of which listened below:

1. Autors' reply Thanks for the thorough revision provided. We have corrected all suggested minor changes, and commented further on the critical ones below.

2. L. 16: Suggest removing 'the' before 'global'.

   (a) Authors' reply: Additionally added a "negative" for emphasis.

   (b) Change: the global impacts -> negative global impacts

3. L. 27: Spring is not a phenological event. Could use onset of bud-flush or similar.

   (a) Authors' reply: Thanks for catching this detail.

   (b) Changed from: In general, phenological patterns are changing in the wake of climate change, leading primarily to changes in the onset of spring (**??**).

   (c) Changed to: Changes in the onset of spring and autumn (**?**) change the length of the growing season, and cause large scale changes in phenological patterns ({**?**}schwartzgreen-wave1998, parmesanecological2006}.

4. L. 74: Not clear how standardization accounts for differences in scales. What scales? Spatial? Temporal?

   (a) Authors' reply: In deed, the wording is a bit ambiguous. We have changed it to make clear that we mean scale in a statistical sense here.

   (b) Changed from: In this study, each variable was normalized globally to zero mean and unit variance to account for the differences in scales. Because the area of the pixel changes with latitude, the pixels were weighted according to the represented surface area.

   (c) Changed to: In this study, each variable was normalized globally to zero mean and unit variance to account for the different units of the variables, i.e. transform the variables to have standard deviations from the mean as the common unit.

5. L. 138: The breakpoint detection comes out of the blue. Why is this done? What was the rational behind? This needs a decent introduction.

(a) Authors' reply: The reviewer is right that we do breakpoint detection without properly introducing it. We have added a reference to fig. A1 and some introduction.

(b) Added:

    i. End of Section 3.2.5 We added a paragraph at the "Trends in Trajectories" section to show the reader that trends are not the only way to detect changes in a trajectory and reference fig. A1

    ii. L29, added: Extreme events are temporary shifts, shifts where ecosystems changes their qualitative state permanently can also occur due to changing environmental conditions or direct human influence (**?**), detecting these changes is of vital importance for their mitigation (**?**).

6. L. 142: Same as above. The term hysteresis is never introduced before, but then explained in the results section (L. 239). As a reader, I would love to hear the details upfront, instead of reading about them in the results/discussion.

    (a) Authors' reply: This was missing from the introduction, we thank the reviewer for noticing this, we have remedied the situation.

    (b) Changes:

       • Moved the definition of "Hysteresis" to the "Methods" section and changed the first paragraph of Section "Hysteresis" to:
"The alternative return path between ecosystem states forming the hysteresis loops arise from the ecosystem tracking seasonal changes in the environmental condition, e.g. summer–winter or dry–rainy seasons (fig. **??**b))."

       • Added to the introduction (L 27):
"Hysteresis in ecosystems requires a better understanding as it can give us important information on limiting factors (**?**) and memory effects

({**?**)mahechacharacterizing2007, blonderpredictability2017} and may inhibit the return of ecosystems to the original state."

7. L. 148: Maybe include an example figure here, instead of referencing to the results already.

   (a) Authors' reply: The hysteresis may be a complex topic for people not familiar with it, we thank the reviewer for pointing this out and have added a conceptual figure that hopefully makes the concept easier to understand.

   (b) Changes: Added an example figure (Figure 1 in the new version of the manuscript, "Methods" section) with the four most common cases and changed the reference.

8. L. 151: 'We see that...' is not a good opener. Directly describe the result, be precise and upfront (e.g., The first two components explained 73% of the variance (Figure 1a))

   (a) Authors' reply: Removed "We see that"

9. L. 160: What is the pre-imaging problem? Please do not assume that the reader reads up the details in the reference provided. Either avoid naming it or give a brief description.

   (a) Authors' reply: Again we thank the reviewer for pointing out that this is a topic that the target audience may not be acquainted to. We have improved the description and hopefully made the concept understandable to everyone.

   (b) Changed from: The salient feature of PCA is that an inverse projection is well defined and allows for a deeper inspection of the errors, which is not the case for nonlinear methods due to the pre-imaging problem (**??**).

   (c) Changed to: The salient feature of PCA is that an inverse projection is well defined and allows for a deeper inspection of the errors, which is not the

case for nonlinear methods which learn a highly flexible transformation that is hard to invert. Therefore interpretability of the transform in meaningful physical units in the input space is often not possible. In the machine learning community, this problem is known as the "pre-imaging problem" (**??**) and is a matter of current research.

10. L. 162: Again, not the best opener. The first sentence of a paragraph should summarize the main point of the paragraph (topic sentence), allowing the reader to skim through the manuscript. This sentence just describes where the reader can find a result, but nothing about the result itself.

   (a) Authors' reply: Thanks for pointing this out, we have changed some thing and hope that the manuscript is more readable now.

   (b) Changes:
     • Removed the sentence.
     • Added a reference to the rotation matrix equation to the caption of the first figure in the results. (Eq. **??**)
     • added "..., see fig. **??**b." at the descriptions of the components (L. 163, and L. 174)

11. L. 164: Odd formulation (two times related).

   (a) Authors' reply: Thank you for noticing! We have changed the sentence accordingly.
      i. From: These variables are related because they are all directly related to primary productivity.
      ii. To: These variables are related due to their importance for primary productivity.

12. L. 174ff: his paragraph actually described the indicators used and does not discuss the results. This could go into the methods description or should be more clearly related to the actual results.

   (a) Authors' reply: This paragraph describes PC2 and discusses how the variables that make up PC2 are related, therefore we have decided to leave it in as a discussion of PC2.

13. Figure 2: What are 'some points'? How were they chosen?

   (a) Authors' reply: It says so in the caption: "The trajectories were chosen to fill a large area in the space of the first two principal components."
   (b) Changes: "fill" -> "cover"

14. L. 139: As said before, this is rather introduction than results/discussion. I would have very much appreciated reading this in the introduction.

   (a) Authors' reply: This is the wrong line number, the Reviewer is probably referring to the description of the Bowen ratio as this should be mentioned in the introduction, indeed. We have added the Bowen ratio to the introduction and changed the paragraph to highlight the main result.
   (b) Changed L. 43
      i. from: Extracting the dominant dynamics from high-dimensional observations is a well-known problem in many disciplines. In climate science, for example, it is common to summarize atmospheric states using Empirical Orthogonal Functions (EOF), also known as Principal Component Analysis (PCA; **?**).
      ii. to: Extracting the dominant features from high-dimensional observations is a well-known problem in many disciplines, one approach is to

manually define indicators that are know to represent important prop-
erties, such as the "Bowen Ratio" (**?**), another one consists in using
machine learning to extract these features. In climate science, for ex-
ample, it is common to summarize atmospheric states using Empirical
Orthogonal Functions (EOF), also known as Principal Component Anal-
ysis (PCA; **?**).

(c) Moved the the following sentence from L. 201 to L. 194 The Bowen ratio
embeds well into the subspace spanned by the first two PCs, see fig. **??**.

15. L. 258: rephrase: . . . and can therefore be interpreted. . .

   (a) Authors' reply: Thank you for finding this, rephrased the entire sentence.

      i. From: These anomalies have a directional component and can be there-
      fore be interpreted the same way as the original PCs which contain in-
      formation of the underlying variables that were affected. In this sense,
      one can infer the state of the ecosystem during an anomalous state.

      ii. To: These anomalies have a directional component which makes them
      interpretable the same way as the original PCs, therefore one can infer
      the state of the ecosystem during an anomaly.

16. L. 282: Again, put the result in the spotlight, not the figure showing the result.

   (a) Authors' reply: The reviewer is right, this also counts for some of the other
   paragraphs describing that figure, thank you for pointing this out. We hope
   to have remedied the situation with the following changes:

   (b) Added L. 282: The seasonal amplitude of the trajectory in the Brazilian Ama-
   zon increases due to deforestation and crop growth cycles.

   (c) Added L. 290: The 2010 Russian heatwave has a very clear signal in the
   trajectories, . . .

(d) Added L. 300: The 2003 European heatwave is reflected in the trajectories just a the 2010 Russian heatwave.

17. L. 305: Occur instead of occurring.

    (a) Authors' reply: Changed, thanks.

18. L. 312: Move 'especially' after 'showed'.

    (a) Authors' reply: Changed, thank you.

19. L. 313: Repeats methods.

    (a) Authors' reply: Thanks for noticing, we have removed the phrase and added . . . patterns of trends . . . to the next sentence.

20. L. 320: Why did you calculate the trends from the full data? Would it have been better to use the growing season as well to facilitate comparison? Please give a reasoning why you do it differently.

    (a) Authors' reply: The reviewer is right, that usually these kind of analyses are made on the growing season only. Because of simplicity of the analysis we opted to do the analysis this way, just as with the breakpoints we did not want to develop complicated methods for detecting the growing season from PC1 because this is not the scope of this paper. The analyses on the resulting indicators are simple and straightforward because of their exploratory nature. The next question would have been, how to limit PC2 and PC3? Use the wet/dry season for PC2 because it shows water, and summer/winter for PC3, or also use the growing season? Using growing season data only, we probably could have found stronger trends in PC1, but this could be an interesting topic for future research.

21. L. 324: Something odd with the sentence starting with 'Inside. . .'.

(a) Authors' reply: Thanks for finding this one, fixed!

    i. From: The finding of **?** is not reflected in our data, especially compared to the areas surrounding the Congo basin, we can find only minor browning effects. Inside the basin and our findings are more in line with the global greening (**?**), which show a browning mostly outside the Congo basin.

    ii. To: The finding of **?** is not reflected in our data, especially compared to the areas surrounding the Congo basin, we can find only minor browning effects inside the basin and our findings are more in line with the global greening (**?**), which show browning mostly outside the Congo basin.

22. L. 327: Remove 'a' before 'browning'.

    (a) Authors' reply: Removed, thank you.

23. L. 349: The breakpoints are actually never shown, nor discussed. The conclusion is thus not really based on data here.

    (a) Authors' reply: The reviewer is right, we have added the breakpoints to the introduction, thank you for pointing this out.

    (b) L. 29, added: Extreme events are temporary shifts, shifts where ecosystems changes their qualitative state permanently can also occur due to changing environmental conditions or direct human influence (**?**), detecting extreme events and breakpoints is of vital importance for their mitigation (**?**).

24. L. 352: in, not 'ina'.

    (a) Authors' reply: Changed, thank you.

**2  Anonymous Referee #2**

The authors thank the reviewer for the his time and thorough comments, we think that the comments greatly improved the manuscript. We have addressed them below.

**2.1  General Remarks**

**2.1.1  General assessment**

This is a very interesting paper addressing some important issues of big data analysis for ecology studies. It is rich in analyses and provides some new views on an old method (PCA). I particularly liked the analysis of trajectories that I found quite powerful, notably for case studies.

1. Authors' reply We thank the reviewer for this positive review, we have addressed all the concerns below.

**2.1.2  Key research question**

Yet I found it difficult to understand what key research questions are addressed in this paper. This is important to clarify at the end of the introduction as the authors is providing us with a suit of analyses that may resemble (for non PCA-expert) an attempt of addressing many (all?) questions without real rationale. The readers need to have a clear (concise) view of the objectives of this paper, and they need to be guided through the analyses by referring back to the main research questions.

1. Authors' reply: Thank you for pointing this out, the reviewer is right, the paper may appear to try to solve too many problems. We have added a paragraph at the end

of the introduction to clarify the focus of the paper. The main motivation and goal of this paper is the lack of a systematic data-driven approach to explain the main features in Earth system data cubes in the literature. We first introduce a method to create such summarizing indicators in the form of a simple yet effective PCA, then we apply the method to a global set of representative variables describing the biosphere. Finally, to prove the effectiveness of the method, we give interpretations of the resulting set of indicators and explore the information contained in the indicators by analyzing them in different ways and relating them to well known phenomena. We have explicitly declared such motivation and approach at the end of the introduction section. Thanks for pointing this out.

2. L. 65 Added: First we introduce a method to create such indicators, then we apply the method to a global set of variables describing the biosphere. Finally, to prove the effectiveness of the method, we give interpretations of the resulting set of indicators and explore the information contained in the indicators by analyzing them in different ways and relating them to well known phenomena.

**2.1.3   Input data may cause the resulting axes**

In addition, I also have a major concern related to the set of inputs data used to feed the PCA. I agree that PCA is a powerful tool to deal with correlated variables, yet I have difficulties understanding why the authors have decided to include variables that are obviously highly correlated. To my opinion, vegetation productivity proxies are overrepresented as well as those related to water availability and stress. It puts some doubts in my head as to whether the finding of PC1 (primary productivity) and PC2 (surface hydrology) driving the state of the biosphere in space and time is truly original (or just purely mathematical). It is therefore important for the authors to justify the set of original variables. A suggestion could also be to decrease the number of input variables (removing obvious redundant proxies) as the amount of data to be condensed

is mainly coming from the 8days interval used for the analysis.

1. Authors' reply: PCA extracts correlated variables, therefore the resulting axes will not change much if more or less variables are added that represent a certain aspect of the ecosystem. What does change are the explained variances of the resulting axes, i.e. including more variables that are proxies for primary productivity will cause this axis to explain more variance. The set of covariates we chose constitutes a large complementary and representative set that describe the exchange of mass and energy of the biosphere with the atmosphere. We have added a justification for the used variables:

   - The data was chosen from the entirety of the variables in the ESDL (at the time of analysis), meteorological variables were discarded (e.g. air temperature and precipitation), as well as variables with obvious problems in their distribution (e.g. burnt area contains too many zeros).
   - The used variables are mostly describing the mass and energy exchanges of the ecosystem with the atmosphere and we have shown that here, the most important drivers are found by a PCA.

2. L. 73 Added: For this study we chose all the variables available in the ESDL v1.0 (the most recent version available at the time of analysis), divided the available variable into meteorological and biospheric variables and discarded the biospheric variables. We also discarded variables with distributions that are badly suited for a linear PCA (e.g. burnt area contained too many zeros) and variables with too many missing values. The only data set that was added post hoc was fA-PAR which represents an important aspect of vegetation which was not available in the data cube at the time on analysis (it is part of the data cube now).

**2.2 Detailed comments**

Finally I also have other comments and concerns - notably related to the structure of the manuscript - that would need to be addressed by the authors prior publication of their research (see attached report for details).

**2.2.1 (1) Abstract**

The authors start off the abstract by mentioning the importance of detecting abrupt and gradual changes in terrestrial ecosystem but do not develop further in the introduction. In the method section, the detection of breakpoints reappears but no results are presented or discussed (except for the appendix A). The authors should decide whether to consider the detection of abrupt changes as a real research question for this study.

1. Authors' reply: The reviewer is right, do not really go into detail in the analysis of breakpoints. To remedy this, we have changed the first sentence and made it clear that there is a proof of concept analysis in the appendix.

2. Changed the first sentence to: In times of global change, we must closely monitor the state of the planet in order to understand the full complexity of these changes.

3. L. 339 added: Another way to detect changes to the biosphere consists in the detection of breakpoints, which has been applied successfully to detect changes in global NDVI time series (**??**), or generally to detect changes in time series (**?**). A proof of concept analysis can be found in fig. **??**, we hope that applying this method to indicators instead of variables can detect a wider range of breakpoints analyzing a single time series.

**2.2.2 (2) Introduction**

As stated in my main comment, I find that there is somewhat a mismatch between the introduction and the method section. In the introduction, the authors touch upon many issues related to assessing and attributing changes of biosphere properties. However apart from creating a new set of independent, 'essential' variables, they do not clearly mention what other research questions this study is going to address; whereas in the methods they mention PCA, trend and breakpoints analyses. Clearly stating the research questions for this study would help the readers to understand the rationale behind each analysis.

1. Authors' reply: The reviewer is right, we have added the research questions to the introduction. We have extended L. 22–31 to contain the research questions.

2. L. 22ff, changed to: Regional trends of vegetation greening and browning that have been attributed to fertilization effects on the one hand, and long-term climate change on the other, need to be understood (**???**). Changes in the seasonal cycles of primary production, e.g. decreased seasonal amplitudes in "cold" ecosystems due to warmer winters (**?**) or increased seasonal amplitude in agricultural areas due to the so called "green revolution", are expected (**??**). Changes in the onset of spring and autumn (**?**) change the length of the growing season, and cause large scale changes in phenological patterns (**??**). Hysteresis in ecosystems requires a better understanding as it can give us important information on limiting factors (**?**) and memory effects (**??**) and may inhibit the return of ecosystems to the original state. Additionally, we are confronted with cascading effects induced by today's increasing frequencies and magnitudes of extreme events (**??**) which are yet to be fully understood (**??**). Extreme events are temporary shifts, shifts where ecosystems changes their qualitative state permanently can also occur due to changing environmental conditions or direct human influence (**?**), detecting extreme events and breakpoints is of vital importance for their

mitigation (**?**). The question is, how to uncover and summarize effects of this kind from the wealth of available global data streams? Do we need to develop specific solutions for every observed phenomenon or can we develop a single approach to uncover a wide variety of phenomena?

Extracting the dominant features from high-dimensional observations is a well-known problem in many disciplines, one approach is to manually define indicators that are know to represent important properties, such as the "Bowen Ratio" (**?**), another one consists in using machine learning to extract these features.

**2.2.3  (3) Data and methods**

1. Better description of the data The description of the data slightly too minimalistic, including in the appendix F. Mentioning the input data (satellite, climate or others) feeding into each dataset would be helpful. The observation period used for this study is also not mentioned.

   (a) Authors' reply: We thank the reviewer for pointing this out. We have added the limits of the time dimension and the type of grid in
   (b) L 72, changed to: The data streams are harmonized as analysis ready data on a common spatiotemporal grid (equirectangular 0.25° in space and 8 days in time, 2001–2011), forming a 4d hypercube, which we call a data cube.
   (c) Appendix F: Was augmented with the origins of the data

2. L. 75, Mention projection This statement is not always valid (e.g. in the case of equal-area projection). The sentence would be clearer if the authors would mention the projection system used here.

   (a) Authors' reply: We thank the reviewer for pointing this out. See previous response.

3. L77. Better explanation of PCA The authors mentioned that they used a modified PCA, reading from the description given in the following lines, the PCA applied here seems to be standard. Could the authors provide some explanations to why / how the PCA has been modified? It should also clarify whether they applied the PCA in s or t-mode.

   (a) Authors' reply: We have clarified the PCA analysis by discussing it in the context of frameworks describing PCA in the context of climatology and ecology and hope that this will help with the understanding of the method.
      - The PCA is a decomposition of the correlation matrix.
      - Building the correlation matrix is not standard due to the big data aspects,
      - and the spatial extension, both of which require a lot of care in the calculation of the covariance matrix, which is described in the "Methods" section.
      - The dimensions we summarize are new, there are a number of different frameworks (S- vs. T-mode in climatology, Q- vs. R-mode in ecology, and primal vs. dual modes in machine learning) that describe standard applications of PCA, none of which give an exact description of the analysis done here. We have added a section describing the relation of the present analysis with these frameworks.

   (b) L201, added:

      2.3   Relations to other PCA-type analyses

      One of the most popular applications of PCA in meteorology is the EOF analysis, which is typically done with single variables, i.e. on a data set with the dimensions $lat \times lon \times time$, although EOFs can be calculated from multiple fields.

EOFs can be calculated in S-mode and R-mode. If we matricize our data cube $X$ so that we have time in rows and $lat \times lon$ in columns, then S-mode PCA works on the correlation matrix of the combined variable and space dimension. In T-mode, the PCA works on the correlation matrix formed the time dimension (**?**). The PCA presented here works slightly different: (1) We did a different matricization ($lat \times lon \times time$ in rows and variables in columns) and then (2) the PCA works on the correlation matrix formed by the variables, therefore in this framework we could call this a V-mode PCA.

Ecological analyses use PCA usually with matrices of the shape $object \times descriptors$, when calculating the PCA on the correlation matrix formed by the objects, then we it is called a Q-mode analysis, when the PCA is applied on the correlation matrix formed by the variables, then it is called an R-mode analysis (**?**). The PCA done in this study is closest to an R-mode analysis, in the present case the descriptors are the various data streams and the objects are the spatiotemporal pixels.

4. Per-pixel analysis It would be nice here to make a link to the (extended – see comment above) research questions in order to understand directly the rationale for such analyses.

   (a) Authors' reply: The link was really unclear, we have added more research questions to the introduction (see previous replies) and are now mentioning the research questions.

   (b) L. 127, added: We calculated the trends of the indicators and the trends of the seasonal amplitude of the indicators, for the trends we used the Theil–Sen estimator.

   (c) L. 138, added: When looking for disruptions in trajectories, breakpoint detection provides a good framework for analysis.

**2.3.1 (4) Results**

1. General comment: I highly suggest to split the results and discussion into two separate sections. It will facilitate the reading and will allow the authors to emphasise better the originality of their work. Example: L155-161, L164-173, L175-182, L235-246, etc. should not be in a results section s.s., but would rather belong to a discussion (or even introduction or method). Please consider at least moving all methods description and introduction to new concepts to the respective adequate sections.

    (a) Authors' reply We thank the reviewer for this suggestion, but we think that a joint results and discussions section is the better choice, as it allows for the results and their discussions to be closer and easier to follow.

2. L153 and Figure 1 The authors mentioned that there is a knee at component 2. I believe it is rather at component 3. This component still contribute to the total variability to a share of almost 10%, therefore the authors should either include it in the rest of the analysis or provide an adequate justification not to. Also I generally miss a figure presenting together the temporal and the spatial patterns for the main PCs. This could be put as supplementary material. In the caption of Fig.1 I would recommend to change the term axis 1 and 2 by PC1 and 2. The comment also applies to the text itself (Ex. L190).

    (a) Authors' reply: The third component was missing, indeed. We have added it to the paper, we thank the reviewer for pointing this out, as it improved the manuscript substantially. The spatiotemporal figure was also missing and we have added it, this was an oversight of our part and we corrected it. We have also unified the terminology, axis is now never used to describe principal components.

    (b) Changes:

- We have added the third component to the manuscript.
- Added appendix F with joint time and space patterns.
- Removed the term axis when in designated a component in the entire manuscript.

3. L183 Please describe in the first sentence what the triangle is made of.

   (a) Authors' reply We have provided a better description of the figure.

   (b) Changed to: The bivariate distribution of the first two principal components form a triangle (gray background in fig. **??**a).

4. L203 'movement of a spatiotemporal pixel in variable space', please rephrase. A pixel cannot be moving spatiotemporally, like in a sliding puzzle.

   (a) Authors' reply The pixel is moving in the vector space, this formulation is easily misunderstood and we have changed it therefore. We thank the reviewer for pointing this out.

   (b) From: The principal components may be used to summarize the movement of a spatiotemporal pixel in variable space, so that they represent the current state of the ecosystem at a certain location in space and time (fig. **??** left column) or time of year of the mean seasonal cycle of the pixel (fig. **??** right column).

   (c) To: Because the first few principal components represent most of the variability of the space spanned by the observed variables, they summarize the state of a spatiotemporal pixel efficiently. This means that they track the state of a local ecosystem over time (fig. **??** left column) or, in case of the mean seasonal cycle, time of the year (fig. **??** right columns). For a representation of the state of the first three components in time and space, see appendix fig. **??**.

5. L221-224 This should be described in the methods section and should be linked to a key research questions.

   (a) Authors' response We thank the reviewer for pointing out this oversight. We have added a definition of the means seasonal cycle to the methods and mention it in the introduction.

   (b) L. 56, changed to: Here, we aim to summarize these high-dimensional surface dynamics and make them accessible to subsequent interpretations and similar analyses as the original variables, such as mean seasonal cycles(MSC), anomalies, trend analyses, breakpoint analyses, and the characterization of ecosystems.

   (c) L. 126, added: The mean seasonal cycle at a certain day of year is the mean of all values of a variable at a certain day of year and describes the average characteristics of a location. The anomaly of the mean seasonal cycle is the observed value at a certain point in time minus the mean seasonal cycle. The anomaly gives an idea, if there is the current value is normal or extreme.

**2.3.2 (5) Conclusion**

1. L341 The results of the breakpoints analyses were not reported or discussed in the main text, therefore the statement 'To monitor gradual and abrupt changes in times of global change' do not hold.

   (a) Authors' reply: We thank the reviewer for pointing this out and hope that we have remedied the situation.

   (b) Changed the beginning of the conclusion to: To monitor the complexity of the changes occurring in times of an increasing human impact on the environment . . .

2. Appendixes Some results presented in the appendixes do not appear in the main text, e.g. Figures A1 and B1. The authors should maybe decide on the key results to be presented here and maybe save some others for a follow-up paper?

3. Authors' reply: We thank the reviewer for pointing this out and all appendices should be referenced in the text now.

**2.3.3 (6) Two final comments for reflexion:**

1. Legacy effects: The authors have applied PCA on time series of 8day variables without considering any lag or accumulation effect in the response of a given variable. Would it be fair to say that legacy effects might not be captured adequately by such analysis?

   (a) Authors' reply: The method ignores lag and memory effects, lag effects may still be captured implicitly in the components but there will never be a "memory axis". Something like this may be captured using a combination of autoencoders and LSTMs but as far as the authors know, no one ever attempted an analysis like this.

2. Operationalization: The authors refer to the MEI in the introduction as an example of a successful PCA-based indicator. Could the authors elaborate on the requirement for operationalising their methods (e.g. if one would like to use the new indicators operationally, how frequently should the PCA be updated?).

   (a) Authors' reply: Applying a trained PCA is very simple and computationally efficient, the trained PCA should also be quite stable and therefore we assume that updates don't have to happen frequently. The implementation with 'WeightedOnlineStats.jl' would theoretically allow a very efficient update with every step, but we assume that this will not be necessary. For a real time

application of the method, the most important limitation is that only real time data can be used. This limits the type of data that that can be used, as most of the data we used here are created years after collecting the satellite or field observations.

**3 Anonymous Referee #3**

The authors thank the reviewer for the his time and thorough comments, we think that the comments greatly improved the manuscript. We have addressed them below.

**3.1 General Remarks**

This manuscript entitled "Summarizing the state of the terrestrial biosphere in few dimensions" is well-thought and well-written, and fits the scope of Biogeosciences, so overall, I am favourable to get it published there. I do have some concerns which I would like to see addressed by the authors, and I also have several recommendations to improve the manuscript before getting it published. Please find these points below.

**3.1.1 Authors' reply:**

We thank the reviewer for the positive comment and hope that we can address all mentioned concerns and recommendations.

**3.2 Better explanation for the interpretation**

My first point regards the interpretation of the first to PCA components. Having the first related to productivity and the second to water availability is indeed interesting and

useful to summarize that state of vegetation. However, I believe some more effort is needed to more clearly separate these 2 in their interpretation. Productivity is inevitably dependent on water availability, so in principle, one wonders why these would be the first 2 components, which by definition should be orthogonal and 'unrelated'. I suppose this is perhaps because these refer to signals at different scales, PC1 describing an overall general state of potential productivity of the system at that location, while PC2 describes more events of water shortages and or excesses that are not directly related to the stationary potential productivity. Am I correct? Could you please clarify/elaborate on this to help readers better understand how these two axes should be 'read'.

Much related to the previous point, isn't it surprising that the 2 first principal components have such similar spatio-temporal patterns in Figure 3? These seem very highly correlated, which is something I would not have expected from the first two components which explain the maximum of variance in two orthogonal direction. Can you help me grasp this apparent paradox? In a way having such similar patterns make me wonder how useful having 2 PC is instead of only 1? Of course you do show the value of the 2D space in figure 2, but even there, much of the variation goes along the PC1 axis. Your selected cases in the anomalies in Figure 5 also generally go in the same direction of lower productivity coinciding with dryer conditions (Russian heatwave, droughts in Amazon), or vice versa (Floods in horn of Africa). Perhaps a stronger focus in general throughout the paper should be made on highlighting the much more specific cases where the two PCs give different but complementary information rather that going in the same direction.

**3.2.1 Authors' response:**

While the reviewer is right that ecosystem productivity is dependent on water availability, the availability of water can be restricted due to several reasons which are reflected by PC2. We have added a paragraph to explain this more extensively.

1. L 189 added: PC2 separates the two most important factors that limit water availability: The lack of available water and frozen water. The first two components are (due to their construction by PCA) orthogonal, if we only took ecosystem with a dry season, then the "water component" would explain much less variance or even disappear, because ecosystem productivity would have a strong negative correlation with dryness. This would be equivalent to removing the lower left corner of the distribution triangle in fig. **??**a and b. If we only looked at ecosystems that ceased productivity in winter, then we would find a strong positive correlation with the current "water component" and ecosystem productivity and the component would explain much less variance or even disappear. This would be equivalent of removing the lower left corner of the distribution triangle in fig. **??**a and b. But because we have both relations, which can be seen from the triangle in the background shading of fig. **??**a and b, the "water component" is (1) orthogonal to the productivity component and (2) is the second most important component.

2. L 232 added: Although the principal components are globally uncorrelated, they covary locally (see fig. **??**). Ecosystems with a dry season have a negative covariance between PC1 and PC2 while ecosystems that cease productivity in winter have a positive covariance.

**3.3 Explain component 3**

I think you should also explore the third component. It does represent 9% of the variance, which is not so little, but above all it seems to be quite different from the first 2 in that it reacts much more to the albedo, which you hardly mention in the entirety of the manuscript. Could this be related to biophysical effects that vegetation could have on the climate? E.g. to understand where radiative vs non-radiative mechanisms dominate their effect on local temperature, for instance.

**3.3.1  Authors' response:**

This is really a good suggestion and we added component three.

**3.4  Include static variables?**

The behavior of the biosphere is much related to the elevation. While I know the effect of elevation should be reflected in the other variables, this is still dependent of modelling assumptions that may end up diluting the effect of elevation. Yet elevation is a variable that is very well measured, and which could contribute to summarizing the terrestrial biosphere. So why not including such a variable in the PCA? I know changes in elevation are minimal (and probably very difficult to detect) and having a static variable with respect to all the other dynamic ones you propose is a bit odd, but still, what are your arguments for not doing so? I think some discussion on this is warranted.

**3.4.1  Authors' response:**

We thank the reviewer for this suggestion, but we are only including variables that are affected by the biosphere, it is true that elevation has a strong effect on the biosphere, the biosphere has no impact on elevation (excluding long term effects, such as erosion).

**3.5  General structure**

The paper generally could be improved by curating more the structure. Several points on this:

- Section 3.2 could benefit from some introduction naming what you intend to calculate first (get trends, test significativity, get breakpoints, hysteresis) before going in the details. This part could also be more pedagogic, providing more rational on why you do these things.

- Parts of the 'discussion' should be much further after the 'results', such as lines 155-162 which should come in some kind of 'caveats and perspective about the method' section

- Section 3.2 is very unbalanced with respect to 3.1. Probably best to reorganize to avoid 'sub-sub-sections' and have subsections from 3.1 to 3.5

- Parts describing concepts, such as Hysteresis (lines 235-246) should not appear in the results but before, either in methods or introduction.

**3.5.1   Authors' response:**

We thank the reviewer for these suggestions and hope that we have addressed satisfactorily.

**3.5.2   L 205: Added introduction to section 3.2**

Because we exchange the variables for (fewer) components we can do the same kinds of analysis on the components as we usually do on variables. In the following apply some of the commonly used methods of analysis on the components to analyze their general properties. First we calculate the mean seasonal cycle to analyze the general behavior of the components, we see if we can find hysteresis effects and explore their origins, we calculate anomalies to find extreme events. We analyze single trajectories to find non-obvious changes, and apply a number of change detection algorithms, i.e. trend detection, breakpoint detection and trends in amplitudes.

**3.5.3  L183:**

The mentioned paragraph is too short for an entire section, we moved it further back, after a paragraph comparing PC1 and PC3, where we observe that PC3 could probably be avoided by using a nonlinear method.

**3.5.4  Reordering Section 3.2**

Section 3.2 is about trajectories and therefore we thought it would be useful to keep these in a shared hierarchy level.

**3.5.5  Move the description of Hysteresis into Methods**

Moved the description of Hysteresis into the "Methods" section, see comment of previous reviewer.

**3.6  Minor stuff**

**3.6.1  Lines 74, 75:**

how do you manage intermittent gaps in the data? Does this affect your averages and your normalization? Also, please clarify if the normalization is based on the entire data cube for each variable, or is the normalization done per time frame?

1. Authors' response: We should mentionthis, this was partially already addressed by responses to previous reviewers.

2. L75, added: Spatiotemporal pixels with missing values were ignored in the calculation of the covariance matrix.

**3.6.2 Line 182:**

don't you mean sensible heat instead of latent heat?

1. Authors' response: Yes, thank you for noting this, changed.

**3.6.3 Figure 1:**

caption could be more instructive, perhaps somehow say there what the reader should understand/read from the "rotation matrix".

1. Authors' response: Thank you for pointing out that the term rotation matrix may not be understood by everyone. We have added the word "loadings", which is the standard jargon for PCA and an explanatory sentence.

2. Added the following sentence: The columns of the rotation matrix describe the linear combinations of the (centered and standardized) original variables that make up the principal components.

**3.6.4 Figure 7:**

surprised to see the strong pattern in Eastern Australia. Is this corroborated in other studies?

1. Authors' reply This is indeed interesting, we added a paragraph describing the reasons for this particular trend:

2. L328 Added: In eastern Australia we find a strong wetness and greenness trend which is due to Australia having a "milennium drought" since the mid nineties with a peak in 2002 (**??**) and extreme floods in 2010–2011 (**?**).

3.6.5   Mention the time period for trend analysis.

Regarding all trend analyses, make sure you more clearly mention in the captions the extend of the period you are considering, as these are not long-term trends and could thus be misinterpreted.

1. Authors' reply: Good point, added the year to the captions of fig. E1 and 8

3.6.6   Add contour for coast lines

For clarity and readability, figures with maps could benefit from either a dark background on the oceans or a line vector showing the coasts, as many of the colour scales use very light colours which are confounded with the white background.

1. Authors' reply: Done, improved the figures quite a bit, thanks for the suggestion!

3.6.7   Move breakpoint detection to SI, including description

I wonder if the breakpoint detection is really useful if it is not more mentioned and elaborated in the main text and just left in appendix. I would recommend to bring it in as a main figure if something strong can be extracted from there, and otherwise remove it entirely from the methods. Eventually you could include it in supplementary, but then include the description of the breakpoint methodology only there.

1. Authors' reply The breakpoint detection is an example analysis that showcases one of the possible set of changes that can occur and that can be detected, therefore we think it has it's place in the paper as an example what can be possible, without going into too much detail.

**3.6.8 Move Fig C1 into the main text**

On the other hand, I would strongly recommend to integrate the Figure C1 in the main text as you do talk in detail about the Bowen ratio and how the 2 PCs do characterize it well.

1. Authors' reply Thank you for the good suggestion, we have moved the figure into the main text.

**3.6.9 Unify scale ranges for fig D1**

Figure 1D I have a bit of a hard time to make good use of it as it is. Are the values in normalized units or absolute values? Would it not be prefereable to have the same scale for MSC min and MSC max? Do you refer to this figure in the main text.

1. Authors' reply Thank you for this suggestion, but this figure is entirely about showing, that very different ecosystems can be very similar at certain points in time, for this, we don't need to compare across subfigures and therefore a single scale won't help for this, they will just remove contrast, especially across MSC min and MSC max.

**3.6.10 Typos**

There are some typos in several places. Make sure to address them.

1. Authors' reply We have fixed many and hope we did not forget any.

**4 Anonymous Referee #4**

The authors thank the reviewer for the his time and thorough comments, we think that the comments greatly improved the manuscript. We have addressed them below.

**4.1 General remarks**

The authors present a well-written manuscript on the analysis of two principal components derived from a set of biosphere variables, one related to vegetation productivity and the other one related to water stress. The trajectories of those components over time reveal interesting seasonal patterns, inter-annual changes and anomalies, and can be used to track extreme events and state shifts of ecosystems/biomes. Therefore, I believe that this is a novel and relevant contribution to Biogeosciences.

**4.1.1 Authors' reply:**

We thank the reviewer for the positive comment.

**4.2   Major concern**

**4.2.1   Advantage of PCA**

My major concern lies in the fact that the authors select mainly variables related to productivity and water availability, and thus not surprisingly the PCA shows those two major axes. I wonder whether just selecting for example GPP and evaporative stress for the analysis of time trajectories would give the same results, but it might be easier to interpret than principal components representing a mix of variables. Can the authors elaborate in more depth what is the advantage of using PCs in this context?

1. Authors' reply There are multiple advantages,

   - Having to observe less dimensions.
   - Information on the covariance structure of the covariates.
   - If some event happens only on one of the variables constituting a component, then it can still be observed on the final component.
   - Directional information, when observing extremes.

**4.2.2   More data streams**

For describing the state of the terrestrial biosphere, I think the authors are missing a very important component related to biodiversity, habitat quality, intactness, forest degradation and fragmentation. These aspects are crucial to describe the state of the terrestrial biosphere. There is still research needed to develop these as operational data streams, but a few examples are available at least at one point in time, e.g. Global Habitat Heterogeneity from EarthEnv, datasets from Global Forest Watch, Dynamic Habitat Indices DHI from Silvislab. This might not be sufficient (in terms of temporal

resolution) to include it for this analysis, but the results from this study could be compared to those datasets (especially the DHI) and the need and relevance of global biodiversity and habitat intactness/quality information should be discussed.

1. Authors' reply We think that the reviewer has a very relevant point here, we would have loved to include more data streams that are relevant to the biosphere. The major problem is the availability of relevant of open data streams at a sufficiently high resolution in space and time which is currently very limited. As we want to track the change of the indicators over time, including static variables did not really make sense in this analysis. Including variables that have a yearly temporal resolution would require to aggregate our data by year which would also have made for a very interesting analysis but outside of the scope of this study.

**4.3 Minor comments**

**4.3.1 L18:**

new satellite missions, add: Schimel, D., Schneider, F., Bloom, A., Bowman, K., Cawse-Nicholson, K., Elder, C., . . . Zheng, T. (2019). Flux towers in the sky: global ecology from space. New Phytologist, nph.15934. https://doi.org/10.1111/nph.15934

1. Authors' reply: added

**4.3.2 L25:**

green revolution, add: Chen, C., Park, T., Wang, X., Piao, S., Xu, B., Chaturvedi, R. K., . . . Myneni, R. B. (2019). China and India lead in greening of the world through land-use management. Nature Sustainability, 2(2), 122–129. https://doi.org/10.1038/s41893-019-0220-7

1. Authors' reply: added

**4.3.3  L27:**

changes are not only occurring in the onset of spring, but also browning trends, see:

- Garonna, I., de Jong, R., de Wit, A. J. W., Mücher, C. A., Schmid, B., & Schaepman, M. E. (2014).  Strong contribution of autumn phenology to changes in satellite-derived growing season length estimates across Europe (1982 - 2011). Global Change Biology, 20(11), 3457–3470. https://doi.org/10.1111/gcb.12625

- Garonna, I., de Jong, R., & Schaepman, M. E. (2016). Variability and evolution of global land surface phenology over the past three decades (1982-2012). Global Change Biology, 22(4), 1456–1468. https://doi.org/10.1111/gcb.13168

1. Authors' reply: Thanks for the suggestion, we added the browning and changed the line to:

   In general, phenological patterns are changing in the wake of climate change, leading to changes in the growing season (**?**) due changes in the onset of spring (**??**) and autumn (**?**).

**4.3.4  L35:**

if a principal component is a mix of productivity measures, I don't necessarily think it's more intuitive to interpret than a simple GPP map.

1. Authors' reply: Thanks for pointing this out, changed the sentence to:

   The rationale is that dimensionality reduction only retains the main data features, which makes them easier accessible for analysis.

**4.3.5 L63:**

What do you mean by "of parts"? Parts of what?

1. Authors' reply: changed "parts" to "observations"

**4.3.6 L75:**

Isn't this dependent on the coordinate system and/or projection? What is the coordinate system used? And why not try to use an equal-area projection (e.g. equal earth projection)?

1. Authors' reply: Added the coordinate system, thank you for pointing out this oversight.

2. L 72, changed to: The data streams are harmonized as analysis ready data on a common spatiotemporal grid (equirectangular 0.25° in space and 8 days in time, 2001–2011), forming a 4d hypercube, which we call a {data cube}.

**4.3.7 L152:**

So what is contributing to the third component. It's still 9% of explained variance!

1. Authors' reply: Thank you for pointing this out, we have added the third component to the manuscript.

**4.3.8 L162:**

Figure 1b is not very intuitive to me. What exactly does it show and how do you read from this that the first component represents productivity and the second hydrology? The figure doesn't seem to show any clear patterns to me. Could you also show the biplots of PC1 and 2, and PC2 and 3?

1. Authors' reply: As biplots are the "standard" way do describe this type of information, we have thought about adding biplots, but decided against it for the following reasons: 1) Biplots don't really contain any information that is not already contained in fig 1b and fig. 2. 2) The number of observations is so high, that it would be impossible to add all the observations to a plot, we worked our way around this by showing bivariate histograms as a background shading in fig. 2. 3) The manuscript contains too many figures already.

**4.3.9 L177/178:**

check spelling

1. Authors' reply: Thanks for finding this!

2. Changed to: While surface moisture is a rather direct measure, evaporative stress is a modeled quantity summarizing the level of plant stress: A value of zero means that there is no water available for transpiration, while a value of one means that transpiration equals the potential transpiration (**?**).

**4.3.10 Figure 2:**

Very interesting figure! A degraded or stressed system might show different trajectories, could you somehow visualize the difference between intact and degraded ecosystems?

1. Authors' reply: Thank you for the positive comment, in this figure we are trying to show trajectories that are diverse. You can see a comparison between a degraded and non-degraded trajectory in fig. 6a.

**4.3.11 L258:**

check spelling

1. Authors' reply: Thanks for finding this one. This sentence was changed in reply to another comment.

**4.3.12 Figure 5:**

third line, the effects of the drought

1. Authors' reply: Changed drought -> floods. Thank you for finding this mistake.

**4.3.13 Figure 6:**

This figure is a bit confusing to me. Could you improve the legends? I don't see an increase in seasonal amplitude in 6a, but maybe I just don't read this figure correctly.

(b-c-d) seem to show the mean seasonal cycle and an event, but what do we see in 6a?

1. Authors' reply: Thank you for pointing out that this may be confusing, we have added an explanatory sentence to the caption.

2. Added: The red line shows the trajectory before 2003, the blue line the trajectory 2003 and later, a strong increase in seasonal amplitude can be observed after 2003.

**4.3.14 L305:**

changes that occurring?

1. Authors' reply: Thank you for finding this, this sentence was changed in reply to another comment.

**4.3.15 L340:**

Additional research is needed to better represent biodiversity, habitat quality and intactness, forest degradation and fragmentation, etc. . . See:

- Jetz, W., Cavender-Bares, J., Pavlick, R., Schimel, D., Davis, F. W., Asner, G. P., . . . Ustin, S. L. (2016). Monitoring plant functional diversity from space. Nature Plants, 2(3), 16024. https://doi.org/10.1038/nplants.2016.24

- Chiarucci, A., & Piovesan, G. (2019). Need for a global map of forest naturalness for a sustainable future. Conservation Biology, 00(0), cobi.13408. https://doi.org/10.1111/cobi.13408

- Nicholas C. Coops, Michael A. Wulder, (2019). Breaking the Habit(at), Trends in Ecology & Evolution, Volume 34, Issue 7, https://doi.org/10.1016/j.tree.2019.04.013.

1. Authors' reply: We think that the reviewer has a very valid point here, it would be very desirable to include these variables into the analysis. Unfortunately these variables do not exist, yet

2. L356, changed to: Future research should consider nonlinearities, adding data streams that represent different aspects (e.g. biodiversity, and habitat quality), and work to include different subsystems, such as the atmosphere or the anthroposphere.

3. L201, added: Because the number of available data streams that describe the biosphere globally in a sufficiently high resolution in space and time is limited, the resulting components reflect the dimensions contained in these data streams. The used data streams mostly describe the exchange of energy and matter of ecosystem with the atmosphere. Data streams that describe the biology more closely, such as habitat fragmentation (**??**), diversity (**??**), and ecosystem intactness (**?**) have great potential to be included into analyses like the present but still require substantial amounts of research.

4.3.16   L352:

detected ina a similar fashion

1. Authors' reply: Thanks for finding this one.

**5 List of other changes**

**5.0.17 L8:**

1. Reason: PC 3

2. From: We find that two indicators account for 73% of the variance of the state of the biosphere in space and time.

3. To: We find that three indicators account for 82% of the variance of the state of the biosphere in space and time

**5.0.18 L11:**

1. Reason: PC 3

2. Added Sentence: The third indicator represents mostly changes in albedo

**5.0.19 L31:**

1. Reason: Typo.

2. From: Do we need to develop specific solutions for every observed phenomenon or can we develop a single approach to uncover a wide variety of phenomena.

3. To: Do we need to develop specific solutions for every observed phenomenon or can we develop a single approach to uncover a wide variety of phenomena?

**5.0.20 L58:**

1. Added: These indicators should also be uncorrelated, so that one can study the system state by looking and interpreting each indicator independently.

**5.0.21 L151:**

1. Reason: PC 3

2. From: We see that the first two components explain 73% of the variance from the 12 variables; additional components contribute little $< 10$ variance each. This results in a "knee" at component 2, which suggests that two indicators are sufficient to capture the major global. We see that the first three components explain 82% of the variance from the 12 variables; additional components contribute little $< 7$ variance each. This results in a "knee" at component 3, which suggests that three indicators are sufficient to capture the major g

**5.0.22 Figure 1:**

1. Reason: PC 3

2. Change: Flipped component 3 so that it is positively correlated with albedo.

**5.0.23 Caption Figure 1:**

1. Reason: PC 3; Changes Figure 1.

2. From: (a) Fraction of explained variance of the PCA by component. Components three and higher do not conrtibute much to total variance. (b) Rotation matrix of the global PCA model, axis one describes primary productivity related variables, axis two describe water availability.

3. To: (a) Fraction of explained variance of the PCA by component. Knee at component three suggest that components four and higher do not contribute much to total variance. (b) Rotation matrix of the global PCA model (also called *loadings*, eq. **??**). The columns of the rotation matrix describe the linear combinations of the (centered and standardized) original variables that make up the principal components. PC1 is dominated by primary productivity related variables, PC2 two by variables describing water availability, PC3 by variables describing albedo. }

**5.0.24 L183:**

1. Reason: PC 3

2. Added: We observe that the third axis is most strongly related to albedo (fig. **??**b). Albedo describes the overall reflectiveness of a surface. Light surfaces, such as snow and sand, reflect most of the incoming radiation, while surfaces that have a high liquid water content or active vegetation absorb most of the incoming radiation. Local changes to albedo can be caused by a large array of reasons, e.g. snow fall, vegetation greening/browning, autumn leaf shedding or land use change.

   The third axis can be seen as an axis that introduces the binary decision of snow cover into the model and should be used mostly as such. On a global scale, effects on PC 3 are dominated by snow cover because they represent the highest absolute change in albedo. The relation to productivity and hydrology are counterintuitive to what we would expect from an albedo axis. In fig. **??**b we can also observe that albedo also plays significant roles on PC 1 and PC 2. Because vegetation uses radiation as an energy source, albedo is negatively correlated with the productivity of vegetation, hence the negative correlation of albedo with PC 1. Albedo is also negatively correlated with PC 2, because surfaces with a higher water content absorb more radiation. We can observe that PC 1 and PC 2 are

positively correlated with PC 3 on the positive portion of their axes (see fig. **??**d and f), which means counterintuitively that the index representing albedo is positively correlated with primary productivity and moisture content. Finally we can observe that PC 1 and PC 2 have a much higher reconstruction error in snow covered regions, which is strongly improved by adding PC 3 (see fig. **??**f). This could probably have been avoided by using a nonlinear method to better compress the nonlinear relation between PC 1 and PC 3 (fig. **??**c and d). Therefore the third axis should be regarded mostly as binary variable that introduces snow cover, the other information that is usually associated with albedo is already contained in the first two axes.

**5.0.25 Figure 2:**

1. Reason: PC 3

2. Change: Added subfigures to show trajectories in all combinations of PC 1–3

**5.0.26 Caption Figure 2:**

1. Reason: PC 3; Changes to figure 2.

2. From: Trajectories of some points (colored lines) and the area weighted density over principal components one and two (the gray background shading shows the density) for (a) the raw trajectories and (b) the mean seasonal cycle. The trajectories were chosen to fill a large area in the space of the first two principal components. Some of the trajectories in (b) have an arrow indicating the direction. The numbers illustrate the value of some variables, for units, see tab. **??**. Description of the points: Red: Tropical Rainforest, 67.625°W, 2.625°S; Blue: Maritime

climate, 7.375 °E, 52.375 °N; Green: Monsoon climate, 82.375 °E, 22.375 °N; Purple: Subtropical, 117.625 °W, 34.875 °N; Orange: Continental climate, 44.875 °E, 52.375 °N; Yellow: Arctic climate, 119.875 °E, 72.375 °N;

3. To: Trajectories of some points (colored lines) and the area weighted density over principal components one and two (the gray background shading shows the density) for (left column) the raw trajectories and (right column) the mean seasonal cycle. The trajectories are shown in the space of PC 1–PC 2 (first row), PC 1–PC 3 (second row), and PC 2–PC 3 (third row). The trajectories were chosen to fill a large area in the space of the first two principal components. Some of the trajectories have an arrow indicating the direction. The numbers illustrate the value of some variables, for units, see tab. **??**. Description of the points: Red: Tropical Rainforest, 67.625 °W, 2.625 °S; Blue: Maritime climate, 7.375 °E, 52.375 °N; Green: Monsoon climate, 82.375 °E, 22.375 °N; Purple: Subtropical, 117.625 °W, 34.875 °N; Orange: Continental climate, 44.875 °E, 52.375 °N; Yellow: Arctic climate, 119.875 °E, 72.375 °N;

**5.0.27   L201:**

1. Reason: Typos

2. From: We can see that the bowen ratio embedds well into the space spanned by the first two PCs.

3. To: We can see that the Bowen ratio embeds well into the space spanned by the first two PCs.

**5.0.28   L210:**

1. Reason: Fix references

2. From: The principal components may be used to summarize the movement of a spatiotemporal pixel in variable space, so that they represent the current state of the ecosystem at a certain location in space and time (fig. **??**a) or time of year of the mean seasonal cycle of the pixel (fig. **??**b).

3. To: The principal components may be used to summarize the movement of a spatiotemporal pixel in variable space, so that they represent the current state of the ecosystem at a certain location in space and time (fig. **??** left column) or time of year of the mean seasonal cycle of the pixel (fig. **??** right column).

5.0.29   L219:

1. Reason: PC 3

2. Added: The third components shows a different picture. Due to a consistent winter snow cover in higher latitudes the albedo is much higher and the amplitude of the mean seasonal cycle is much larger than in other ecosystems. Other areas show comparatively little variance on the third axis and their relation to productivity and moisture content is even positively correlated to the third axis, which is the opposite of what is expected from an albedo axis.

5.0.30   Figure 3:

1. Reason: PC 3

2. Change: Added the MSC of component 3.

5.0.31   Caption Figure 3:

1. Reason: PC 3

2. From: Mean seasonal cycle of the first principal component during the year. Left column: first principal component. Right column: second Principal Component. Rows from top to bottom: equally spaced intervals during the year.

3. To: Mean seasonal cycle of the first principal component during the year. Left column: first principal component. Middle column: second principal component. Right column: third principal component. Rows from top to bottom: equally spaced intervals during the year.

**5.0.32   L228:**

1. Reason: PC 3; changes to Figure 3.

2. From: The second principal component (fig. **??**, right column) tracks water deficiency:

3. To: The second principal component (fig. **??**, middle column) tracks water deficiency:

**5.0.33   Caption Figure C1:**

1. Reason: Typo

2. From: (c) $\log_{10}\left(\frac{Latent Heat}{Sensible Heat}\right)$, the $\log_{10}$ of the Bowen Ratio.

3. To: (c) $\log_{10}\left(\frac{Sensible Heat}{Latent Heat}\right)$, the $\log_{10}$ of the Bowen Ratio.

**5.0.34   L232:**

1. Reason: PC 3

2. Added: The third principal compontent (fig. **??**, right column) tracks surface reflectance. Therefore we can see the highest values in the arctic region during winter, other areas vary much less in their reflectance throughout the year. Again, the third component shows a counterintuitive behavior in midlatitudes, as it is positively correlated with productivity and therefore shows the oposite behaviour of what would be expected from an indicator tracking albedo.

**5.0.35 Figure 4:**

1. Reason: PC 3. Error in calculation

2. Change: Added hysteresis for all combination of PC 1–3. There was an error in the calculation of the previous version.

**5.0.36 Caption Figure 4:**

1. Reason: PC 3; Changes to Figure 4.

2. From:

3. To:

**5.0.37 L235:**

1. Reason: Rewrite of the hysteresis section due to adding more combination of indicators and a calculation error in the original version of the manuscript

[revised manuscript text omitted]

**5.0.38  Figure 5:**

1. Reason: PC 3

2. Change: added panes for PC 3

**5.0.39  Caption Figure 5:**

1. Reason: PC 3, changes to Figure 5

2. From: Anomalies of the first three principal components; Brown-green contrast shows the anomalies on PC1, a relative low productivity or greening respectively. Blue-green contrast shows the anomalies on PC2, a relative wetness or dryness respectively. (a) Map showing the PC1 anomalies on the 1/1/2001. (b) and (c) show longitudinal cuts of PC1 and PC2 at the red vertical line in sub-figure (a) respectively. The effects of of the drought on the Horn of Africa (2006) and the Russian heatwave (2010) are highlighted by circles. (d) Map showing the PC2 anomalies on the 1/1/2001. (e) and (f) show longitudinal cuts of PC1 and PC2 at the red vertical line in sub-figure (d) respectively. Strong droughts in the Amazon
during 2005 and 2010 can be observed as large red spots on the fringes of the Amazon basin (highlighted by circles).

3. To: Anomalies of the first two principal components; Brown–green contrast shows the anomalies on PC1, a relative low productivity or greening respectively. Blue–red contrast shows the anomalies on PC2, a relative wetness or dryness respectively. Brown–purple contrast shows the anomaly on PC3, a relative deviation in albedo. (a), (e), and (i) are map showing the anomalies of PC1–3 on the 1/1/2001 respectively. (b), (c), and (d) show longitudinal cuts of PC1–3 at the red vertical line in sub-figure (a) respectively. The effects of of the drought on the Horn of Africa (2006) and the Russian heatwave (2010) are highlighted by circles. (f), (g), and (h) show longitudinal cuts of PC1–3 at the red vertical line in sub-figure (e) respectively. Strong droughts in the Amazon during 2005 and 2010 can be observed as large red spots on the fringes of the Amazon basin (highlighted by circles). (j), (k), and (l) show longitudinal cuts of PC1–3 at the red vertical line in sub-figure (i) respectively. A strong snowfall event affecting Central and Southern China is marked in circles.

**5.0.40 L257:**

1. Reason: Language

2. From: These anomalies have a directional component and can be therefore be interpreted the same way as the original PCs which contain information of the underlying variables that were affected. In this sense, one can infer the state of the ecosystem during an anomalous state.

3. To: These anomalies have a directional component and can be therefore be interpreted the same way as the original PCs which contain information of the

underlying variables that were affected, therefore one can infer the state of the ecosystem during an anomaly.

**5.0.41 L272:**

1. Reason: PC 3

2. Added: Another extreme event that can be seen is the extreme snow and cold event affecting Central and South China in January 2008, causing the temporary displacement of 1.7 million people and economic losses of approximately US $ 21 billion (**?**). This event shows up clearly on PC2 and PC3 as cold and light anomalies respectively (see fig. **??**k and f).

**5.0.42 Figure 7:**

1. Reason: PC 3

2. Change: Added PC3 trend. Split up the bivariate color map. Added Bivariate trends for all combinations of PC1–3

**5.0.43 Caption Figure 7:**

1. Reason: Changes Figure 7

2. From: Trends in PC1 and PC2 indicators. Trends were calculated using the Theil-Sen estimator. (a) The spatial distribution of slopes, only significant slopes are shown ($p < 0.05$, Benjamini-Hochberg adjusted). The maximum cutoff for the legend limits was set symmetrically around zero to the maximum absolute value of the 0.1 and 0.9 quantiles. (b) Distribution of spatial points in the space of the first two PCs. The colors correspond to the ones used in (a).

3. To: (a), (c), (e) Trends in PC1–3 respectively. (b), (d), (f) Bivariate distribution of trends. Trends were calculated using the Theil-Sen estimator, (a), (c), and (e) show significant trends only ($p < 0.05$, Benjamini–Hochberg adjusted).

5.0.44 L308:

1. Reason: large -> larger, is more correct

2. From: The accumulation of CO2 in the atmosphere should cause an increase in global productivity of plants due to CO2 fertilization, while large and more frequent droughts and other extremes may counteract this trend.

3. To: The accumulation of CO2 in the atmosphere should cause an increase in global productivity of plants due to CO2 fertilization, while larger and more frequent droughts and other extremes may counteract this trend.

5.0.45 L313:

1. Reason: PC 3

[revised manuscript text omitted]

**5.0.60 L183:**

1. Reason: Referencing the figures and results in the appendix (reconstruction error)

2. Added: On a global scale, effects on PC3 are dominated by snow cover because they represent the highest absolute change in albedo, this can also be seen from the reconstruction error that increases strongly towards the poles for the first two principal components but evens out if the third component is added (see fig. **??**).

**5.0.61 L146:**

1. Reason: Add example figure for areas of polygon.

2. From: If the vertices run clockwise, the area is negative. If the polygon is shaped as an 8, the clockwise and counterclockwise parts will cancel each other (partially) out, e.g. the green trajectory in fig. **??**b. Trajectories that cover a larger range will also tend to have larger areas.

3. To: If the vertices run clockwise, the area is negative. If the polygon is shaped as an 8, the clockwise and counterclockwise parts will cancel each other (partially) out. Trajectories that cover a larger range will also tend to have larger areas. For some example polygons, see fig. **??**.

**5.0.62 L142:**

1. Reason: New Figure 1 and caption.

2. Added: Example polygons and their areas, $A$ (Eq. **??**), the arrows indicate the directionality. (a) Clockwise polygon, has a negative area. (b) Counterclockwise

polygon, has a positive area. (c) Chaotic polygon, has a very low area. (d) Polygon with a single intersection, has both a clockwise and counterclockwise portion. The clockwise portion is slightly larger than the counterclockwise portion, therefore the area is slightly negative.

**5.0.63 Appendix F:**

1. Reason: Add data sources

2. From: **Black Sky Albedo** is the reflected fraction of total incoming radiation under direct hermispherical reflectance, i.e. direct illumination (**?**).

   **White Sky Albedo** is the reflected fraction of total incoming radiation under bi-hemispherical reflectance, i.e. diffuse illumination (**?**). Together with black sky albedo it can be used to estimate the albedo under different illumination conditions.

   **Evaporation** $[\mathrm{mm/day}]$ is the amount of water evaporated per day (**?**), depends on the amount of available water and energy.

   **Evaporative Stress** modeled water stress for plants, zero means that the vegetation has no water available for transpiration and one means that transpiration equals potential transpiration (**?**).

   **fAPAR** the fraction of absorbed photosynthetically active radiation, a proxy for plant productivity (**?**).

   **Gross Primary Productivity (GPP)** $\{[\mathrm{gC\ m^{-2}\ day^{-1}}]\}$ the total amount of carbon fixed by photosynthesis (**?**).

   **Terrestrial Ecosystem Respiration (TER)** $\{[\mathrm{gC\ m^{-2}\ day^{-1}}]\}$ the total amount of carbon respired by the ecosystem, includes autotrophic and heterotrophic respiration (**?**).

**Net Ecosystem Exchange (NEE)** {[gC m$^{-2}$ day$^{-1}$]} The total exchange of carbon of the ecosystem with the atmosphere $NEE = GPP - TER$ (**?**).

**Latent energy (LE)** {[W m$^{-2}$]} the amount of energy lost by the surface due to evaporation (**?**).

**Sensible Heat (H)** {[W m$^{-2}$]} the amount of energy lost by the surface due to radiation (**?**).

**Root-Zone Soil Moisture** {[m$^3$ m$^{-3}$]} the moisture content of the root zone, estimated by the GLEAM model (**?**).

**Surface Soil Moisture** {[mm$^3$ mm$^{-3}$]} the soil moisture content at the soil surface (**?**).

3. To: **Black Sky Albedo** is the reflected fraction of total incoming radiation under direct hermispherical reflectance, i.e. direct illumination (**?**). This dataset is derived from the SPOT4-VEGETATION, SPOT5-VEGETATION2, and the MERIS satellite sensors.

**White Sky Albedo** is the reflected fraction of total incoming radiation under bihemispherical reflectance, i.e. diffuse illumination (**?**). Together with black sky albedo it can be used to estimate the albedo under different illumination conditions. This dataset is derived from the SPOT4-VEGETATION, SPOT5-VEGETATION2, and the MERIS satellite sensors.

**Evaporation** $[\text{mm/day}]$ is the amount of water evaporated per day, depends on the amount of available water and energy. This dataset is based on the GLEAMv3 model (**?**), using satellite data from ESA CCI and SMOS to derive a number of variables.

**Evaporative Stress** modeled water stress for plants, zero means that the vegetation has no water available for transpiration and one means that transpiration equals potential transpiration. This dataset is based on the GLEAMv3 model (**?**), using satellite data from ESA CCI and SMOS to derive a number of variables.
**fAPAR** the fraction of absorbed photosynthetically active radiation, a proxy for plant productivity (**?**). This dataset is based on the GlobAlbedo dataset (http://globalbedo.org) and the MODIS fAPAR and LAI products.

**Gross Primary Productivity (GPP)** {[gC m$^{-2}$ day$^{-1}$]} the total amount of carbon fixed by photosynthesis (**?**). This dataset is derived from upscaling eddy covariance tower observations to a global scale using machine learning methods.

**Terrestrial Ecosystem Respiration (TER)** {[gC m$^{-2}$ day$^{-1}$]} the total amount of carbon respired by the ecosystem, includes autotrophic and heterotropic respiration (**?**). This dataset is derived from upscaling eddy covariance tower observations to a global scale using machine learning methods.

**Net Ecosystem Exchange (NEE)** {[gC m$^{-2}$ day$^{-1}$]} The total exchange of carbon of the ecosystem with the atmosphere $NEE = GPP - TER$ (**?**). This dataset is derived from upscaling eddy covariance tower observations to a global scale using machine learning methods.

**Latent energy (LE)** {[W m$^{-2}$]} the amount of energy lost by the surface due to evaporation (**?**). This dataset is derived from upscaling eddy covariance tower observations to a global scale using machine learning methods.

**Sensible Heat (H)** {[W m$^{-2}$]} the amount of energy lost by the surface due to radiation (**?**). This dataset is derived from upscaling eddy covariance tower observations to a global scale using machine learning methods.

**Root-Zone Soil Moisture** {[m$^3$ m$^{-3}$]} the moisture content of the root zone. This dataset is based on the GLEAMv3 model (**?**), using satellite data from ESA CCI and SMOS to derive a number of variables.

**Surface Soil Moisture** {[mm$^3$ mm$^{-3}$]} the soil moisture content at the soil surface. This dataset is based on the GLEAMv3 model (**?**), using satellite data from ESA CCI and SMOS to derive a number of variables.

5.0.64   Whole document:

1. Reason: Typography

2. From: PCx

3. To: PCx

5.0.65   Whole document:

1. Typography

   (a) From: data set(s)
   (b) To: dataset(s)

2. Affiliation

   (a) Reason: Added the affiliation to the iDiv of the lead author due to a recent change in employment.
   (b) What: Added affiliation to the iDiv of the lead author.

---

## Author Response (AR1)

**Replies to the Anonymous Referees**

Guido Kraemer Gustau Camps-Valls Markus Reichstein Miguel D. Mahecha

January 20, 2020

We thank the editors and anonymous reviewers for their input and devoted time to our work. The detailed revisions were very helpful and have allowed us to improve the manuscript greatly. The major changes in the new version of the manuscript include:

- Adding results and analysis of the principal component 3 to the manuscript.
- A revised introduction picking up all research questions that appear later in the manuscript.
- A better explanation of what differentiates the present study from other studies using PCA.
- An expanded "Hysteresis" section, with an improved and more intuitive explanation.
- Move some of the results from the appendix into the main text.
- Many other revisions.

We have addressed all concerns in detail below, reviewers' comments are in normal font, replies are in *italics and blue*. Additionally, many small improvements have been made to the manuscript and many errors have been corrected. For a full list of changes, please see the attached document at the end of this document. We hope that with all these changes the manuscript meets the high quality standards of the journal.

Kind regards,

The Authors

**1 Anonymous Referee #1**

**1.1 General Remarks**

I appreciated reading the discussion paper Summarizing the state of the terrestrial biosphere in few dimensions by Guido Kraemer and colleagues. The paper presents an approach for summarizing key variables on the terrestrial biosphere into fewer independent components using established multi-variate methods. They exemplify their approach by showing several trajectories across space and time and by highlighting some major anomalies visible in their data. While the work is well presented and scientifically sound, I have some major concerns regarding the publication of the manuscript in its current form.

We thank the reviewer for his positive and very thorough review and the helpful comments. We have now addressed the open issues as we will show below. We especially thank the reviewer for the detailed review of the overall structure of the manuscript and pointing out the many small details that have been improved in the new version of the manuscript.

**1.2 Concerns**

**1) The number of dimensions**

The authors state that the first two components explain large parts of the variance and that the 'knee' is reached with the second component. However, inspecting Figure 1a, it seems that the 'knee' is reached with the third component, which still explains 9% of the variance. I was a little confused that the third component was disregarded throughout the whole manuscript, without giving a strong justification. Figure 2b indicates that the third component might be strongly connected to albedo. I encourage the authors to either expand their analysis to also include the third component, or to give a very strong argument for its exclusion. As it stands now, the decision to only inspect the first two components is very subjective.

The reviewer is right that the 3rd principal component still contains important information. Therefore, we included results and analysis of component 3 in the manuscript. Accordingly, we have implemented the following changes:

- Added axis 3 to the manuscript (for details, see the attached document showing all changes).
- Flipped axis 3 so that higher values for PC3 mean higher albedo.

We think that the addition of the third component improved the manuscript substantially and want to thank the reviewer for recommending this.

**2) Scientific novelty and usefulness**

I am missing a strong discussion/conclusion on how the manuscript advances scientific progress. Putting it into simple terms, the authors apply PCA – a widely used and established method – to a set of existing data sets. As such, it is not really a novel methodological development, but rather a demonstration of what could be done with global datasets as provided though the Earth System Data Lab. While this is not a deal-breaker per sé, the authors could greatly advance their manuscript by explaining how this approach can be used by other scientists, that is how it will advance the science of the terrestrial biosphere.

Thank you for this critique and comment which can be viewed from several angles. At first glance, the reviewer is right: we simply applied a PCA to a highly curated global data set—a data cube contained in the Earth system data lab. But, although the method is similar to EOFs in climatology, where the matricization (the flattening of the 4th order tensor, variables × time × longitude × latitude, to a matrix) happens maintaining time. We are maintaining both, space and time and reduce only over the variables, as far as the authors are aware, this has not been done on global data.

This is in our view an innovation, as we account, for the first time, for the many redundancies in high-dimensional Earth observations. We have carefully reviewed the literature, but do not find a study that has investigated the global covariations of multiple Earth observation data streams. This is the main novelty of our work, to better explain our approach in more depth, we have added some explanations to better explain these differences to the reader.

Also, the use of a simple PCA algorithm is not incidental here: we seek for a method that learns a data transformation that is invertible, and allows us to measure/compute the reconstruction error in meaningful physical units. This cannot be done with more complicated/sophisticated nonlinear machine learning methods, where the (probably more accurate) transform is hard to analyze.

**3) Too many results in the appendix**

Many of the results are buried in the Appendix but never picked-up in the main text. In fact, Figure A1, B1, D1 and C1 were never referenced in the main text. The authors thus present many results in the Appendix that

are not discussed in the main manuscript and thus the reader is left alone with her own interpretation. As some of the results are quite crucial for evaluating the method (e.g., the errors presented in B1), I strongly encourage the authors to thoroughly discuss them in their manuscript.

Thank you for the observation! We fully agree that we have a lot of results in the appendix, some of them very relevant for the discussion. Following the suggestion, we moved parts of the appendix into the main text and have also added references to the figures into the text:

- Moved the section "Reconstruction Error" from the appendix to the text.
- Moved figure C1 ("Bowen Ratio") into the text.
- We added the corresponding references into the text.

**4) Writing**

The writing needs improvement for turning this already good manuscript into an excellent manuscript. For example, the authors often describe their figures, instead of the results (Figure X shows...). It would be much more interesting to read about the main result instead (A influences B (Figure X)). I am sure the senior authors of this manuscript can do a great job in revising the manuscript to make it more accessible and exciting for the reader.

We thank the reviewer for the pointing this out and have revised many aspects of the paper, see the marked up manuscript version showing all changes. We hope that we have corrected the manuscript accordingly.

**5) Spelling/grammar**

There are some wording and spelling/grammar issues, some of which listened below:

Thanks for the thorough revision provided. We have corrected all suggested minor changes, and commented further on the critical ones below.

L. 16: Suggest removing 'the' before 'global'.

Thank you, changed.

L. 27: Spring is not a phenological event. Could use onset of bud-flush or similar.

Thanks for catching this detail.

L. 74: Not clear how standardization accounts for differences in scales. What scales? Spatial? Temporal?

Indeed, the wording is a bit ambiguous. We have changed it to make clear that we mean scale in a statistical sense here.

L. 138: The breakpoint detection comes out of the blue. Why is this done? What was the rational behind? This needs a decent introduction.

The reviewer is right that we do breakpoint detection without properly introducing it, we mentioning the topic in the introduction now.

L. 142: Same as above. The term hysteresis is never introduced before, but then explained in the results section (L. 239). As a reader, I would love to hear the details upfront, instead of reading about them in the results/discussion.

This was missing from the introduction, we thank the reviewer for noticing this, we have remedied the situation.

L. 148: Maybe include an example figure here, instead of referencing to the results already.

The hysteresis may be a complex topic for people not familiar with it, we thank the reviewer for pointing this out and have added a conceptual figure to the "Methods" section that hopefully makes the concept easier to understand.

L. 151: 'We see that...' is not a good opener. Directly describe the result, be precise and upfront (e.g., The first two components explained 73% of the variance (Figure 1a))

Removed "We see that".

L. 160: What is the pre-imaging problem? Please do not assume that the reader reads up the details in the reference provided. Either avoid naming it or give a brief description.

Again we thank the reviewer for pointing out that this is a topic that the target audience may not be acquainted to. We have improved the description and hopefully made the concept understandable to everyone.

L. 162: Again, not the best opener. The first sentence of a paragraph should summarize the main point of the paragraph (topic sentence), allowing

the reader to skim through the manuscript. This sentence just describes where the reader can find a result, but nothing about the result itself.

Thanks for pointing this out, we have have removed the sentence.

L. 164: Odd formulation (two times related).

Thank you for noticing! We have changed the sentence accordingly.

L. 174ff: his paragraph actually described the indicators used and does not discuss the results. This could go into the methods description or should be more clearly related to the actual results.

This paragraph describes PC2 and discusses how the variables that make up PC2 are related, therefore we have decided to leave it in, as a discussion of PC2.

Figure 2: What are 'some points'? How were they chosen?

It says so in the caption: "The trajectories were chosen to fill a large area in the space of the first two principal components."

L. 139: As said before, this is rather introduction than results/discussion. I would have very much appreciated reading this in the introduction.

This is the wrong line number, the reviewer is probably referring to the description of the Bowen ratio as this should be mentioned in the introduction, indeed. We have added the Bowen ratio to the introduction and changed the paragraph to highlight the main result.

L. 258: rephrase: ... and can therefore be interpreted...

Thank you for finding this, we have rephrased the sentence.

L. 282: Again, put the result in the spotlight, not the figure showing the result.

The reviewer is right, this also counts for some of the other paragraphs describing that figure, thank you for pointing this out. We hope to have remedied the situation with the changes made to the paragraphs referring to the same figure.

L. 305: Occur instead of occurring.

Changed, thanks.

L. 312: Move 'especially' after 'showed'.

Changed, thank you.

L. 313: Repeats methods.

Thanks for noticing, we have removed the phrase and added "... patterns of trends ... " to the next sentence.

L. 320: Why did you calculate the trends from the full data? Would it have been better to use the growing season as well to facilitate comparison? Please give a reasoning why you do it differently.

The reviewer is right, that usually these kind of analyses are made on the growing season only. Because of simplicity of the analysis, we opted to do the analysis this way, just as with the breakpoints we did not want to develop complicated methods for detecting the growing season from PC1 because this is not the scope of this paper. The analyses on the resulting indicators are simple and straightforward because of their exploratory nature. The next question would have been, how to limit PC2 and PC3? Use the wet/dry season for PC2 because it shows water, and summer/winter for PC3, or also use the growing season? Using growing season data only, we probably could have found stronger trends in PC1, but this could be an interesting topic for future research.

L. 324: Something odd with the sentence starting with 'Inside...'.

Thanks for finding this one, fixed!

L. 327: Remove 'a' before 'browning'.

Removed, thank you.

L. 349: The breakpoints are actually never shown, nor discussed. The conclusion is thus not really based on data here.

The reviewer is right, we have added the breakpoints at several places throughout the manuscript. Thank you for pointing this out.

L. 352: in, not 'ina'.

Changed, thank you.

**2 Anonymous Referee #2**

**2.1 General Remarks**

**2.1.1 General assessment**

This is a very interesting paper addressing some important issues of big data analysis for ecology studies. It is rich in analyses and provides some new views on an old method (PCA). I particularly liked the analysis of trajectories that I found quite powerful, notably for case studies.

The authors thank the reviewer for the positive review, his time and thorough comments. We think that the comments allowed us to greatly improve the manuscript. We have addressed all of the reviewer's concerns as detailed below.

**2.1.2 Key research question**

Yet I found it difficult to understand what key research questions are addressed in this paper. This is important to clarify at the end of the introduction as the authors is providing us with a suit of analyses that may resemble (for non PCA-expert) an attempt of addressing many (all?) questions without real rationale. The readers need to have a clear (concise) view of the objectives of this paper, and they need to be guided through the analyses by referring back to the main research questions.

Thank you for pointing this out. The reviewer is right in that the paper may appear to try to solve too many problems. We have done some major revisions and hope that thate focus of the paper is clearer now. Thanks for the comment.

The main motivation and goal of this paper is a lack of a systematic data-driven approach to explain the main features in Earth system data cubes in the literature. We first introduce a novel way of applying PCA as a method to create such summarizing indicators, then we apply the method to a global set of representative variables describing the biosphere. Finally, to prove the effectiveness of the method, we give interpretations of the resulting set of indicators and explore the information contained in the indicators by analyzing them in different ways and relating them to well known phenomena. We have explicitly declared such motivation and approach at the end of the "Introduction" section.

**2.1.3 Input data may cause the resulting axes**

In addition, I also have a major concern related to the set of inputs data used to feed the PCA. I agree that PCA is a powerful tool to deal with correlated variables, yet I have difficulties understanding why the authors have decided to include variables that are obviously highly correlated. To my opinion, vegetation productivity proxies are overrepresented as well as those related to water availability and stress. It puts some doubts in my head as to whether the finding of PC1 (primary productivity) and PC2 (surface hydrology) driving the state of the biosphere in space and time is truly original (or just purely mathematical). It is therefore important for the authors to justify the set of original variables. A suggestion could also be to decrease the number of input variables (removing obvious redundant proxies) as the amount of data to be condensed is mainly coming from the 8days interval used for the analysis.

PCA extracts uncorrelated components, therefore the resulting axes will not change much if more or less variables are added that represent a certain aspect of the ecosystem. Intuitively, adding correlated variables in the analysis means that geometrically they point in the same direction in the feature space and do not change much the selection of the corresponding principal component. What does change are the explained variances of the resulting axes, i.e. including more variables that are proxies for primary productivity will cause this axis to explain more variance. The set of covariates we chose constitutes a large complementary and representative set that describes the exchange of mass and energy of the biosphere with the atmosphere. We have added a justification for the variables used to the "Data" section.

**2.2 Detailed comments**

Finally I also have other comments and concerns - notably related to the structure of the manuscript - that would need to be addressed by the authors prior publication of their research (see attached report for details).

**(1) Abstract**

The authors start off the abstract by mentioning the importance of detecting abrupt and gradual changes in terrestrial ecosystem but do not develop further in the introduction. In the method section, the detection of breakpoints reappears but no results are presented or discussed (except for the appendix A). The authors should decide whether to consider the detection of abrupt changes as a real research question for this study.

The reviewer is right, do not really go into detail in the analysis of breakpoints. To remedy this, we have changed the first sentence and made it clear that there is a proof of concept analysis in the appendix.

**(2) Introduction**

As stated in my main comment, I find that there is somewhat a mismatch between the introduction and the method section. In the introduction, the authors touch upon many issues related to assessing and attributing changes of biosphere properties. However apart from creating a new set of independent, 'essential' variables, they do not clearly mention what other research questions this study is going to address; whereas in the methods they mention PCA, trend and breakpoints analyses. Clearly stating the research questions for this study would help the readers to understand the rationale behind each analysis.

The reviewer is right, therefore we have also extended the "Introduction" section to contain all research questions.

**(3) Data and methods**

Better description of the data The description of the data slightly too minimalistic, including in the appendix F. Mentioning the input data (satellite, climate or others) feeding into each dataset would be helpful. The observation period used for this study is also not mentioned.

We thank the reviewer for pointing this out. We have added the limits of the time dimension and the type of grid in the "Data" section and have extended the descriptions of the variables in the appendix.

L. 75, Mention projection This statement is not always valid (e.g. in the case of equal-area projection). The sentence would be clearer if the authors would mention the projection system used here.

We thank the reviewer for pointing this out. See previous response.

L77. Better explanation of PCA The authors mentioned that they used a modified PCA, reading from the description given in the following lines, the PCA applied here seems to be standard. Could the authors provide some explanations to why / how the PCA has been modified? It should also clarify whether they applied the PCA in s or t-mode. We have clarified the PCA analysis by discussing it in the context of frameworks describing PCA in the context of climatology and ecology and hope that this will help with the understanding of the method.

- The PCA is a decomposition of the correlation matrix.
- Building the correlation matrix is not standard due to the big data aspects, and the spatial extension, both of which require a lot of care in the calculation of the covariance matrix, which is described in the "Methods" section.
- We summarize the dimensions in a novel way, there are a number of different frameworks (S- vs. T-mode in climatology, Q- vs. Rmode in ecology and multivariate statistics, and primal vs. dual modes in machine learning) that describe standard applications of PCA, none of which give an exact description of the analysis done here. We have added a section ("Relations to other PCA-type analyses") describing the relation of the present analysis with these frameworks.
- **Per-pixel analysis** It would be nice here to make a link to the (extended see comment above) research questions in order to understand directly the rationale for such analyses.

The link was really unclear, we have added more research questions to the introduction (see previous replies) and are now mentioning the research questions.

**(4) Results**

General comment: I highly suggest to split the results and discussion into two separate sections. It will facilitate the reading and will allow the authors to emphasise better the originality of their work. Example: L155-161, L164-173, L175-182, L235-246, etc. should not be in a results section s.s., but would rather belong to a discussion (or even introduction or method). Please consider at least moving all methods description and introduction to new concepts to the respective adequate sections.

We thank the reviewer for this suggestion, but we think that a joint results and discussions section is a better choice, as it allows for the results and their discussions to be closer and easier to follow. L153 and Figure 1 The authors mentioned that there is a knee at component 2. I believe it is rather at component 3. This component still contribute to the total variability to a share of almost 10%, therefore the authors should either include it in the rest of the analysis or provide an adequate justification not to. Also I generally miss a figure presenting together the temporal and the spatial patterns for the main PCs. This could be put as supplementary material. In the caption of Fig.1 I would recommend to change the term axis 1 and 2 by PC1 and 2. The comment also applies to the text itself (Ex. L190).

The third component was missing to simplify the analysis. Looking back at this decision, we fully agree with the reviewer that it should be included for the sake of completeness. We have added it to the paper now, we thank the reviewer for pointing this out, as it improved the manuscript substantially. The spatiotemporal figure was also missing and we have added it, this was an oversight of our part and we corrected it. We have also unified the terminology, axis is now never used to describe principal components.

- We have added the third component to the manuscript.
- Added an appendix with joint time and space patterns.
- Removed the term axis when in designated a component in the entire manuscript.

L183 Please describe in the first sentence what the triangle is made of.

Thank you for noticing. We have provided a better description of the figure.

L203 'movement of a spatiotemporal pixel in variable space', please rephrase. A pixel cannot be moving spatiotemporally, like in a sliding puzzle.

The pixel is moving in the vector space of the principal components, this formulation was easily misunderstood and therefore we have changed it. We thank the reviewer for pointing this out.

L221-224 This should be described in the methods section and should be linked to a key research questions.

We thank the reviewer for pointing out this oversight. We have added a definition of the means seasonal cycle to the methods and link to it in the introduction.

**(5) Conclusion**

L341 The results of the breakpoints analyses were not reported or discussed in the main text, therefore the statement 'To monitor gradual and abrupt changes in times of global change' do not hold.

We thank the reviewer for pointing this out and hope that we have remedied the situation. We have changed the beginning of the conclusion to: "To monitor the complexity of the changes occurring in times of an increasing human impact on the environment ..."

**Appendixes** Some results presented in the appendixes do not appear in the main text, e.g. Figures A1 and B1. The authors should maybe decide on the key results to be presented here and maybe save some others for a follow-up paper?

We thank the reviewer for pointing this out and all appendices should be referenced in the text now.

**(6) Two final comments for reflexion:**

Legacy effects: The authors have applied PCA on time series of 8 day variables without considering any lag or accumulation effect in the response of a given variable. Would it be fair to say that legacy effects might not be captured adequately by such analysis?

The method ignores lag and memory effects, lag effects may still be captured implicitly in the components but there will probably not be a "memory axis". Something like this may be captured using a combination of more advanced machine learning algorithms (e.g. autoencoders and recurrent neural networks) but as far as the authors know, no one ever attempted an analysis like this.

**Operationalization:** The authors refer to the MEI in the introduction as an example of a successful PCA-based indicator. Could the authors elaborate on the requirement for operationalising their methods (e.g. if one would like to use the new indicators operationally, how frequently should the PCA be updated?).

Applying a trained PCA is very simple and computationally efficient. The trained PCA should also be quite stable and therefore we assume that updates do not have to happen frequently. The implementation with WeightedOnlineStats.jl would theoretically allow a very efficient update with every step, but we assume that this will not be necessary. For a real time application of the method, the most important limitation is that only real time data can be used. This limits the type of data that that can be used, as most of the data we used here are created years after collecting the satellite or field observations.

**3 Anonymous Referee #3**

**3.1 General Remarks**

This manuscript entitled "Summarizing the state of the terrestrial biosphere in few dimensions" is well-thought and well-written, and fits the scope of Biogeosciences, so overall, I am favourable to get it published there. I do have some concerns which I would like to see addressed by the authors, and I also have several recommendations to improve the manuscript before getting it published. Please find these points below.

We thank the reviewer for the positive comment and hope that we can address all mentioned concerns and recommendations.

**3.2** Better explanation for the interpretation**

My first point regards the interpretation of the first to PCA components. Having the first related to productivity and the second to water availability is indeed interesting and useful to summarize that state of vegetation. However, I believe some more effort is needed to more clearly separate these 2 in their interpretation. Productivity is inevitably dependent on water availability, so in principle, one wonders why these would be the first 2 components, which by definition should be orthogonal and 'unrelated'. I suppose this is perhaps because these refer to signals at different scales, PC1 describing an overall general state of potential productivity of the system at that location, while PC2 describes more events of water shortages and or excesses that are not directly related to the stationary potential productivity. Am I correct? Could you please clarify/elaborate on this to help readers better understand how these two axes should be 'read'.

Much related to the previous point, isn't it surprising that the 2 first principal components have such similar spatio-temporal patterns in Figure 3? These seem very highly correlated, which is something I would not have expected from the first two components which explain the maximum of variance in two orthogonal direction. Can you help me grasp this apparent paradox? In a way having such similar patterns make me wonder how useful having 2 PC is instead of only 1? Of course you do show the value of the 2D space in figure 2, but even there, much of the variation goes along the PC1 axis. Your selected cases in the anomalies in Figure 5 also generally go in the same direction of lower productivity coinciding with dryer conditions (Russian heatwave, droughts in Amazon), or vice versa (Floods in horn of Africa). Perhaps a stronger focus in general throughout the paper should be made on highlighting the much more specific cases where the two PCs give different but complementary information rather that going in the same direction.

While the reviewer is right that ecosystem productivity is dependent on water availability, the availability of water can be restricted due to several reasons which are reflected by PC2. We have added extensive descriptions to the text.

**3.3 Explain component 3**

I think you should also explore the third component. It does represent 9% of the variance, which is not so little, but above all it seems to be quite different from the first 2 in that it reacts much more to the albedo, which you hardly mention in the entirety of the manuscript. Could this be related to biophysical effects that vegetation could have on the climate? E.g. to understand where radiative vs non-radiative mechanisms dominate their effect on local temperature, for instance.

This is really a good suggestion and we added component three and a comprehensive interpretation.

**3.4 Include static variables?**

The behavior of the biosphere is much related to the elevation. While I know the effect of elevation should be reflected in the other variables, this is still dependent of modelling assumptions that may end up diluting the effect of elevation. Yet elevation is a variable that is very well measured, and which could contribute to summarizing the terrestrial biosphere. So why not including such a variable in the PCA? I know changes in elevation are minimal (and probably very difficult to detect) and having a static variable with respect to all the other dynamic ones you propose is a bit odd, but still, what are your arguments for not doing so? I think some discussion on this is warranted.

We thank the reviewer for this suggestion, but we are only including variables that are affected by the biosphere, it is true that elevation has a strong effect on the biosphere, the biosphere has no impact on elevation (excluding long term effects, such as erosion).

**3.5 General structure**

The paper generally could be improved by curating more the structure. Several points on this:

We thank the reviewer for these suggestions and hope that we have addressed satisfactorily.

• Section 3.2 could benefit from some introduction naming what you intend to calculate first (get trends, test significativity, get breakpoints, hysteresis) before going in the details. This part could also be more pedagogic, providing more rational on why you do these things.

We have done extensive restructuring of the text and hope that this solved this problem.

• Parts of the 'discussion' should be much further after the 'results', such as lines 155-162 which should come in some kind of 'caveats and perspective about the method' section

We have done a lot of text reorganization and hope that we have addressed this issue.

• Section 3.2 is very unbalanced with respect to 3.1. Probably best to reorganize to avoid 'sub-sub-sections' and have subsections from 3.1 to 3.5

We have done a lot of reorganization of the text and hope that the text is better balanced now.

• Parts describing concepts, such as Hysteresis (lines 235-246) should not appear in the results but before, either in methods or introduction.

We have moved the description of hysteresis into the "Methods" section, see comments to previous reviewer.

**3.6 Minor stuff**

Lines 74, 75: how do you manage intermittent gaps in the data? Does this affect your averages and your normalization? Also, please clarify if the

normalization is based on the entire data cube for each variable, or is the normalization done per time frame?

This was not entirely clear in the original text, we thank the reviewer for pointing this out. Normalization for each variable is done globally, pixels with missing values are ignored. We have added an explanation to the "Data"-section.

Line 182: don't you mean sensible heat instead of latent heat?

Yes, thank you for noting this, changed.

Figure 1: caption could be more instructive, perhaps somehow say there what the reader should understand/read from the "rotation matrix".

Thank you for pointing out that the term rotation matrix may not be understood by everyone. We have changed the caption to be more instructive.

Figure 7: surprised to see the strong pattern in Eastern Australia. Is this corroborated in other studies?

This is indeed interesting, we added a paragraph describing the reasons for this particular trend to the "Trends in Trajectories" section: "In eastern Australia we find a strong wetness and greenness trend which is due to Australia having a "milennium drought" since the mid nineties with a peak in 2002 (Nicholls, 2004; Horridge et al., 2005) and extreme floods in 2010–2011 (Hendon et al., 2014)."

Mention the time period for trend analysis. Regarding all trend analyses, make sure you more clearly mention in the captions the extend of the period you are considering, as these are not long-term trends and could thus be misinterpreted.

Good point, added the year to the captions of the figures related to trends.

Add contour for coast lines For clarity and readability, figures with maps could benefit from either a dark background on the oceans or a line vector showing the coasts, as many of the colour scales use very light colours which are confounded with the white background.

Done, improved the figures quite a bit, thanks for the suggestion!

Move breakpoint detection to SI, including description I wonder if the breakpoint detection is really useful if it is not more mentioned and elaborated in the main text and just left in appendix. I would recommend to bring it in as a main figure if something strong can be extracted from there, and otherwise remove it entirely from the methods. Eventually you could include it in supplementary, but then include the description of the breakpoint methodology only there.

The breakpoint detection is an example analysis that showcases one of the possible set of changes that can occur and that can be detected, therefore we think it has it's place in the paper as an example what can be possible, without going into too much detail.

Move Fig C1 into the main text On the other hand, I would strongly recommend to integrate the Figure C1 in the main text as you do talk in detail about the Bowen ratio and how the 2 PCs do characterize it well.

Thank you for the good suggestion, we have moved the figure into the main text.

**Unify scale ranges for fig D1** Figure 1D I have a bit of a hard time to make good use of it as it is. Are the values in normalized units or absolute values? Would it not be prefereable to have the same scale for MSC min and MSC max? Do you refer to this figure in the main text.

Thank you for this suggestion, but this figure is entirely about showing, that very different ecosystems can be very similar at certain points in time, for this, we don't need to compare across subfigures and therefore a single scale won't help for this, they will just remove contrast, especially across MSC min and MSC max.

Typos There are some typos in several places. Make sure to address them.

We have fixed many and hope we did not forget any.

**4 Anonymous Referee #4**

**4.1 General remarks**

The authors present a well-written manuscript on the analysis of two principal components derived from a set of biosphere variables, one related to vegetation productivity and the other one related to water stress. The trajectories of those components over time reveal interesting seasonal patterns, inter-annual changes and anomalies, and can be used to track extreme events and state shifts of ecosystems/biomes. Therefore, I believe that this is a novel and relevant contribution to Biogeosciences.

The authors thank the reviewer for the his time and thorough comments, we think that the comments greatly improved the manuscript. We have addressed them below.

**4.2 Major concern**

**4.2.1 Advantage of PCA**

My major concern lies in the fact that the authors select mainly variables related to productivity and water availability, and thus not surprisingly the PCA shows those two major axes. I wonder whether just selecting for example GPP and evaporative stress for the analysis of time trajectories would give the same results, but it might be easier to interpret than principal components representing a mix of variables. Can the authors elaborate in more depth what is the advantage of using PCs in this context?

There are multiple advantages,

- Having to observe less dimensions: We can quantify the number of dimensions we have to measure. If we simply take GPP, we don't know how much of the variance of the dataset we are explaining.
- Information on the covariance structure of the covariates.
- If some event happens only on one of the variables driving a component, then it can still be observed in the final component.
- Directional information, when observing extremes.

**4.2.2 More data streams**

For describing the state of the terrestrial biosphere, I think the authors are missing a very important component related to biodiversity, habitat quality, intactness, forest degradation and fragmentation. These aspects are crucial to describe the state of the terrestrial biosphere. There is still research needed to develop these as operational data streams, but a few examples are available at least at one point in time, e.g. Global Habitat Heterogeneity from EarthEnv, datasets from Global Forest Watch, Dynamic Habitat Indices DHI from Silvislab. This might not be sufficient (in terms of temporal resolution) to include it for this analysis, but the results from this study could be compared to those datasets (especially the DHI) and the need and relevance of global biodiversity and habitat intactness/quality information should be discussed.

We think that the reviewer has a very relevant point here, we would have loved to include more data streams that are relevant to the biosphere. The major problem is the availability of relevant of open data streams at a sufficiently high resolution in space and time which is currently very limited. As we want to track the change of the indicators over time, including static variables did not really make sense in this analysis. Including variables that have a yearly temporal resolution would require to aggregate our data by year which would also have made for a very interesting analysis but outside of the scope of this study.

**4.3 Minor comments**

L18: new satellite missions, add: Schimel, D., Schneider, F., Bloom, A., Bowman, K., Cawse-Nicholson, K., Elder, C., . . . Zheng, T. (2019). Flux towers in the sky: global ecology from space. New Phytologist, nph.15934. https://doi.org/10.1111/nph.15934

**added**

L25: green revolution, add: Chen, C., Park, T., Wang, X., Piao, S., Xu, B., Chaturvedi, R. K., . . . Myneni, R. B. (2019). China and India lead in greening of the world through land-use management. Nature Sustainability, 2(2), 122–129. https://doi.org/10.1038/s41893-019-0220-7

**added**

- L27: changes are not only occurring in the onset of spring, but also browning trends, see:
  - Garonna, I., de Jong, R., de Wit, A. J. W., Mücher, C. A., Schmid, B., & Schaepman, M. E. (2014). Strong contribution of autumn phenology to changes in satellite-derived growing season length estimates across Europe (1982 - 2011). Global Change Biology, 20(11), 3457–3470. https://doi.org/10.1111/gcb.12625
  - Garonna, I., de Jong, R., & Schaepman, M. E. (2016). Variability and evolution of global land surface phenology over the past three

decades (1982-2012). Global Change Biology, 22(4), 1456–1468. https://doi.org/10.1111/gcb.13168

Thanks for the suggestion, we have added the suggesting changes.

L35: if a principal component is a mix of productivity measures, I don't necessarily think it's more intuitive to interpret than a simple GPP map.

Thanks for pointing this out, changed the sentence to:

The rationale is that dimensionality reduction only retains the main data features, which makes them easier accessible for analysis.

L63: What do you mean by "of parts"? Parts of what?

We have changed "parts" to "observations" to clarify the sentence.

**L75:** Isn't this dependent on the coordinate system and/or projection? What is the coordinate system used? And why not try to use an equal-area projection (e.g. equal earth projection)?

We have added the coordinate system, thank you for pointing out this oversight.

L152: So what is contributing to the third component. It's still 9% of explained variance!

Thank you for pointing this out, we have added the third component to the manuscript.

L162: Figure 1b is not very intuitive to me. What exactly does it show and how do you read from this that the first component represents productivity and the second hydrology? The figure doesn't seem to show any clear patterns to me. Could you also show the biplots of PC1 and 2, and PC2 and 3?

As biplots are the "standard" way do describe this type of information, we have thought about adding biplots, but decided against it for the following reasons: 1) Biplots don't really contain any information that is not already contained in fig 2b and fig. 4. 2) The number of observations is so high that it would be impossible to add all the observations to a plot: we worked our way around this by showing bivariate histograms as a background shading in fig. 4, and 3) the manuscript contains too many figures already and adding even more would hamper readability.

**L177/178: check spelling**

Thanks for finding this, fixed!

**Figure 2:** Very interesting figure! A degraded or stressed system might show different trajectories, could you somehow visualize the difference between intact and degraded ecosystems?

Thank you for the positive comment, in this figure we are trying to show trajectories that are diverse. You can see a comparison between a degraded and non-degraded trajectory in fig. 9a.

L258: check spelling

Thanks for finding this one. This sentence was changed in reply to another comment.

Figure 5: third line, the effects of the drought

Changed drought  $\rightarrow$  floods. Thank you for finding this mistake.

Figure 6: This figure is a bit confusing to me. Could you improve the legends? I don't see an increase in seasonal amplitude in 6a, but maybe I just don't read this figure correctly. (b-c-d) seem to show the mean seasonal cycle and an event, but what do we see in 6a?

Thank you for pointing out that this may be confusing, we have added an explanatory sentence to the caption.

L305: changes that occurring?

Thank you for finding this, this sentence was changed in reply to another comment.

- **L340:** Additional research is needed to better represent biodiversity, habitat quality and intactness, forest degradation and fragmentation, etc. . . See:
  - Jetz, W., Cavender-Bares, J., Pavlick, R., Schimel, D., Davis, F. W., Asner, G. P., . . . Ustin, S. L. (2016). Monitoring plant functional diversity from space. Nature Plants, 2(3), 16024. https://doi.org/10.1038/nplants.2016.24
  - Chiarucci, A., & Piovesan, G. (2019). Need for a global map of forest naturalness for a sustainable future. Conservation Biology, 00(0), cobi.13408. https://doi.org/10.1111/cobi.13408

• Nicholas C. Coops, Michael A. Wulder, (2019). Breaking the Habit(at), Trends in Ecology & Evolution, Volume 34, Issue 7, https://doi.org/10.1016/j.tree.2019.04.013.

We think that the reviewer has a very valid point here, it would be very desirable to include these variables into the analysis. Unfortunately these variables do not exist, yet.

L352: detected in aa similar fashion

Thanks for finding this one.

**Summarizing the state of the terrestrial biosphere in few dimensions**

Guido Kraemer1,2,3, Gustau Camps-Valls2, Markus Reichstein1,3, and Miguel D. Mahecha1,3

1Max Planck Institute for Biogeochemistry, 07745 Jena, Germany

[revised manuscript text omitted]

Table 1 gives an overview of the data streams used in this analysis (for a more detailed description in appendix see Appendix A). For an effective joint analysis of more than a single variable, the variables have to be harmonized and brought to a single grid in space and time. The Earth System Data Lab (ESDL; www.earthsystemdatalab.net; Mahecha et al., 2019) curates a comprehensive set of data streams to describe multiple facets of the terrestrial biosphere and associated climate system. The data streams are harmonized as analysis ready data on a common spatiotemporal grid (equirectangular 0.25° grid in space and

- 100 8 days in time, 2001–2011), forming a 4d hypercube, which we call a *data cube*"data cube". The ESDL not only curates Earth system data, but also comes with a toolbox to analyze this data efficiently. For this study we chose all variables available in the ESDL v1.0 (the most recent version available at the time of analysis), divided the available variable into meteorological and biospheric variables and discarded the atmospheric variables. We also discarded variables with distributions that are badly suited for a linear PCA (e.g. burnt area contains mostly zeros) and variables with too many missing values. The only dataset
- 105 that was added post hoc was fAPAR which represents an important aspect of vegetation which was not available in the data cube at the time on analysis (it is part of the most recent version of the data cube).

In this study, each variable was normalized globally to zero mean and unit variance to account for the differences in scales.Because the different units of the variables, i.e. transform the variables to have standard deviations from the mean as the common unit. Because in the equirectangular coordinate system used by the ESDL the area of the pixel changes with

110 latitude, the pixels were weighted according to the represented surface area. Spatiotemporal pixels with missing values were ignored in the calculation of the covariance matrix.

**2.2 Dimensionality Reduction with PCA**

As a method for dimensionality reduction, we used a modified principal component analysis (PCA) to summarize the information contained in the observed variables. PCA transforms the set of d centered and, in this case, standardized variables into a

115 subset of  $p_{\pm} \le d \ge d$ , principal components (PCs). Each component is uncorrelated with the other components, while the

first PCs explain the largest fraction of variance in the data.

The data streams consist of d = 12 observed variables at the same time and location. Each observation is defined in a *d*dimensional space,  $\mathbf{x}_i \in \mathbb{R}^d$ , and we define the dataset by collecting all samples in the matrix  $\mathbf{X} = [\mathbf{x}_1 | \cdots | \mathbf{x}_n] \in \mathbb{R}^{d \times n} \mathbf{X} = [\mathbf{x}_1 | \cdots | \mathbf{x}_n] \in \mathbb{R}^d$ . The observations are repeated in space and time and lie on a grid of lat×lon×time, which in our caseare n = #lat × #lon × #time = 720 ×

120 where # denotes the length. In our case, we have  $n = |lat| \times |lon| \times |
[revised manuscript text omitted]

Figure 2a shows the explained fraction of variance (Eq. 5) for the global PCA based on the entire data cube. We see that the first two The two leading components explain 73% of the variance from the 12 variables; additional components contribute
little < 10% explained variance relatively little additional variance (PC3 contributes 9%, all subsequent PCs less than 7%) each. This results in a "knee" at component 23, which suggests that two indicators are sufficient to capture the major global dynamics of the terrestrial land surfaceand therefore we focus on these, but we will also consider the third components in the following analyses - (Cattell, 1966).

9

Using PCA as a method for dimensionality reduction means that we are assuming linear relations among features. A

230

nonlinear method could possibly be more efficient in reducing the number of variables, but would also have significant disadvantages. In particular: nonlinear methods typically require tuning of specific parameters, objective criteria are often lacking, a proper weighting of observations is difficult, and it is harder to interpret the resulting indicators due to their nonlinear nature (Kraemer et al., 2018). The salient feature of PCA is that an inverse projection is well defined and allows for a deeper inspection of the errors, which is not the case for nonlinear methods due to the pre-imaging problem (Mika et al., 1999; Arenas-Garcia et al., ---

235

The contributions of each variable to the resulting indicators can be understood from the rotation matrix (Eq. 4, fig. 2b). The We estimated the reconstruction error sequentially up to the first three principal components (fig. 3). Regions that do not fit the model well show a higher reconstruction error. Considering one component only, highest reconstruction errors appear in high latitudes but decrease strongly with each additional component and nearly vanish if the third component is included.

**240 3.2 Interpretation of the PCA**

The first PC summarizes variables that are closely related to vegetation primary productivity to primary production (GPP, LE, NEE, fAPAR). These variables are related because they are all directly related to primary productivity., and therefore highly interrelated (see fig. 2b). The energy for photosynthesis comes from solar radiation, and fAPAR is an indicator for the fraction of light used for photosynthesisis given by fAPAR. Photosynthesis fixes carbon from gaseous CO2 producing sugars to . The

- 245 available photosynthetic radiation is used by photosynthesis to fix  $CO_2$  and producing sugars that maintain the metabolism of plants, this the plant. The total uptake of  $CO_2$  is reflected in GPP. However, the  $CO_2$  uptake is, which is also closely related to water consumption. The actual uplift of water within the plant is not only essential to enable photosynthesis, but also drives the transport of nutrients from the roots and is ultimately reflected in transpiration—together with evaporation from soil surfaces one can observe the integrated latent energy needed for the phase transition (LE). However, ecosystems also respire<del>and hence</del>
- 250 : CO2 is produced by plants in energy consuming processes as well as by the decomposition of dead organic materials via soil microbes and other heterotrophic organisms. This total respiration can be observed as terrestrial ecosystem respiration (TER). The difference between GPP and TER is the net ecosystem exchange (NEE) rate of CO2 between ecosystems and the atmosphere (Chapin et al., 2006), and both variables are also well represented by the first dimension.
  On the second axis we observe variables that are The second component represents variables related to the surface hydrology.
- 255 of ecosystems -(see fig. 2b). Surface moisture, evaporative stress, root-zone soil moisture, and sensible heat, are all essential indicators for the state of plant available water. While surface moisture is a rather direct measure, evaporative stress is a modeled quantity summarizing the level of plant stress, a: A value of zero means that there is no water available for transpiration, while a value of one means that transpiration equals the potential transpiration (Martens et al., 2017). Root-zone soil moisture is the moisture content of the root zone in the soil, the moisture directly available for root uptake. If this quantity is below the wilting
- 260 point, there is no water available for uptake by the plants. Sensible heat is the exchange of energy by a change of temperature, if there is enough water available, then most of the surface heat will be lost due to evaporation (latent heat), with decreasing water availability more of the surface heat will be lost due to latent sensible heat, making this also an indicator of dryness.

---

## Author Response (AR2)

**Replies to the Anonymous Referees**

Guido Kraemer      Gustau Camps-Valls      Markus Reichstein

Miguel D. Mahecha

February 21, 2020

We thank the editors and anonymous reviewers for their input and devoted time to our work. We are happy that we could address the concerns of the reviewers and thank them for their positive and constructive feedback. We have revised the manuscript according to the suggestions of the reviewer, see the point by point reply below. Reviewers' comments are in normal font, replies are in *italics and blue*. Additionally, some small improvements have been made to the manuscript and errors have been corrected. For a full list of changes, please see the attached document at the end of this document. We hope that with all these changes the manuscript meets the high quality standards of the journal.

Kind regards,

The Authors

**Anonymous Referee #3**

The manuscript has been considerably improved after the first round of review, including thanks to the considerable restructuring of the text. There are still the following issues that should be addressed before publication.

*We are happy that we could address the concerns of the reviewer of the previous round of reviews and hope that we can address the remaining ones in this round of reviews.*

The paper is a lacking a discussion of some of the caveats of the input variables used in the analysis. This does not undermine the methodological contribution of the work, but rather affects the interpretation the authors make of it. This should probably include some statements on how the data from Tramontana et al for instance depend strongly on the spatial distribution of fluxtowers, which typically represent better temperate regions (due to their higher number there) than tropical ones for instance.

*We thank the reviewer for pointing this out and have added a paragraph to the "Data" section that addresses this issue.*

Also, I believe that data is known to not be able to represent the CO2 fertilization effect. Could you comment on that?

*We thank the reviewer for bringing this up and it is correct that CO2 is not amongst the predictor variables of the data product used, therefore it is sometimes criticized for not being able to model CO2 fertilization. CO2 fertilization is an important factor in greening (see e.g. Zhu et al., 2016) and therefore appears as a signal in predictors used for the modeling of GPP. We agree that a direct modeling of GPP using CO2 would be better.*

Finally, I would also add some words on the possibilities of now using sun-induced chlorophyll fluorescence (SIF) to better estimate GPP and photosynthesis, for instance. This could all be part of the conclusion, but should be there in my opinion.

*We agree with the reviewer that SIF should be included in such an analysis, but SIF datasets for the time range of the analysis (2001–2011) are not available in sufficient quality or resolution.*

In the conclusion suggest how these 3 principal components you detect could perhaps be used for operational monitoring of the state of the terrestrial biosphere. Could you link this to the Copernicus Climate Change Services (C3S) for instance?

*It would be possible and desirable to link the components to a service like C3S. Currently the problem is that C3S does not provide the data streams used in our analysis. A similar analysis would have to be carried out on the available data streams or we will have to wait until most of the data streams of the current analysis are available on C3S.*

I have the impression I did not find the temporal range of the data cube explicitly in the text (e.g. in the data section description).

*It says so in the "Data" section: 'The data streams are harmonized as analysis ready data on a common spatiotemporal grid (equirectangular 0.25° grid in space and 8 days in time, 2001–2011), forming a 4d hypercube, which we call a "data cube".'*

Furthermore, if I understand correctly from figures later on, it ends around 2012? How do you justify that in 2020 we would not have a time record that lags by 8 years? I know some datasets are not all open access and free in the same way, but now a days, most of the variables are effectively available beyond 2012 (even if from other sources than those used by the authors). Please add somewhere more justification explaining this, perhaps precising that you are doing a demonstration and do not need the latest years. But in the conclusion, stress that this work should be extended up to present times.

*It is true that most of the data streams should be available until a more recent data by now, but at the time we started the analysis, they where not. The resulting PCs will probably not change much. We expect the most pronounced differences for the trends, e.g. we show a wetness trend is parts of Australia, which is due to drought during the first years of our analysis and a more rainy epoch during the last years of the analysis.*

It should be specified up-front (from the abstract and throughout) that albedo refers to surface albedo, and not planetary albedo (which is much different due to cloud cover). Also, please specify somewhere that this is broadband albedo (i.e. across the shortwave spectrum), and that this combines both the albedo from the visible range and then near-infrared, two components that react quite differently with respect to vegetation (darkening the first and lightening the second).

*We agree with the reviewer that we should be more specific about this, therefore we have extended the description of black and white sky albedo in Appendix A, and specifically use "surface albedo" in the abstract and the first occurrence in the text to avoid confusion.*

L40: explain here what the Bowen ratio is to be more complete in your illustration

*We are referencing the section now. Thank you!*

L61: Land does not need capital letter... also, why not add a reference to this phrase? And event to the next?

*Corrected and added the missing citations. Thanks for pointing this out!*

L100: to be clear, please confirm in the text: if there is one missing data in one variable, all are considered missing, correct?

*Corrected, this should be more explicit now.*

L158: correct "deviatiosn"

*Corrected. Thank you!*

L176: correct "booth"

*Corrected. Thank you!*

L181: you have not yet said you consider only 3 components, so here there is an issue with specifying up to PC3 only... make it generic

*Corrected, also removed the mentions of PC1–3 two lines above.*

L207: correct "whiche"

*Corrected*

Why do Hovmoller plots in B1 have so many gaps? If this is due to occasional data gaps can't more tolerance be added?

*They have so many gaps because the datasets are land-only and therefore an ocean pixel will cause a complete horizontal line of missing values. We chose the longitudes for the Hovmöller plots to be the same as in fig. 8 for consistency and not to minimize the amount of missing values.*

**Anonymous Referee #4**

The authors addressed all my concerns and improved the manuscript substantially. Therefore, the manuscript could be published after minor revisions (mainly spell checking, and a thorough proof reading is necessary).

*We are happy that we could address all previous concerns of the reviewer and hope that we catch all remaining errors. Some of the line numbers mentioned by the reviewer are off. In this case, we tried to guess the correct line numer.*

L9-10: I still think that those variables do not fully describe the state of the biosphere, so it seems more accurate to write: account for 82% of the variance of the selected biosphere variables in space and time

*The reviewer is right that the selected variables do not fully describe the state of the biosphere (which is impossible) and we have corrected the phrase accordingly. The sentence before narrows down the parts of the biosphere we are describing.*

L13: Rephrase 'they also allow us'. What does they refer to (the anomalies, or the indicators/pcas)?

*Clarified, thanks.*

L19: add: Díaz, S., Settele, J., Brondizio, E. S., Ngo, H. T., Agard, J., Arneth, A., . . . Zayas, C. N. (2019). Pervasive human-driven decline of life on Earth points to the need for transformative change. Science, 1327(December). https://doi.org/10.1126/science.aaw3100

*Added, thank you for making us aware of the publication.*

L41: change

*Corrected.*

L41: what do you mean by qualitative state?

*We have added an example to clarify the meaning of "qualitative state".*

L49: decide whether to use 'high-dimensional' or 'high dimensional'

*Corrected, it should be "high-dimensional".*

L51: known

*Corrected.*

L78: rephrase, maybe just simplify to 'and analyses such as mean seasonal cycles . . . '

*Rephrased.*

L99: for this study,

*Corrected, thank you.*

L102: available variables

*Corrected, thank you.*

L106: of analysis

*Corrected, thank you.*

L109: rephrase: Because the area of the pixel changes with latitude in the equirectangular coordinate system used by the ESDL

*Rephrased, thank you.*

L142: remove 'n'

*Corrected, thank you.*

L145: hols?

*Corrected, thank you.*

L149: using

*Corrected, thank you.*

L197: something is missing in this sentence

*Corrected, thank you.*

L202: Rephrase and clarify the first sentence. What type of dynamics are you referring to?

*We have rephrased the sentence and hope that it is more understandable now.*

L216: Rephrase to be more readable (and add the section numbers accordingly): In the following, we first briefly present and discuss the quality of the global dimensionality reduction and interpret the individual components from an ecological point of view. We summarize the global dynamics that we uncovered in the low-dimensional space. We characterize the contained seasonal dynamics, including spatial patterns of hysteresis. . . . finally, we present global trends and their breakpoints.

*Rephrased, thank you.*

Figure 2b) I agree that it makes sense to visualize the variable loadings to the PCs in this way, but it is still pretty hard to read the contributions for

the first three components, because the values all lie close together at maybe around -0.5 to 0.5. Is there a way to rescale the colorbar/colormap (are the extremes of -1 and 1 even reached by the data range)? Or choose a colormap, which is more distinguishable at the lower values.

*We have increased the contrast by limiting the color bar range between* $-0.5$ *and* $0.5$*. See the updated figure caption for details. We thank the reviewer for this suggestion.*

L241-246: Check spelling of that whole paragraph!

*We have rephrased the paragraph. Thanks for pointing this out!*

Figure 3: This figure could be improved: add titles for top, middle and bottom rows, don't cut the top MSE distribution at 1.2, or maybe use log-scale if needed, and I think it would be ok to use different value ranges for each subfigure (b-d-f) to illustrate relative error distributions, since the color already indicates the absolute differences.

*Thanks for pointing this out. We simply don't limit the range of the x-axis of the plots any more, the maximum range is only slightly above* $1.5$*. We have also changed the placement of the labels.*

L263: Be more precise about the definition of albedo! See the following paper describing the differences between black-sky and white-sky albedo: Schaepman-Strub, G., Schaepman, M. E., Painter, T. H., Dangel, S., & Martonchik, J. V. (2006). Reflectance quantities in optical remote sensing-definitions and case studies. Remote Sensing of Environment, 103(1), 27–42. https://doi.org/10.1016/j.rse.2006.03.002

*Thank you for the suggestion, we have made a number of changes to the manuscript and are now more precise about albedo.*

L265: can be caused by many reasons

*Corrected, thank you.*

L279/280: Check spelling!

*Corrected, thank you.*

L325: check spelling

*Corrected, thank you.*

L338: third component

*Corrected, thank you.*

L340-341: It is indeed hard to say what exactly PC3 represents, especially since the northern boreal forests should show a very low albedo, especially during summer. There are a few other aspects though, potentially related to bare soil albedo (using cover-crops vs bare soil in agricultural practices) and changing bare soil albedo due to soil wetness. So if there is a large soil fraction, PC2 and 3 could be related due to lower surface albedo of wet soils.

*We think and have expressed so that the most important factor (as evidenced in the much lower reconstruction error in high latitudes in figure 6) is the binary decision between snow and no snow. There may indeed be other factors that also play a role but their contribution seems to be very low. The resolution of the dataset is 0.5° and therefore areas with homogenous cultural practices have to be quite large in order to show a clear and strong signal.*

Figure 6: nice figure, but I would also optimize the color range of column two and three. There is only light colors, which indicates that a smaller range could show the patterns more clearly (e.g. just disregard extreme values when defining the color-range).

*We have limited the range of the color scale to increase contrast. See updated figure caption for details.*

L355: It seems to track surface reflectance changes, which are not caused by vegetation phenology (since that is included in PC1). So maybe soil albedo plays a role too.

*The reviewer is right that most of the albedo change in PC1 is caused by vegetation phenology. Nevertheless, this does not necessarily mean that vegetation may not cause changes to PC3. E.g. nonlinear relations between variables cannot be modeled by PCA.*

*Looking at maps of soil albedo (e.g. Houldcroft et al. (2009)) we cannot see and obvious correlations between any of our maps and the map of soil background albedo in Houldcroft et al. (2009). This does also not mean that soil albedo does not play a role, as soil albedo can easily be masked by vegetation but attributing changes in PC3 to soil albedo would be purely speculative.*

L395: instead of

*Corrected, thank you.*

L402 onwards: check spelling

*We have corrected some mistakes in the paragraph and removed a sentence that did not make sense.*

L488 onwards: split into several sentances

*Corrected, thank you.*

L532: formed by

*Corrected, thank you.*

L553: three indicators

*Corrected, thank you.*

L553/554: the second ... while the third

*Corrected, thank you.*

L566: Future research should consider non-linearities, adding data streams describing other important biosphere variables (e.g. related to biodiversity and habitat quality), and including different subsystems...

*Corrected, thank you.*

[revised manuscript text omitted]

---

## Author Response (AR3)

**Point by point reply**

Guido Kraemer    Gustau Camps-Valls    Markus Reichstein
Miguel D. Mahecha

March 26, 2020

We thank the editors and anonymous reviewers for their input and devoted time to our work. We are happy that the manuscript now conforms with the high quality standards of the journal. We have uploaded the code to reproduce the analysis and have added this to the "Code availability" section. We also thank the anonymous reviewers for their efforts in the manuscript. Apart from this, the manuscript has not been changed.

Kind regards,

The Authors

[revised manuscript text omitted]